# Potent but transient immunosuppression of T-cells is a general feature of CD71$^+$ erythroid cells

Tomasz M. Grzywa [1,2,3], Anna Sosnowska[1,4], Zuzanna Rydzynska[1,15], Michal Lazniewski[5,6], Dariusz Plewczynski [5,7], Klaudia Klicka [2,8], Milena Malecka-Gieldowska [9], Anna Rodziewicz-Lurzynska[10], Olga Ciepiela [9], Magdalena Justyniarska[1], Paulina Pomper[11], Marcin M. Grzybowski[11], Roman Blaszczyk [11], Michal Wegrzynowicz [12], Agnieszka Tomaszewska[13], Grzegorz Basak[13], Jakub Golab [1,14✉] & Dominika Nowis [1,3✉]

CD71$^+$ erythroid cells (CECs) have been recently recognized in both neonates and cancer patients as potent immunoregulatory cells. Here, we show that in mice early-stage CECs expand in anemia, have high levels of arginase 2 (ARG2) and reactive oxygen species (ROS). In the spleens of anemic mice, CECs expansion-induced L-arginine depletion suppresses T-cell responses. In humans with anemia, CECs expand and express ARG1 and ARG2 that suppress T-cells IFN-γ production. Moreover, bone marrow CECs from healthy human donors suppress T-cells proliferation. CECs differentiated from peripheral blood mononuclear cells potently suppress T-cell activation, proliferation, and IFN-γ production in an ARG- and ROS-dependent manner. These effects are the most prominent for early-stage CECs (CD71$^{high}$CD235a$^{dim}$ cells). The suppressive properties disappear during erythroid differentiation as more differentiated CECs and mature erythrocytes lack significant immunoregulatory properties. Our studies provide a novel insight into the role of CECs in the immune response regulation.

[1] Department of Immunology, Medical University of Warsaw, Warsaw, Poland. [2] Doctoral School of the Medical University of Warsaw, Warsaw, Poland. [3] Laboratory of Experimental Medicine, Medical University of Warsaw, Warsaw, Poland. [4] Postgraduate School of Molecular Medicine, Medical University of Warsaw, Warsaw, Poland. [5] Laboratory of Functional and Structural Genomics, Centre of New Technologies, University of Warsaw, Warsaw, Poland. [6] Centre for Advanced Materials and Technologies, Warsaw University of Technology, Warsaw, Poland. [7] Faculty of Mathematics and Information Science, Warsaw University of Technology, Warsaw, Poland. [8] Department of Methodology, Medical University of Warsaw, Warsaw, Poland. [9] Department of Laboratory Medicine, Medical University of Warsaw, Warsaw, Poland. [10] Central Laboratory, University Clinical Center of Medical University of Warsaw, Warsaw, Poland. [11] OncoArendi Therapeutics, Warsaw, Poland. [12] Laboratory of Molecular Basis of Neurodegeneration, Mossakowski Medical Research Institute, Polish Academy of Sciences, Warsaw, Poland. [13] Department of Hematology, Transplantation and Internal Medicine, Medical University of Warsaw, Warsaw, Poland. [14] Centre of Preclinical Research, Medical University of Warsaw, Warsaw, Poland. [15] Present address: Department of Pediatrics, Oncology and Hematology, Medical University of Lodz, Lodz, Poland. ✉email: jakub.golab@wum.edu.pl; dominika.nowis@wum.edu.pl

CD71[+] erythroid cells (CECs) normally reside in the bone marrow and are progenitors and precursors to over $2 \times 10^{11}$ of oxygen-transporting red blood cells (RBCs) generated per day[1]. In mice, when steady-state erythropoiesis becomes insufficient to meet increased tissue oxygen demands, CECs are released from the bone marrow to the circulation and expand in the extramedullary hematopoietic sites. In humans, increased RBCs damage or loss of blood is compensated by increased erythropoietic activity in the bone marrow. Recent studies revealed an unexpected complexity of CEC functions. CECs arose as a relevant population of cells regulating immunity[2–5]. Initially, CECs were reported to suppress both innate and humoral immune responses in neonates[4,6,7] and it was suggested that their immunomodulatory functions are restricted to early life events[4]. However, further studies revealed a crucial role of CECs in the regulation of multiple phenomena such as fetomaternal tolerance[8], immune response in cancer patients[9,10], systemic inflammation in colitis[11], and anti-viral response in human immunodeficiency virus (HIV) infection[12], as well as SARS-CoV-2-induced disease (COVID-19)[13]. It has been reported that CD45[+] CECs induced by advanced tumors inhibit CD8[+] and CD4[+] T cell proliferation and impair antimicrobial immunity[10]. Interestingly, the authors demonstrated that CECs from mice with acute hemolytic anemia (HA), induced by systemic phenylhydrazine (PHZ) administration, are not immunosuppressive as compared with CECs from tumor-bearing mice[10]. This could lead to the conclusion that only CECs in newborns and patients with advanced cancer have robust immunosuppressive properties. In this study, we provide evidence that CECs in anemic mice do have immunoregulatory properties, but PHZ used to induce hemolysis affects the mechanisms of immune suppression used by these cells masking their phenotype. Moreover, we comprehensively elucidate the role of CECs in the regulation of immune response in both mice and humans and demonstrate that immunomodulatory properties of CECs are robust but transient and disappear during their maturation.

## Results

**Early-stage CECs expand in the spleens of anemic mice**. We initially compared the expansion of CECs in 3-day-old neonatal and adult anemic mice (Fig. 1a). Non-hemolytic anemia (NHA) was induced by phlebotomy and HA was induced either by administration of PHZ (HA-PHZ) or anti-TER119 antibodies (HA-TER119) (see Supplementary Fig. 1 for hematological parameters of these mice). Since in mice stress erythropoiesis rely on the erythropoietic activity of the spleen[14,15], we assessed CECs expansion in this organ. CECs expanded in the spleens of anemic mice as compared with controls but were significantly less frequent than in neonatal mice (Fig. 1b). However, CEC numbers in the spleen were substantially higher in anemic mice than in neonates or controls (Fig. 1c). The percentage of CECs increased also in the blood of anemic mice (Supplementary Fig. 2a) but remained unchanged in the bone marrow (Supplementary Fig. 2b). Recent studies indicated that CECs at the earliest stages of differentiation express CD45 and more potently suppress immune response[9,10]. The proportion of CD45[+] to CD45[−] CECs was the highest in HA-PHZ mice and the lowest in neonatal mice (Fig. 1d). Analysis of developmental stages of CECs based on cell size and CD44 levels (Fig. 1e)[16] revealed enrichment of less differentiated CECs in anemic mice compared to non-anemic controls (Fig. 1f and Supplementary Fig. 2c). These early-stage CECs expressed CD45 (Fig. 1g, h) and were predominantly erythroid progenitors before enucleation (Fig. 1i).

**The T cell immune response is impaired in anemic mice**. Next, we sought to determine whether the expansion of early-stage

CECs induced by anemia might impair the function of the immune system. To this end, we assessed selected functionalities of myeloid cells, B cells, and T cells in control and anemic mice. In contrast to neonatal mice[4,6], production of tumor necrosis factor-α (TNF-α) by splenic CD11b[+] cells after stimulation with heat-killed *E. coli* (HKEc) (Supplementary Fig. 3a, b) or the concentration of anti-ovalbumin (OVA) IgG antibodies after OVA-ALUM immunization (Supplementary Fig. 3c, d) was unimpaired in adult anemic mice as compared with healthy controls. Intriguingly, we found that the proliferation of adoptively transferred SIINFEKL-specific OT-I T cells in response to OVA stimulation was decreased in the spleen of NHA mice compared to healthy controls (Fig. 2a, b).

Since the expansion of CD71[+] cells was the most substantial in the spleens of anemic mice (Supplementary Fig. 3e) and the ratio of CECs number to T cells number was significantly increased in anemia (Supplementary Fig. 3f, g), we hypothesized that CECs might be responsible for T cells suppression. Indeed, CECs isolated from the spleens of both HA and NHA anemic mice (Fig. 2c and Supplementary Fig. 4a) suppressed the proliferation of CD4[+] T cells that were activated with anti-CD3/CD28 beads (Fig. 2d). Altogether, these data document a rather selective impairment of T cell response by CECs in anemic mice.

**Murine CECs have high ROS levels and express ARG2**. Both reactive oxygen species (ROS) generation and expression of L-Arg-degrading enzymes were previously identified as the effectors of the immunoregulatory activity of neonatal CECs[4,17]. Accordingly, we found that both cytoplasmic and nuclear ROS levels were higher in anemia-induced CECs as compared with mature RBCs (Fig. 3a and Supplementary Fig. 5a, b). ROS reached the highest values in the CECs at the earliest stages of their maturation (Supplementary Fig. 5c, d), i.e. in CD45[+] CECs (Supplementary Fig. 5e, f). Interestingly, in contrast to human CECs[13], ROS levels in murine CECs, including CD45[+] CECs, were significantly lower than in the cells of non-erythroid lineages such as myeloid cells or T cells (Fig. 3b).

Murine CECs expressed ARG2, a mitochondrial arginase isoform (Fig. 3c, d), but had almost undetectable cytosolic ARG1 based on intracellular staining (Fig. 3e) as well as enhanced yellow fluorescent protein (eYFP) signal in reporter B6.129S4-Arg1[tm1Lky]/J mice that express eYFP under *Arg1* promoter (Fig. 3f, g). Similar to ROS, the levels of ARG1 and ARG2 were the highest in early-stage CECs and consequently decreased during maturation (Supplementary Fig. 6a–g). Intriguingly, while the level of ARG2 (Fig. 3c), as well as the percentage of ARG2[+] CECs, were similar in all groups (Fig. 3d), the fraction of ARG1[+] cells was substantially higher in HA-PHZ mice as determined by intracellular staining (Fig. 3e). This finding seems counterintuitive considering that ARG-dependent degradation of L-arginine leads to T cell suppression[18,19], and we did not observe the suppression of T cells in HA-PHZ mice in vivo (Fig. 2b). Moreover, CECs from HA-PHZ mice exerted the weakest suppressive effects on T cell proliferation (Fig. 2d). Increased expression of ARG1 in HA-PHZ CECs was further confirmed by ARG1 mRNA detection (Supplementary Fig. 7a) and in reporter B6.129S4-Arg1[tm1Lky]/J mice (Fig. 3g, h) indicating that flow cytometry findings were not artifactual. HA-PHZ CECs had increased expression of ARG2 mRNA as compared with NHA CECs (Supplementary Fig. 7b), but no increase in ARG2 protein levels was observed (Fig. 3d). Surprisingly, despite robust upregulation of ARG1 levels, total arginase activity in both CECs isolated from HA-PHZ mice and CECs-conditioned medium was lower even than that in CECs from NHA mice

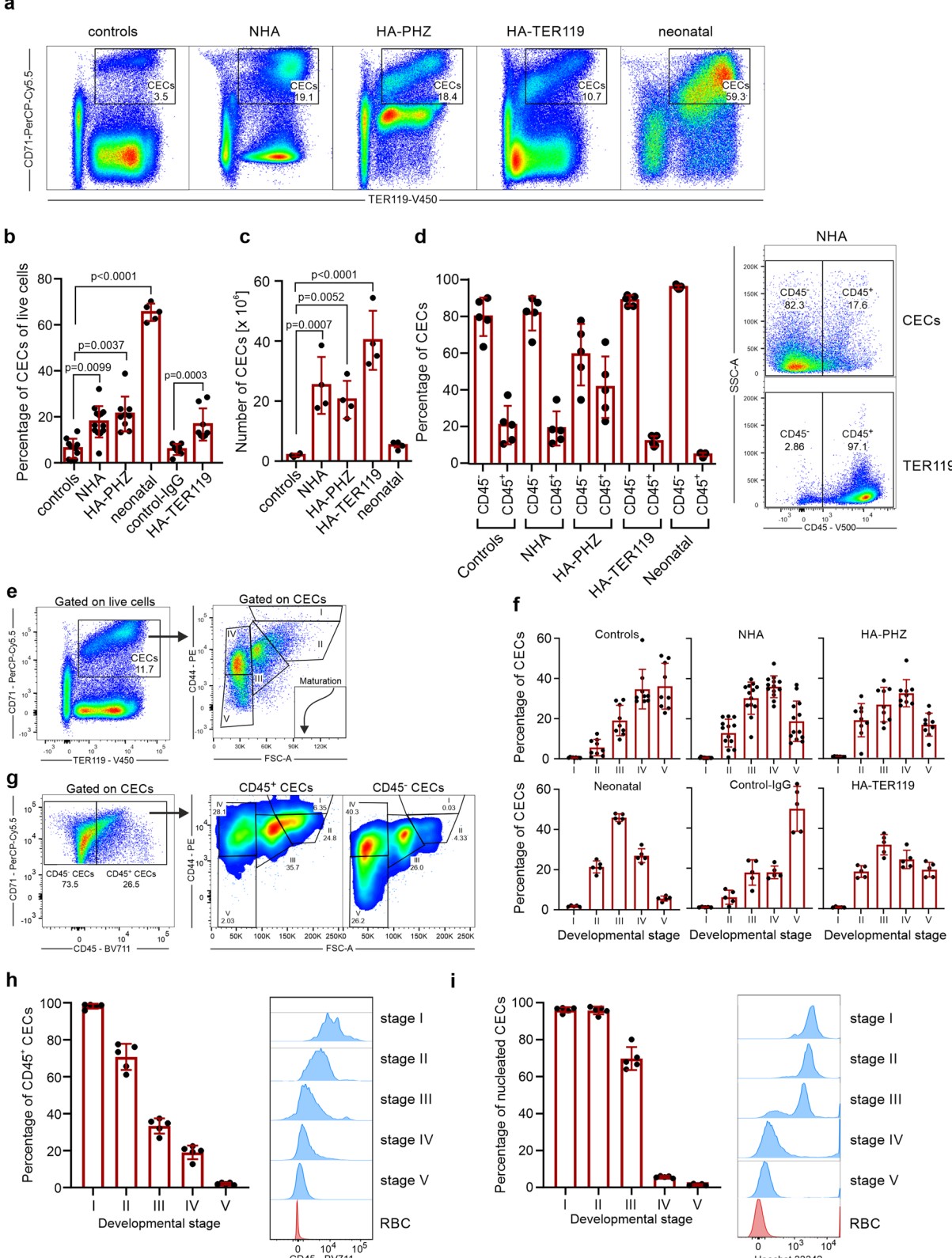

(Fig. 3i, j). Moreover, CECs cultured ex vivo in the presence of PHZ strongly upregulated ARG1 expression (Fig. 3k, l). Then, we sought to confirm whether PHZ is responsible for the attenuation of CECs immunoregulatory properties. Indeed, we found that CECs isolated from NHA lose their suppressive effects on T cells proliferation in the presence of PHZ (Supplementary Fig. 8a, b).

**PHZ targets arginase and suppresses its activity**. Increased expression with a concomitant decrease in arginase activity suggested an interaction between PHZ and arginase. Further studies showed that indeed PHZ inhibits the activity of recombinant human ARG1 and ARG2, with an $IC_{50}$ of 1017 and 61 μM, respectively (Fig. 4a). However, PHZ did not affect the production of nitric oxide (NO) by nitric oxide synthase, which is also

**Fig. 1 Anemia induces CECs expansion in the spleen. a** Representative plots of CD71+TER119+ CECs of total live cells in the spleens of control, anemic, and 3-day-old neonatal mice. **b** The frequency of CD71+TER119+ CECs of total splenocytes in control ($n = 10$), control-IgG ($n = 7$), anemic (NHA, $n = 13$; HA-PHZ, $n = 9$; HA-TER119, $n = 8$), and 3-day-old neonatal mice ($n = 5$). P values were calculated with Kruskal–Wallis test with Dunn's post hoc test. **c** Numbers of CD71+TER119+ CECs in the spleens of control ($n = 4$), anemic (NHA, $n = 4$; HA-PHZ, $n = 4$; HA-TER119, $n = 4$), and neonatal ($n = 4$) mice. P values were calculated with one-way ANOVA with Dunnet's post hoc test. **d** Percentages of CD45.2− and CD45.2+ cells within CECs (CD71+TER119+) population ($n = 5$). Representative plot of CD45 levels in CECs and TER119− cells in the spleen of NHA mouse. **e** Gating strategy for CECs developmental stages based on CD44 expression and cells size[16]. **f** Developmental stages of CECs in control mice ($n = 9$), NHA mice ($n = 13$), HA-PHZ ($n = 9$), HA-TER119 ($n = 5$), and neonatal mice ($n = 5$). **g** Representative plot of CD71 and CD45 levels in CECs in the spleen of NHA mouse and analysis of developmental stages of CD45+ CECs and CD45− CECs. **h** Percentages of CD45+ CECs in different developmental stages in the spleens of NHA mice ($n = 5$). Histograms show the fluorescence of CD45–BV711. Red blood cells (RBCs) are presented as a negative control. **i** Percentages of nucleated CECs (Hoechst 33342+) in different developmental stages in the spleens of NHA mice ($n = 5$). Histograms show the fluorescence of Hoechst 33342-INDO-1. Red blood cells (RBCs) are presented as a negative control. Data show means ± SD. Each point in **b–d**, **h**, **i** represent data from individual mice. n values are the numbers of mice used to obtain the data. The source data underlying **b–d**, **f**, **h**, **i** are provided as a Supplementary Data 2 file.

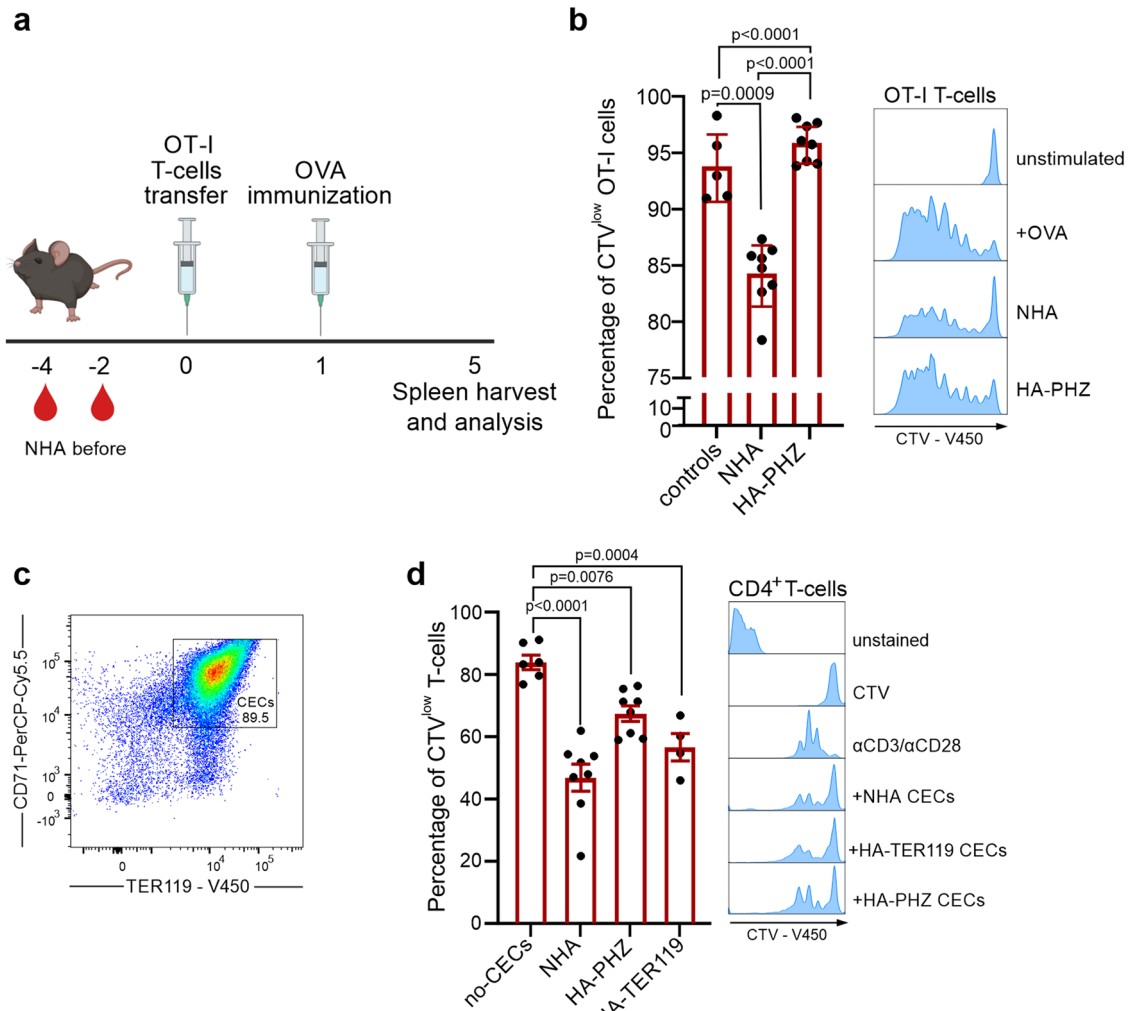

**Fig. 2 Anemic mice have impaired T cell immune response. a** Schematic presentation of the experimental setting. T cells isolated from OT-I mice were labeled with CellTraceViolet (CTV) and adoptively transferred to anemic and healthy control mice and stimulated with OVA. Scheme created using BioRender.com. **b** Percentage of proliferating (CTVlow) OT-I T cells in the spleen of NHA mice ($n = 8$), HA-PHZ mice ($n = 8$), and healthy controls ($n = 5$). Histograms show the fluorescence of CTV (CellTraceViolet)-V450 of OT-I T cells. P values were calculated with one-way ANOVA with Tukey's post hoc test. **c** Representative plot of CD71 and TER119 levels in isolated CECs. Additional plots are presented in Supplementary Fig. 4. **d** Proliferation triggered by αCD3/αCD28 in CTV-labeled CD4+ T cells co-cultured with CECs isolated from the spleens of NHA ($n = 8$), HA-PHZ ($n = 8$), or HA-TER119 ($n = 4$) mice. T cell:CECs ratio was 1:2. Representative proliferation histograms of αCD3/αCD28-stimulated CD4+ T cells co-cultured with CECs. Histograms show the fluorescence of CTV (CellTraceViolet)-V450. P values were calculated with one-way ANOVA with Dunnet's post hoc test. Data show means ± SD. Each point in **b**, **d** represents data from individual mice. n values are the numbers of mice. The source data underlying **b**, **d** are provided as a Supplementary Data 2 file.

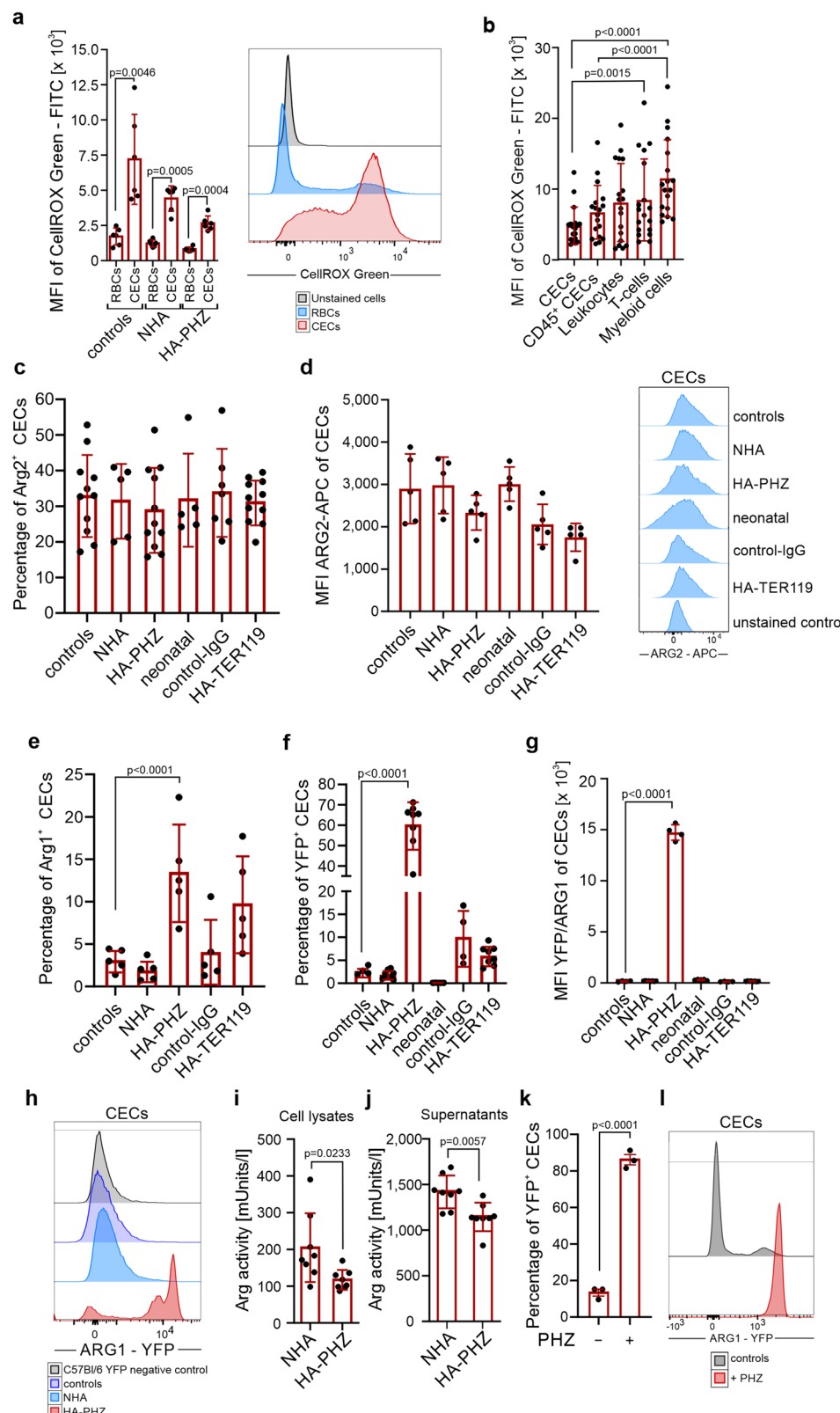

using L-arginine as a substrate (Fig. 4b). To elucidate how PHZ interacts with ARG1 and ARG2 a molecular docking simulation was carried out with PHZ, L-arginine, as well as 2-amino-6-borono-2-(2-(piperidin-1-yl)ethyl)hexanoic acid (ABH) that is a strong ARG1 inhibitor[20]. PHZ binds to the active sites of all arginases, where it forms several polar interactions involving D128, D232, or T246 (Supplementary Fig. 9a). Thus, it may block

the entry of other molecules to the active site. However, predicted binding energies suggest that among the tested ligands PHZ has the weakest affinity for arginases, and thus a significant concentration of this compound may be required to induce any biological effect, which indeed is the case in vivo. The transient nature of interactions between PHZ and arginases was also confirmed by a short 100 ns MD simulation (Supplementary

**Fig. 3 CECs from anemic mice express ARG2 and have high levels of ROS. a** Mean fluorescence intensity (MFI) of CellROX Green-FITC in CECs (CD71+TER119+) and RBCs (CD71−TER119+) of control ($n = 6$), NHA ($n = 6$), and HA-PHZ ($n = 6$) mice. Histograms show representative fluorescence of CellROX Green-FITC in CECs and RBCs from the spleens of the NHA mouse. *P* values were calculated with unpaired *t*-test. **b** MFI of CellROX Green-FITC in CECs (CD71+TER119+), CD45+ CECs (CD45+CD71+TER119+), leukocytes (CD45+TER119−), T cells (CD45+CD3e+), and myeloid cells (CD45+CD11b+) ($n = 18$). *P* values were calculated with Friedman's test with Dunn's post hoc test. **c** Percentages of ARG2+ CECs in control mice ($n = 11$), anemic mice (NHA, $n = 5$; HA-PHZ, $n = 11$; HA-TER119, $n = 11$), neonatal mice ($n = 5$), and isotype control-IgG-treated mice (control-IgG, $n = 7$) based on intracellular staining. **d** MFI of ARG2-APC in CECs from control mice, anemic mice (NHA, HA-PHZ, HA-TER119), neonatal mice, and isotype control-IgG-treated mice (each group $n = 5$). Histograms show the representative fluorescence of ARG2-APC in CECs in different groups and in anti-ARG2-unstained controls. **e** Percentages of ARG1+ CECs based on intracellular staining ($n = 5$). *P* values were calculated with one-way ANOVA with Dunnet's post hoc test and with an unpaired *t*-test for HA-TER119. **f** Percentages of YFP+ CECs in reporter B6.129S4-Arg1tm1Lky/J mice (controls $n = 4$, NHA $n = 8$, HA-PHZ $n = 8$, neonatal $n = 5$, control-IgG $n = 4$, HA-TER119 $n = 8$). *P* values were calculated with one-way ANOVA with Dunnet's post hoc test and with an unpaired *t*-test for HA-TER119. **g** MFI of YFP-FITC in CECs of reporter B6.129S4-Arg1tm1Lky/J mice in control mice ($n = 4$), anemic (NHA $n = 8$, HA-PHZ $n = 4$, HA-TER119 $n = 8$), neonatal mice ($n = 5$), and isotype control-IgG-treated mice ($n = 4$). *P* values were calculated with one-way ANOVA with Dunnet's post hoc test. **h** Representative fluorescence of ARG1-YFP in CECs in reporter B6.129S4-Arg1tm1Lky/J control mice and anemic mice (NHA, HA-PHZ). Background fluorescence of YFP in CECs from wild-type C57Bl/6 mice presented as a negative YFP control. **i, j** Total arginase activity in CECs lysates (**i** $n = 8$) or in the supernatants from CECs cultures (**j** $n = 8$). *P* values were calculated with one-way ANOVA with unpaired *t*-test. **k, l** Percentages of ARG1+ CECs (**k**) isolated from the spleens of B6.129S4-Arg1tm1Lky/J incubated with diluent or PHZ (100 µM for 24 h) ($n = 3$) and representative fluorescence of ARG1-YFP (**l**). *P* values were calculated with unpaired *t*-test. Data show means ± SD. Each point in **a–g**, **i–k** represents data from individual mice. *n* values are the numbers of mice used to obtain the data. The source data underlying **a–g**, **i–k** are provided as a Supplementary Data 2 file.

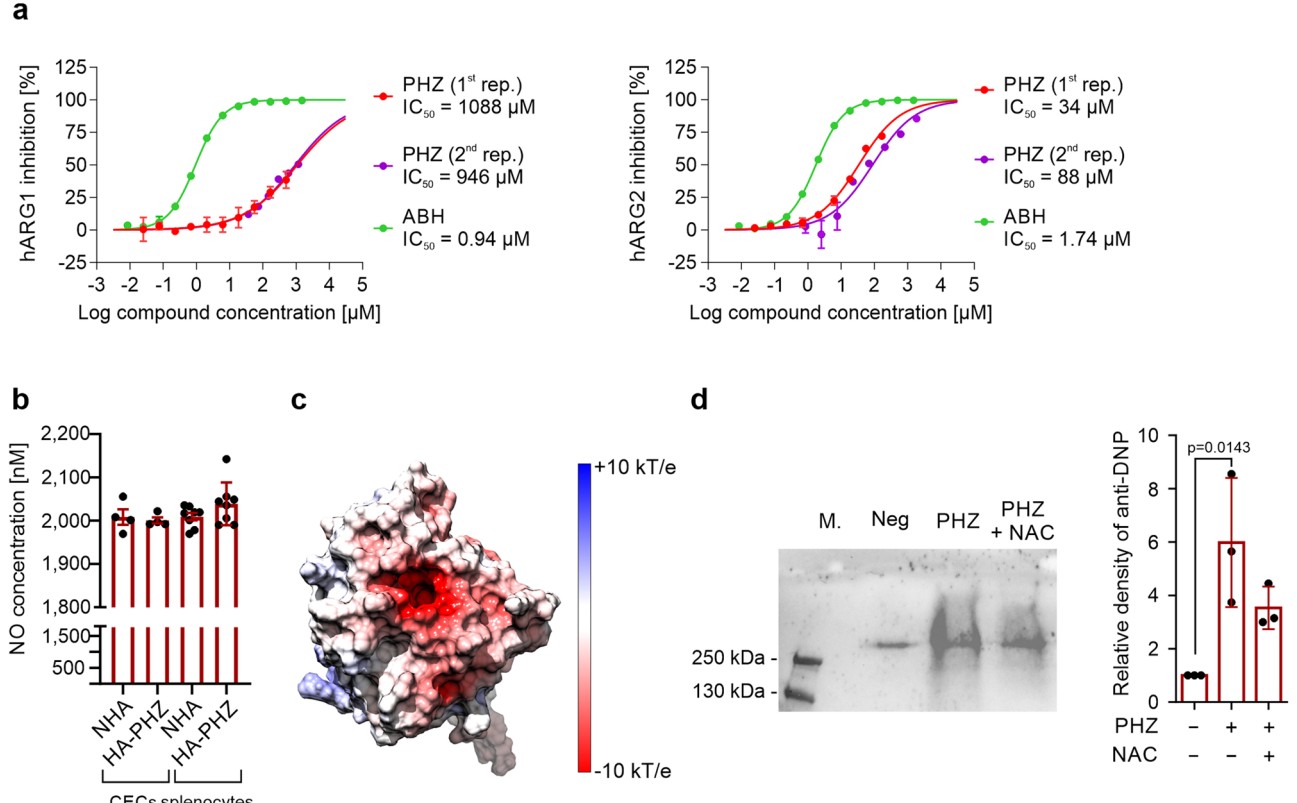

**Fig. 4 Phenylhydrazine targets arginase, inhibits its activity, and induces oxidative damage. a** Inhibition curves for recombinant human ARG1 and ARG2, and IC$_{50}$ values for PHZ ($n = 2$) and 2(*S*)-amino-6-boronohexanoic acid (ABH). **b** NO production from CECs and whole splenocytes population isolated from NHA ($n = 4$) and HA-PHZ ($n = 4$) mice. *P* values were calculated with an unpaired *t*-test. **c** The electrostatic surface potential of the human ARG1. The potential was calculated with APBS and projected onto the molecular surface of the protein. The figure was prepared with UCSF Chimera. **d** Carbonylation of ARG1 in the presence of PHZ ($n = 3$). Representative blot (left) and densitometric analysis done with ImageJ software (right). The negative lane represents ARG1 precipitates treated with 2 N HCl alone. *P* values were calculated with ordinary one-way ANOVA with Dunnett's multiple comparisons test. Data show means ± SD. Each point in **b** represents data from individual mice. *n* values are the numbers of mice used to obtain the data or the number of biological replicates of in vitro experiments. The source data underlying **a**, **b**, **d** are provided as a Supplementary Data 2 file.

Fig. 9b, c). The ligand remained bound to the active site for only 15–30% of the simulation time, despite its initial placement inside the ligand-binding pocket. The analysis of electrostatic surface potential revealed the presence of a large, negatively charged area around the substrate-binding pocket of ARG1 that likely plays a role in attracting positively charged L-arginine to the catalytic site (Fig. 4c). Since PHZ in the presence of oxygen leads to the formation of free radicals and hydrogen peroxide[21], we hypothesized that decreased ARG activity in CECs from HA-PHZ mice might emerge due to non-specific non-covalent interactions of PHZ

with the catalytic pocket of ARG1 that leads to oxidative changes in the enzyme, decreased activity, and compensatory increase in its expression. Indeed, the incubation of recombinant ARG1 with PHZ in the presence of oxygen led to a significant increase in the carbonylation of the enzyme. However, this effect was only slightly reduced by concomitant incubation with $N$-acetylcysteine (ROSi) (Fig. 4d). Moreover, ROS scavengers did not prevent ARG1 induction by PHZ in vivo (Supplementary Fig. 10a, b) nor in vitro (Supplementary Fig. 10c, d). Thus, we demonstrated that PHZ targets ARG leading to the diminishment of CECs immunoregulatory properties; however, the exact mechanism that would explain PHZ-mediated inhibition of ARG activity remains elusive.

**CECs degrade ʟ-Arg and produce ROS leading to the suppression of T cells.** Due to the interaction between PHZ and arginases, we chose NHA as a model of anemia-induced CECs for further studies. We found that CD4$^+$ T cells stimulated with anti-CD3/CD28 beads in the presence of CECs showed downregulation of activation markers CD25 and CD69, which was less pronounced for CD62L (Fig. 5a). Both arginase inhibitor (ARGi, OAT-1746, a membrane-permeable, potent inhibitor of both arginase isoforms[22–24]) and ROS inhibitor (ROSi, $N$-acetylcysteine) nearly completely restored the proliferation of T cells that was inhibited by co-culture with CECs isolated from NHA mice (Fig. 5b), similar to CECs isolated from neonates (Supplementary Fig. 11). Likewise, CEC-conditioned medium had a suppressive effect on T cell proliferation, and supplementation with either of ʟ-arginine or ARGi restored T cell proliferation to percentages akin to the control group (Fig. 5c).

To confirm that early-stage CECs that have the highest ROS levels (Supplementary Fig. 5) and ARG expression (Supplementary Fig. 6) have the most potent suppressive effects on T cells, we isolated the fraction of nucleated cells (erythroid progenitors, developmental stages I–III, Fig. 1i) from CECs using density-gradient centrifugation. We found that the whole CEC population suppressed CD4$^+$ T cells proliferation by 43% while isolated nucleated CECs (nCECs) completely inhibited it (Fig. 5d), confirming that they are responsible for the suppressive effects.

At a 1:10 of T cells to CECs ratio, similar to that observed in anemia (Supplementary Fig. 3f, g), CECs completely suppressed the proliferation of CD4$^+$ (Fig. 5e) and CD8$^+$ T cells (Fig. 5f). Further studies revealed that the expansion of CECs in anemic mice leads to the substantial increase of the total arginase activity (Fig. 5g). This effect was caused by an increased ARG2 but not ARG1 levels in the spleen (Fig. 5h–j). Even though the concentration of ʟ-arginine was only slightly decreased in the serum of anemic mice (Supplementary Fig. 12), their splenic CD4$^+$ T cells and CD8$^+$ T cells had decreased levels of CD3ζ (Fig. 5k, l), a marker of ʟ-arginine T cell starvation[22,25]. Accordingly, ex vivo stimulation of T cells with anti-CD3/CD28 beads in the presence of CECs resulted in a decrease in CD3ζ, which was prevented by ARGi and completely restored by the combination of ARGi and ROSi (Fig. 5m, n). Noteworthy, the decrease in CD3ζ was not observed in the lymph nodes of anemic mice, where CECs are a relatively rare population (Supplementary Fig. 13a–c). Altogether, these results show that CECs suppress T cells response in anemic mice via both arginase and ROS and their local accumulation in the spleen impairs T cell immunity.

To further study the role of ARG2 in the modulation of immune response by CECs, we assessed the suppressive effects of CECs isolated from anemic mice lacking functional $Arg2$ gene ($Arg2^{-/-}$, $Arg2^{tm1Weo}$/J mice[26]). $Arg2^{-/-}$ mice had effective stress erythropoiesis (Supplementary Fig. 14a). Despite a slightly increased percentage of ARG1$^+$ CECs compared to wild-type

mice ($Arg2^{+/+}$) (Supplementary Fig. 14b), no significant changes in total ARG1 levels were observed in these cells (Supplementary Fig. 14c). In contrast to wild-type mice, expansion of CECs in the spleen of anemic $Arg2^{-/-}$ mice was not associated with a significant decrease in CD3ζ in T cells (Fig. 6a, b). Moreover, CECs isolated from $Arg2^{-/-}$ mice had substantially diminished suppressive effects on T cell proliferation as compared with $Arg2^{+/+}$ CECs (Fig. 6c), confirming a critical role of ARG2 in the regulation of T cells function by murine CECs.

**CECs expand in the blood of anemic individuals and suppress IFN-γ production by T cells.** Then, we sought to investigate the role of CECs in anemic patients (Supplementary Table 1 and Supplementary Data 1). The percentages of CECs (CD71$^+$CD235a$^+$) in the peripheral blood were substantially increased in anemic patients compared to non-anemic control individuals (Fig. 7a, b). The number of CECs in the blood (Fig. 7c) reversely correlated with hemoglobin concentration (Fig. 7d) and was the highest in patients with moderate and severe anemia (Fig. 7e).

In anemic patients, CECs constituted a substantial fraction of peripheral blood mononuclear cells (PBMCs) (Fig. 7f, g) and were predominantly at the latest stages of differentiation with a very small percentage of CD45$^+$ CECs (Supplementary Fig. 15a, b). We found that the production of IFN-γ in response to CD3/CD28 stimulation was suppressed in T cells from anemic individuals when compared to non-anemic controls (Fig. 7h, i). However, T cells proliferation (Supplementary Fig. 15c, d) or the production of TNF-α by myeloid cells in response to killed bacteria (Supplementary Fig. 15e) were comparable in anemic patients and control individuals.

**CECs from human bone marrow suppress T cells proliferation.** Since the expansion of CECs in the peripheral blood of anemic individuals was not associated with the suppression of T cells proliferation, we investigated the immunoregulatory properties of CECs from the healthy human bone marrow (Fig. 8a). CECs in the bone marrow are enriched with early-stage CECs (Supplementary Fig. 16a, b) and are predominantly CD45$^+$ (Supplementary Fig. 16c, d). Similar to murine CECs, their counterparts in the human bone marrow express ARG2 (Fig. 8b–d). Importantly, human erythroid cells also express ARG1 (Fig. 8b–d). The expression of both ARG isoforms was higher in CD45$^+$ than in CD45$^-$ CECs (Supplementary Fig. 16e, f). CECs isolated from human bone marrow (Supplementary Fig. 16g) significantly suppressed proliferation of both CD4$^+$ and CD8$^+$ T cells (Fig. 8e, f). These effects were diminished by the ARGi, confirming arginase-dependent effects of human CECs.

**Erythroleukemia-derived erythroid cell lines suppress T cells in an ARG- and ROS-dependent mechanism.** Further, we investigated the immunoregulatory properties of model human erythroleukemia-derived erythroid cell lines: K562, HEL92.1.7, and TF-I. These cells express multiple erythroid-lineage markers, including CD71 and CD235a (Fig. 9a), and spontaneously differentiate into erythroblast-like cells. We found that similarly to primary CECs, erythroid cells have substantial arginase activity (Supplementary Fig. 17a), express both ARG1 (Supplementary Fig. 17b) and ARG2 (Supplementary Fig. 17c), and have high ROS levels (Supplementary Fig. 17d). Notably, all examined types of erythroid cells potently suppressed proliferation of human CD4$^+$ (Fig. 9b–e) and CD8$^+$ (Fig. 9f–i) T cells in an ARG- and ROS-dependent manner (Fig. 9e, i and Supplementary Fig. 18a, b).

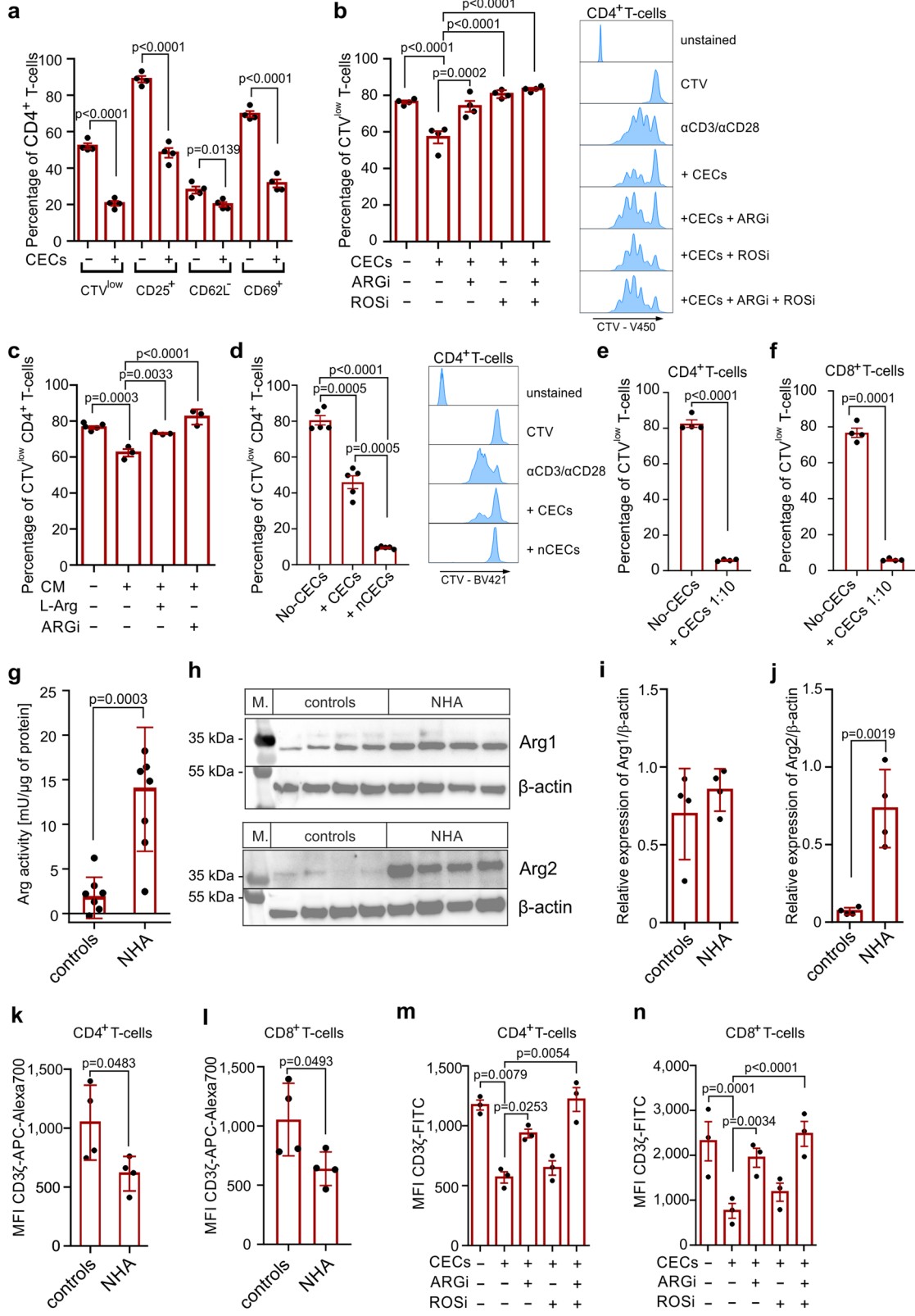

**Suppression of T cell function is a general feature of erythroid cells which disappears during their maturation.** Our results demonstrated that T cell suppression is a common feature of both murine and human CECs. Apparently the immunoregulatory properties were the most potent at the earliest stages of differentiation when the levels of ROS, ARG1, and ARG2 are the

highest. Therefore, we next sought to establish a model of ex vivo differentiation of erythroid cells. To this end, CECs were expanded and differentiated from PBMCs of healthy human donors (Supplementary Fig. 19a). PBMC-derived CECs expressed erythroid markers, including CD71, CD235a, CD36, and CD49d, and had high expression of CD44 and CD45 (Supplementary

**Fig. 5 CECs degrade L-Arg and produce ROS leading to the suppression of T cells. a** Proliferation and surface markers in αCD3/αCD28-stimulated CD4+ T cells co-cultured with CECs isolated from NHA mice ($n = 4$) at a ratio 1:2 (T cells:CECs). $P$ value was calculated with an unpaired $t$-test. **b** Effects of ARGi (OAT-1746, 500 nM) and ROSi ($N$-acetylcysteine, 100 μM) on the proliferation of αCD3/αCD28-stimulated CD4+ T cells co-cultured with CECs isolated from the spleens of NHA mice ($n = 4$) at a ratio 1:2 (T cells:CECs). Representative proliferation histograms of αCD3/αCD28-stimulated CD4+ T cells co-cultured with CECs in the presence of ARGi or ROSi. Histograms show the fluorescence of CTV (CellTraceViolet)-V450. $P$ value was calculated with one-way ANOVA with Bonferroni's post hoc test. **c** Effects of L-arginine supplementation (1000 μM) or ARGi (OAT-1746, 500 nM) on the proliferation of αCD3/αCD28-stimulated CD4+ T cells cultured in full medium or in CECs-conditioned medium (CM) ($n = 3$). $P$ value was calculated with one-way ANOVA with Bonferroni's post hoc test. **d** Proliferation of αCD3/αCD28-stimulated CD4+ T cells co-cultured with total CECs population or with nucleated CECs (nCECs) isolated using density-gradient centrifugation from NHA mice ($n = 5$) at a ratio 1:2 (T cells:CECs). Histograms show the fluorescence of CTV (CellTraceViolet)-V450. $P$ value was calculated with repeated measures ANOVA with Holm–Sidak's post hoc test. **e, f** Proliferation of αCD3/αCD28-stimulated CD4+ T cells (**e**) or CD8+ T cells (**f**) co-cultured with CECs isolated from NHA mice ($n = 4$) at a ratio 1:10 (T cells:CECs). $P$ value was calculated with paired $t$-test. **g** Arginase activity of the splenocytes lysate of control and anemic mice calculated per μg of total protein based on bicinchoninic acid (BCA) protein assay. $P$ value was calculated with an unpaired $t$-test. **h** The level of ARG1 and ARG2 in the splenocytes lysate of control ($n = 4$) and anemic mice ($n = 4$). β-Actin showed as a loading control. **i, j** Relative density of ARG1 (**i**) and ARG2 (**j**) compared to β-actin. $P$ value was calculated with an unpaired $t$-test. **k, l** The level of CD3ζ in CD4+ (**k**) and CD8+ (**l**) T cells in the spleen of control ($n = 4$) and anemic mice ($n = 4$) based on intracellular staining. $P$ value was calculated with an unpaired $t$-test. **m, n** The levels of CD3ζ in CD4+ (**m**) and CD8+ (**n**) αCD3/αCD28-stimulated T cells in the presence of CECs isolated from NHA mice ($n = 4$) based on intracellular staining. $P$ value was calculated with one-way ANOVA with Bonferroni's post hoc test. Data show means ± SEM (**a–f, m, n**) or means ± SD (**h, i–l**). Each point in **a–g, i–n** represents data from individual mice. $n$ values are the numbers of mice used to obtain the data or the number of biological replicates in in vitro experiments. The source data underlying **a–n** are provided as a Supplementary Data 2 file.

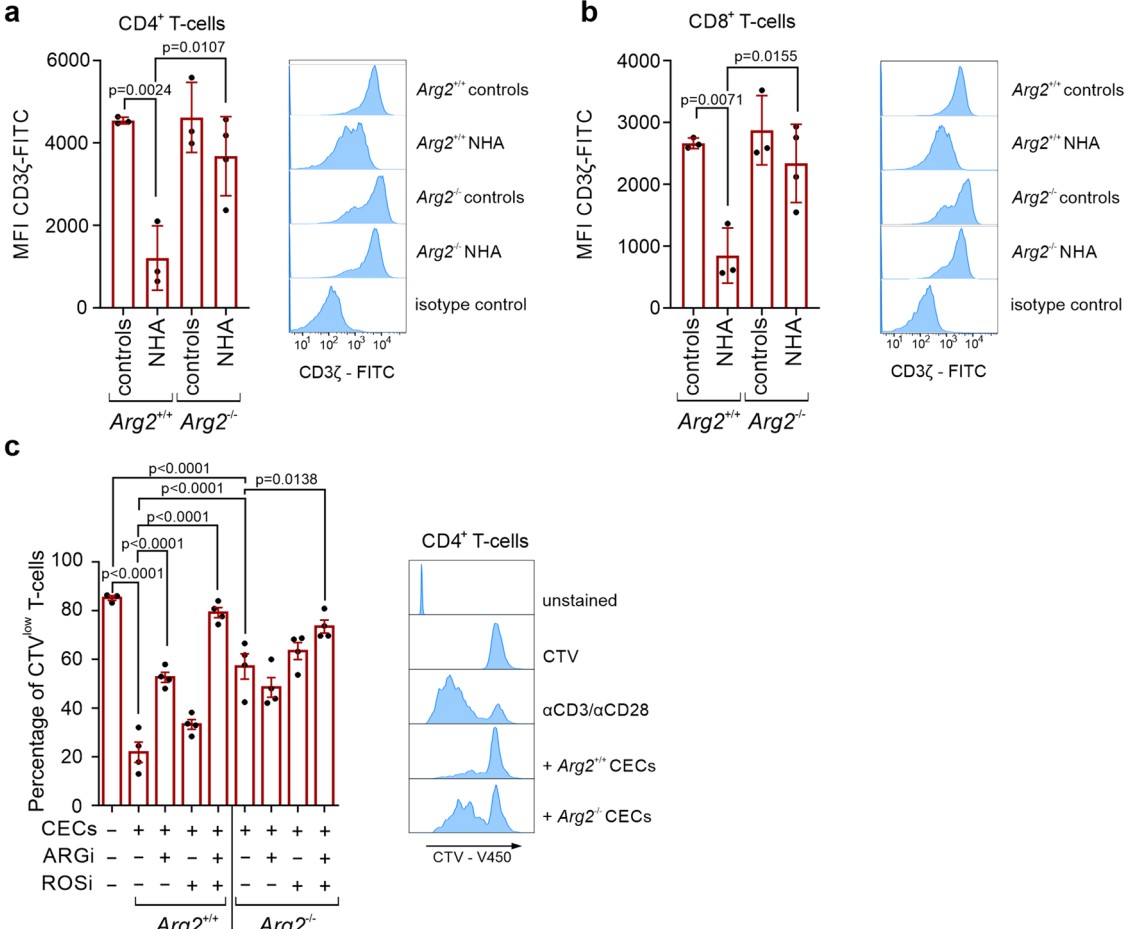

**Fig. 6 CECs from Arg2−/− mice have impaired immunoregulatory properties. a** The levels of CD3ζ in spleen CD4+ (**a**) and CD8+ (**b**) T cells in control ($n = 4$) and anemic mice ($n = 4$) based on intracellular staining. Histograms show the fluorescence of CD3ζ-FITC in CD4+ (**a**) and CD8+ (**b**) T cells. $P$ values were calculated with one-way ANOVA with Tukey's post hoc test. **c** Proliferation of αCD3/αCD28-stimulated CD4+ T cells co-cultured with CECs isolated from NHA Arg2−/− mice or NHA wild-type Arg2+/+ mice at a 1:4 ratio (T cells:CECs). Representative proliferation histograms of αCD3/αCD28-stimulated CD4+ T cells co-cultured with CECs isolated from Arg2−/− mice or wild-type Arg2+/+ mice in the presence of ARGi or ROSi. Histograms show the fluorescence of CTV (CellTraceViolet)-V450. $P$ values were calculated with one-way ANOVA with Bonferroni's post hoc test. Data show means ± SD (**a, b**) or means ± SEM (**c**). Each point in **a–c** represents data from individual mice. $n$ values are the numbers of mice used to obtain the data or the number of biological replicates in in vitro experiments. The source data underlying **a–c** are provided as a Supplementary Data 2 file.

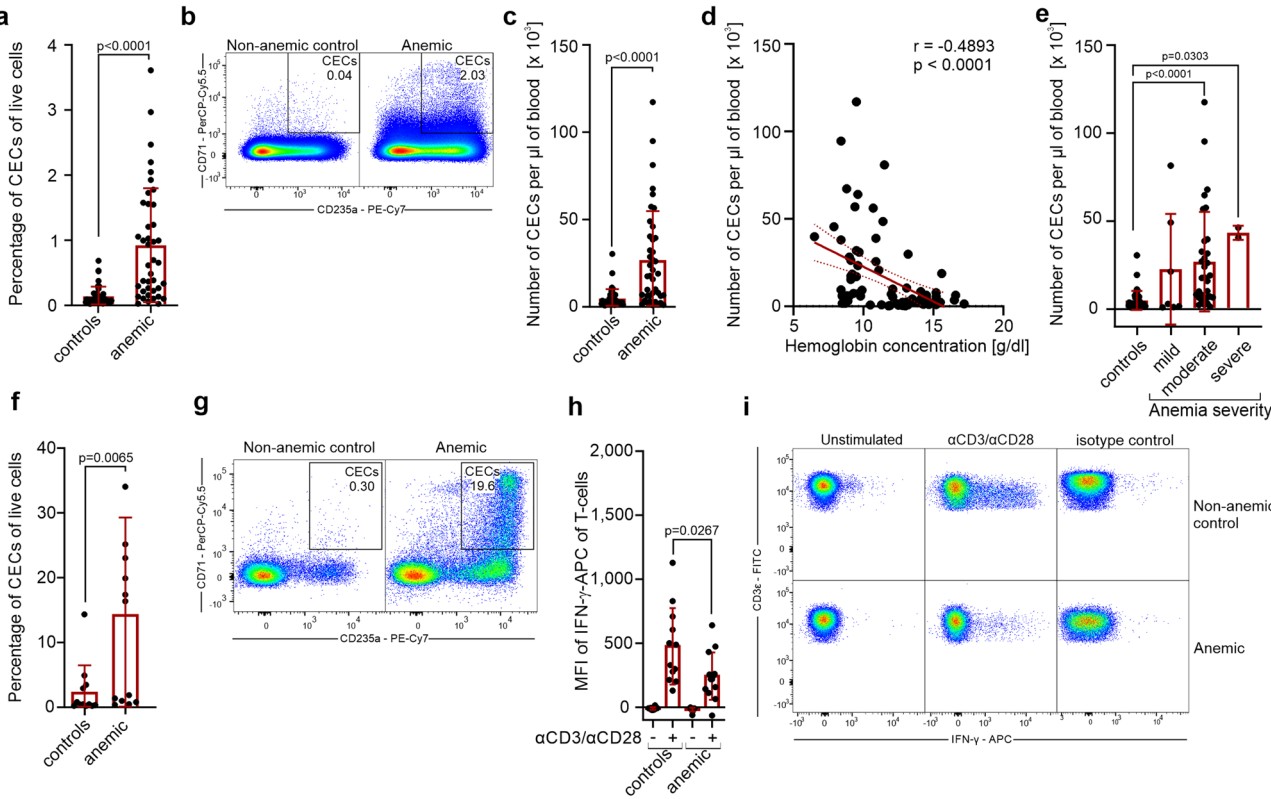

**Fig. 7 CECs expand in the blood of anemic patients and suppress T cells response. a** Percentages of live CD71+CD235a+ CECs in the whole blood of non-anemic (controls, $n = 42$) and anemic patients ($n = 41$). *P* value was calculated with the Mann–Whitney test. **b** Representative dot plots of CECs in the blood of non-anemic and anemic patients. **c** CECs count per µl of blood in controls ($n = 42$) and anemic patients ($n = 41$). *P* value was calculated with the Mann–Whitney test. **d** Correlation of the number of CECs per µl of blood and hemoglobin concentration ($n = 82$). The correlation was calculated with Spearman *r*. **e** CECs count per µl of blood in non-anemic controls ($n = 42$) and patients with mild ($n = 7$), moderate ($n = 32$), and severe ($n = 2$) anemia. *P* values were calculated with Kruskal–Wallis test with Dunn's post hoc test. **f, g** Percentages of CECs in the fraction of peripheral blood mononuclear cells (PBMC) in controls ($n = 12$) and anemic patients ($n = 13$) (**f**) and representative dot plots of CECs (**g**). *P* value was calculated with the Mann–Whitney test. **h** PBMCs of controls ($n = 12$) and anemic patients ($n = 13$) were stimulated with αCD3/αCD28 for 12 h in the presence of a protein transport inhibitor. IFN-γ levels were determined by intracellular staining. *P* values were calculated with an unpaired *t*-test. **i** Representative plots of IFN-γ levels in unstimulated or αCD3/αCD28-stimulated CD3ε+ T cells from PBMCs of anemic patients or non-anemic controls. Isotype control-stained cells are shown as a negative control. Data show means ± SD. Each point in **a**, **c**–**f**, **h** represents data from individual patients. *n* values are the numbers of patients used to obtain the data or the number of biological replicates in in vitro experiments. The source data underlying **a**, **c**–**f**, **h** are provided as a Supplementary Data 2 file.

Fig. 19a, b). Similar to their bone marrow counterparts, PBMCs-derived CECs had high levels of both ARG1 and ARG2 (Supplementary Fig. 20a, b). Moreover, isolated PBMC-derived CECs (Fig. 10a) potently suppressed both CD4+ and CD8+ human T cell proliferation (Fig. 10b, c).

Next, we aimed to study the possible changes in immunoregulatory properties of erythroid cells during differentiation into RBC. First, we investigated whether hematopoietic stem and progenitor cells (HSPCs) exert immunosuppressive effects. Mobilized hematopoietic stem cells obtained from peripheral blood (peripheral blood stem cells, PBSCs, Supplementary Fig. 21a) had high ARG1 and ARG2 levels (Supplementary Fig. 21b) and included only a small percentage of CECs (Supplementary Fig. 21c). Despite high arginase expression, PBSCs had no impact on T cell proliferation (Supplementary Fig. 21d, e).

Then, using continuous CECs culture, we demonstrated that CECs differentiated from PBMCs (Fig. 10d) exert robust, but transient suppressive properties, that disappear during erythroid differentiation (Fig. 10e–g). We found that of all CECs developmental stages, cells at the stage of CD71high CD235amid most strongly inhibited T cell proliferation (Fig. 10f, g). Moreover, these cells potently suppressed T cell activation based

on the CD25 and CD69 levels (Supplementary Fig. 22a, b) as well as inhibited IFN-γ production by T cells (Supplementary Fig. 22c). The suppression depended on both ARG and ROS since only the combination of ARGi and ROSi significantly diminished suppressed T cell activation, IFN-γ production (Supplementary Fig. 22c), and proliferation (Supplementary Fig. 23a, b). Loss of the suppressive properties corresponded with a decrease in CD71 (Fig. 10h), an increase in CD235a (Fig. 10i) as well as a decrease in CD49d (Fig. 10k) levels, the latter being a marker of the transition to the reticulocyte stage[27,28]. Subsequent CEC differentiation resulted in a complete loss of suppressive effects on T cells.

Further, we observed that induction of erythroid differentiation of K562 cells by sodium butyrate[29] (Supplementary Fig. 24a) was associated with a decrease of immunosuppressive effects on T cells (Supplementary Fig. 24b). These differentiated cells had decreased ARG2, but not ARG1 levels (Supplementary Fig. 24c), and decreased total arginase activity as compared with non-differentiated K562 cells (Supplementary Fig. 24d). Downregulation of ARG2 was most probably caused by mitophagy, a crucial process during erythroid differentiation[30], as evidenced by the decreased signal from a mitochondrial probe in differentiated K562-erythroid cells (Supplementary Fig. 24e). Similar changes in

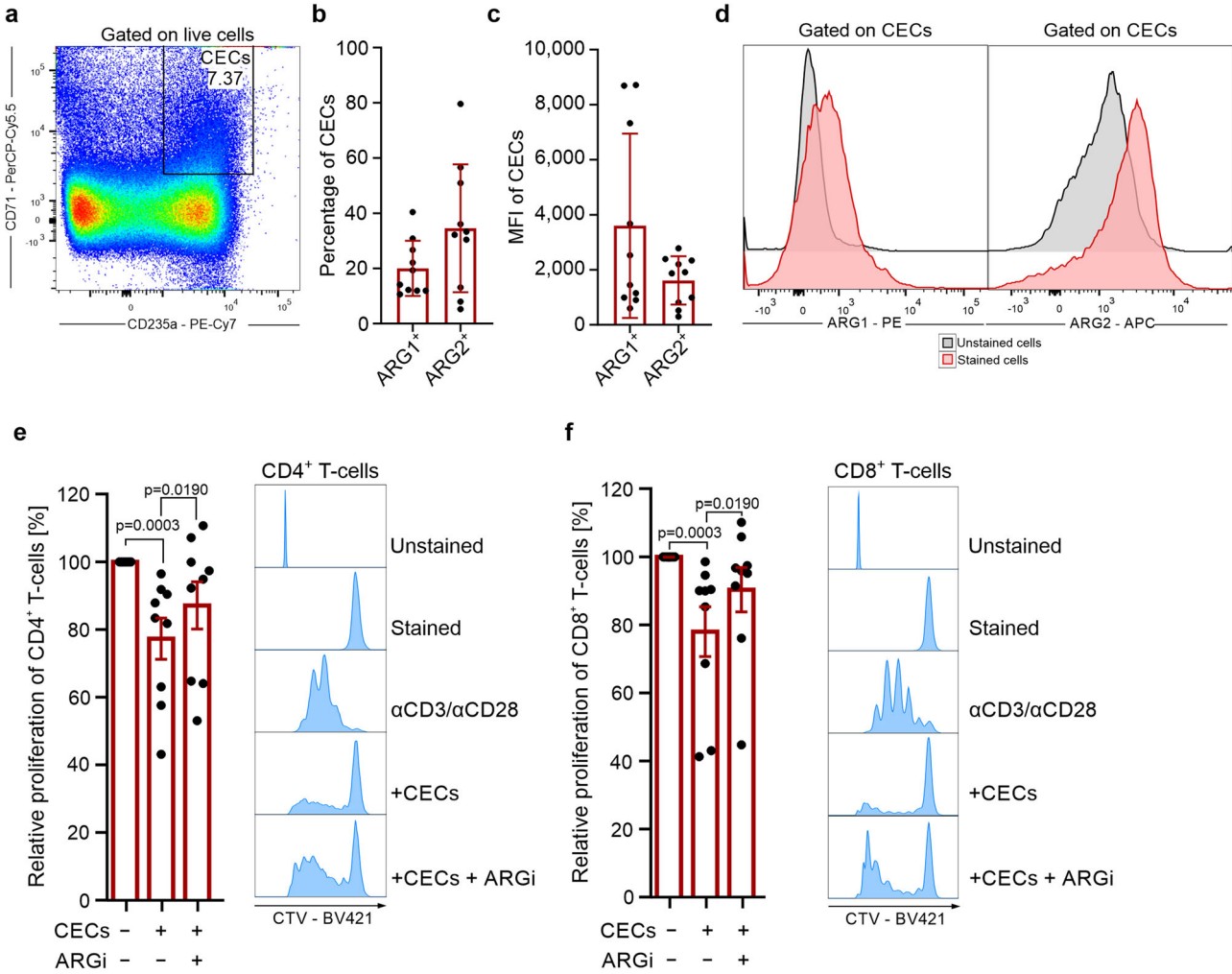

**Fig. 8 CECs from human bone marrow express ARG1 and ARG2 and suppress T cells proliferation. a** Representative plots of CD71$^+$ and CD235$^+$ CECs in the aspirate of human bone marrow of total live cells. **b** Percentages of ARG1$^+$ and ARG2$^+$ CECs in the human bone marrow based on intracellular staining. **c** Mean fluorescence intensity (MFI) of ARG1-PE and ARG2-APC in CECs. **d** Representative histograms of ARG1 and ARG2 levels in CECs from human bone marrow. Fluorescence-minus-one (FMO) is shown as unstained controls. **e, f** Proliferation triggered by αCD3/αCD28 of CTV-labeled CD4$^+$ (**e**) and CD8$^+$ (**f**) T cells co-cultured with CECs isolated from the human bone marrow. T cell to CEC ratio was 1:2 ($n = 9$). The proliferation of T cells in co-culture with CECs was calculated as a percentage of maximum proliferation (100%) of T cells that was triggered by αCD3/αCD28 antibodies and cultured without CECs. Representative histograms shows CTV-BV421 fluorescence of CD4$^+$ (**e**) or CD8$^+$ (**f**) T cells co-cultured with CECs at a 1:2 ratio in the presence of ARGi (OAT-1746, 500 nM). $P$ values were calculated with Friedman's test with Dunn's post hoc test. Data show means ± SD (**b**, **c**) or means ± SEM (**e**, **f**). $n$ values are the numbers of individual patients used to obtain the data or the number of biological replicates in in vitro experiments. The source data underlying **b**, **c**, **e**, **f** are provided as a Supplementary Data 2 file.

ARG expression were also detected in primary murine (Supplementary Fig. 6b–e) and human CECs (Supplementary Fig. 20b). Finally, we demonstrated that mature RBCs obtained from healthy donors had no impact on T cell proliferation (Supplementary Fig. 25a–c). Altogether, we show that human CECs possess robust, but transient suppressive properties that are most potent in the earliest developmental stages and disappear during erythroid cell maturation.

## Discussion

In this study, we demonstrate that suppression of T cells is a general feature of murine and human CECs. Anemic CECs via arginases and ROS suppress proliferation and production of IFN-γ by T cells. Using continuous human erythroid cell culture, we show that the immunoregulatory properties of CECs are transient and disappear during maturation.

Recent studies have broadened our understanding of the many roles played by CECs expanded by different triggers[3]. Immunoregulatory functions of CECs were reported for the first time in neonates that are characterized by a physiological abundance of CECs[4]. Neonatal CECs suppress anti-bacterial immunity via ARG2 by decreasing the production of proinflammatory cytokines by myeloid cells[4] and by suppressing antibody production in response to *B. pertussis*[6]. Moreover, they have been described to be involved in the expansion of inducible regulatory T cells[31]. We found that in adult mice anemia induced the expansion of early-stage CECs that had the highest expression of ARG2. Neither ARG2-expressing CECs nor recombinant ARG1 suppressed the production of TNF-α from myeloid cells. However, arginases seem to primarily impair T cells by decreasing their activation and proliferation[32]. Accordingly, we observed decreased proliferation of adoptively transferred OT-I cells in the spleen of anemic mice, which was reflected ex vivo in the co-culture of

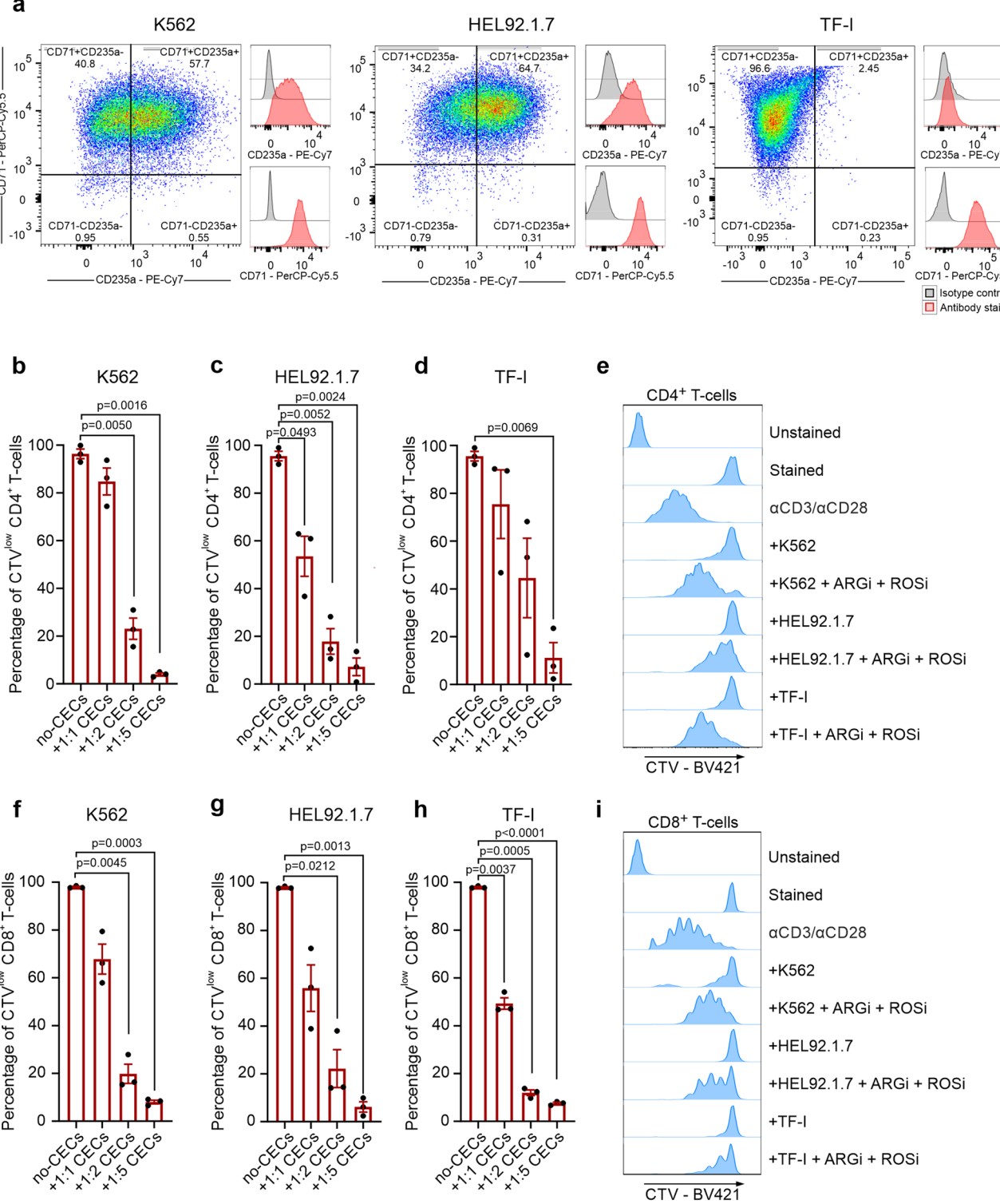

**Fig. 9 Erythroleukemia-derived erythroid cell lines suppress T cells in an ARG- and ROS-dependent mechanism. a** The levels of CD71 and CD235a in erythroleukemia-derived erythroid cell lines. **b–d** Proliferation of CTV-labeled CD4+ T cells triggered by αCD3/αCD28 co-cultured with K562 (**b**), HEL92.1.7 (**c**), and TF-I (**d**) erythroid cell lines at different ratios (T cells:CECs) (n = 3). P values were calculated with repeated measures ANOVA with Dunnett's post hoc test. **e** Representative histograms of CTV-BV421 fluorescence of CD4+ T cells co-cultured with erythroid cells at a 1:2 ratio in the presence of ARGi (OAT-1746, 1.5 μM) and ROSi (NAC, 200 μM) (n = 2). Statistical analyses are provided in Supplementary Fig. 18a. **f–h** Proliferation of CTV-labeled CD8+ T cells triggered by αCD3/αCD28 co-cultured with K562 (**f**), HEL92.1.7 (**g**), or TF-I (**h**) erythroid cells at different ratios (n = 3). P values were calculated with repeated measures ANOVA with Dunnett's post hoc test. **i** Representative histograms of CTV-BV421 fluorescence of CD8+ T cells co-cultured with erythroid cell lines at a 1:2 ratio in the presence of ARGi (OAT-1746, 1.5 μM) and ROSi (NAC, 200 μM) (n = 2). Statistical analyses are provided in Supplementary Fig. 18b. Data show means ± SEM. n values are the numbers of biological replicates in in vitro experiments. The source data underlying **b–d**, **f–h** are provided as a Supplementary Data 2 file.

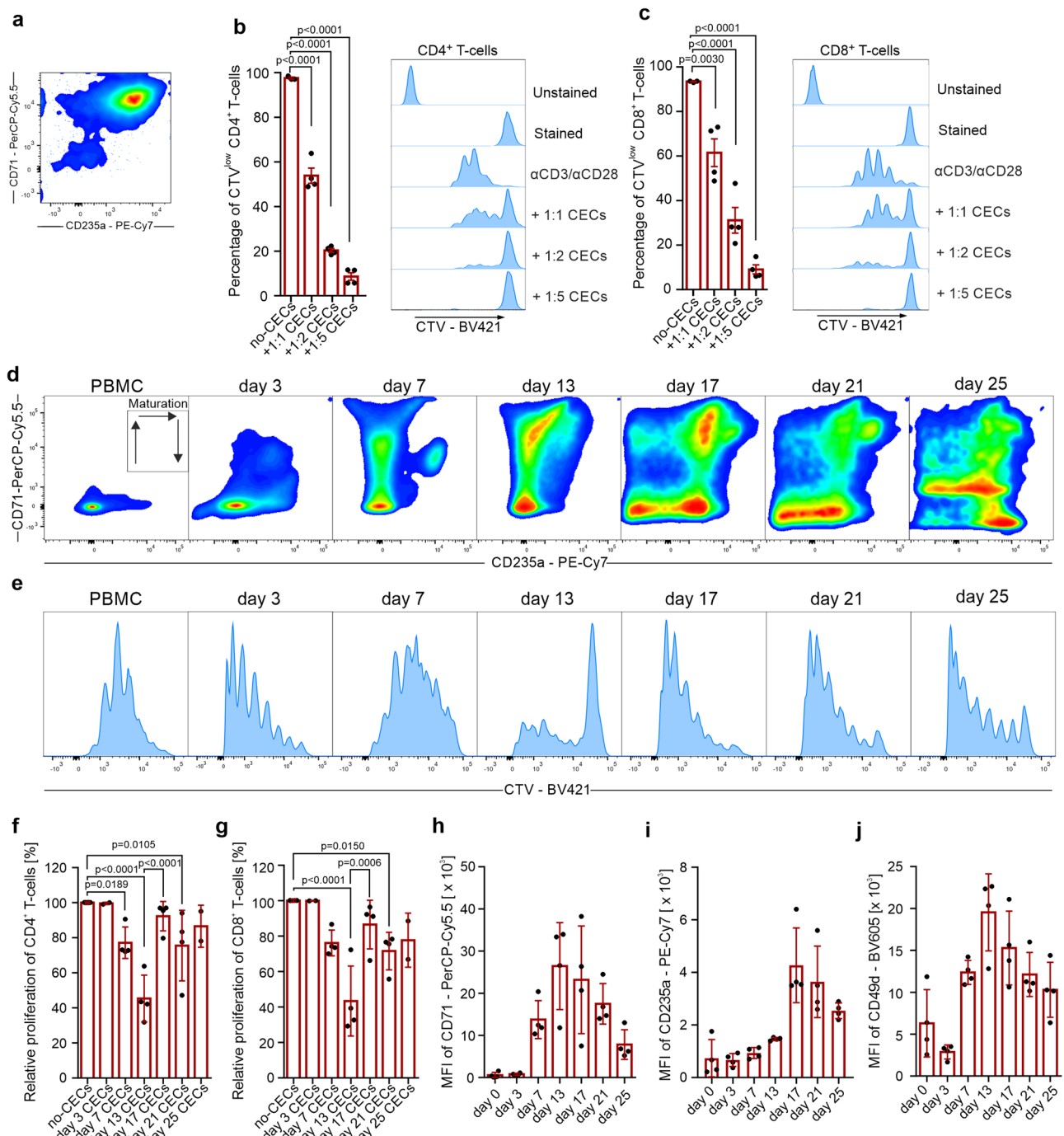

**Fig. 10 Suppression of T cells is a general feature of erythroid cells that diminishes with CECs maturation. a** Representative plot of isolated CECs differentiated from PBMCs. **b, c** Proliferation of CTV-labeled CD4+ (**b**) and CD8+ (**c**) T cells triggered by αCD3/αCD28 co-cultured with CECs differentiated from PBMCs ($n = 4$). P values were calculated with one-way ANOVA with Dunnett's post hoc test. **d** Representative density plots of CEC differentiation from PBMCs based on CD71 and CD235a expression. **e** Proliferation of CTV-labeled CD4+ triggered by αCD3/αCD28 co-cultured with CEC-PBMCs at different developmental stages at 1:4 ratio. **f, g** Relative proliferation of CD4+ (**f**) and CD8+ (**g**) T cells co-cultured with CEC-PBMCs at different time points at a 1:4 ratio. P values were calculated with one-way ANOVA with Bonferroni's post hoc test. **h–j** Levels of CD71 (**h**), CD235a (**i**), and CD49d (**j**) during erythroid differentiation from PBMCs. Data show means ± SD. Each point in **b, c, f–j** represents data from individual patients. n values are the numbers of individual patients used to obtain the data or the number of biological replicates in in vitro experiments. The source data underlying **b, c, f–j** are provided as a Supplementary Data 2 file.

murine T cells with CECs. Expansion of CECs in the spleen of anemic mice resulted in the increased ARG activity in the spleen leading to the L-arginine starvation of T cells, decreased levels of CD3ζ, and suppressed proliferation. Moreover, human CECs expressed both ARG1 and ARG2 and suppressed T cell proliferation in an ARG-dependent manner. Thus, expansion of ARG-expressing CECs in anemia may induce immune suppression, similar to the expansion of ARG-expressing myeloid cells in cancer[33] and in females over males[34], and during pregnancy[35].

CECs were also reported to modulate immune response via ROS in tumor-bearing mice and cancer patients[10]. We found that ROSi restored T cell proliferation in co-culture with CECs from anemic mice to a similar extend as ARGi. Importantly, ROS also may decrease CD3ζ in T cells[36]. However, ROSi restored CD3ζ decreased by CECs only in combination with ARGi, which confirms that ARG cooperates with ROS in CECs to induce T cells hyporesponsiveness to proliferative triggers.

Importantly, we demonstrated that previously described lack of immunosuppressive capacities of CECs in anemic mice[10] resulted from the interaction between PHZ used to induced anemia and ARGs. PHZ-induced HA is one of the most commonly used models of anemia. PHZ leads to the formation of ferrihemoglobin from oxyhemoglobin and production of free radicals that disrupt the interactions between hem and globin chains leading to the formation of Heinz bodies and hemolysis[21]. However, PHZ-induced CECs are less effective in suppressing T cell proliferation as compared with CECs isolated from neonatal or other anemic mice. We show that PHZ targets ARGs, critical immunomodulating enzymes. It needs to be considered in future studies that the interaction between PHZ and ARGs may have considerable effects on the obtained results.

Anemia correlates with worse outcomes in many diseases, including pneumonia[37] or cancer[38]. Moreover, preoperative anemia is associated with an increased risk of infection and mortality in patients undergoing surgery[39,40]. We demonstrated that CECs expand in anemic patients and may suppress the production of cytokines by T cells. A small fraction of CD45+ CECs in the peripheral blood of our cohort subject is consistent with a recent report on CECs in systemic juvenile idiopathic arthritis patients[41]. In line with our results, a recent study showed that anemia status influences the blood transcriptome with enrichment of erythrocyte differentiation genes as well as *ARG1* in anemic children, but decreased signatures of CD4+ T cell activation and differentiation[42]. It remains unknown to which extent CECs are responsible for immune suppression and whether in these conditions supplementation of iron, vitamin B$_{12}$, or administration of erythropoiesis-stimulating agents including EPO may restore immune response.

Erythropoiesis is a continuum of developmental states that gives mature red blood cells from a hematopoietic stem cell (HSC) and is strictly regulated by multiple factors[43]. Recent studies demonstrated that immunomodulatory properties are strong in early-stage CD45+ CECs in contrast to more mature CD45− CECs that lack these capacities[9,10,17,44]. However, other studies also reported the immunomodulatory role of CD45− CECs[41,45], which suggests that CD45 alone may not be a reliable marker of immunosuppressive CECs. Here, we showed that human CECs acquire immunomodulatory properties during erythroid differentiation and are the most potent in an early stage characterized by the CD71$^{high}$ CD235a$^{mid}$ phenotype. Further erythroid maturation is associated with the disappearance of the suppressive properties. These early-stage CECs are relatively rare in the peripheral blood of anemic individuals, which may be a reason for the lack of T cell proliferation suppression. In contrast, human bone marrow is enriched in early-stage CECs that suppress T cells proliferation. Nonetheless, the suppression of T cells by bone marrow CECs was substantially weaker than that of CD71$^{high}$ CD235a$^{mid}$ CECs from ex vivo culture. This may be explained by the fact that in human bone marrow CECs at the earliest stages of differentiation are still much less abundant than late-stage CECs[27,46].

The exact role of transient immunomodulatory properties of CECs remains elusive. It was suggested that expansion of CECs in neonates provides tolerance to harmless antigens, including the commensal microbiota[4], and minimalize damage caused by inflammation in the intestines[4], liver[7], and lungs[47] during the first days of postnatal life. In adults, the role of CECs seems to be similar. Recent studies demonstrated that stress erythropoiesis is a key inflammatory response[48]; therefore, expansion of CECs may prevent progression to chronic inflammation. Indeed, transfer of CECs suppressed inflammatory response and attenuated the wasting syndrome in murine models of colitis[11]. In cancer, which is characterized by a chronic inflammation[49], CECs substantially expand and suppress immune response facilitating tumor growth and increasing the susceptibility to pathogens[10]. On the other hand, impaired immunoregulatory properties of CECs may exacerbate the damage caused by inflammation[50]. Moreover, CECs by suppressing production of IFN-γ, a crucial inflammatory cytokine and a potent inhibitor of erythropoiesis[51,52], may allow maintenance of erythropoiesis and play a role in preventing systemic inflammation.

Importantly, most of the knowledge on CECs has been built based on murine models[5]. However, several crucial divergences between mice and humans may limit the translational character of CECs studies in mice, which include differences in stress erythropoiesis[53]. In mice, stress erythropoiesis primarily takes place in the spleen and relies on the expansion of early-stage CECs[54]. Thus, suppressive CECs may interact with immune cells in the spleen, which is also an active immune organ[55]. We observed that T cells in anemic mice are rather locally suppressed in the spleen, while not affected in the lymph nodes, similar to the conditions described for neonates[4]. In contrast, in humans stress erythropoiesis primarily involves the bone marrow, and expansion of CECs in extramedullary sites is rather occasional[5,53]. Moreover, CECs at the earliest stages of differentiation are relatively rare in healthy individuals[46]. Thus, it seems that CECs may have the most significant immunoregulatory role under conditions that are characterized by the impaired erythroid differentiation and robust enrichment of early-stage CECs, which include cancer[56].

Our findings might be of relevance in better understanding the mechanisms underlying suppressed cell-mediated immunity and anti-bactericidal capacity of leukocytes[57] and the impaired of T cell mediated immunity in anemic children[58]. Moreover, our study suggests that CECs may be a crucial regulator of immune response in different disease conditions.

## Methods

**Reagents**. Recombinant human ARG1 was obtained from Biolegend (San Diego, CA, USA), recombinant murine Arg1 was obtained from Cloud-Clone Corp., arginase inhibitor OAT-1746 was synthesized at OncoArendi Therapeutics, Warsaw, Poland. All other reagents, if not otherwise stated, were obtained from Sigma-Aldrich.

**Cell lines**. K562, HEL92.1.7, and TF-I cell lines were purchased from American Type Culture Collection (ATCC). Cells were cultured in RPMI-1640 medium supplemented with 10% heat-inactivated fetal bovine serum (FBS, HyClone), 2 mM L-glutamine (Sigma-Aldrich) 100 U/ml penicillin and 100 μg/ml streptomycin (Sigma-Aldrich) at 37 °C in a humidified atmosphere of 5% CO$_2$ in the air. Additionally, TF-I cells medium contained 2 ng/ml of recombinant human GM-CSF (R&D Systems). Cells have been cultured for no longer than 4 weeks after thawing and were regularly tested for *Mycoplasma* contamination using PCR technique and were confirmed to be negative.

**Human samples**. Human PBMCs used as a source of CD4+ and CD8+ T cells and used for the expansion and differentiation of PBMC-derived CECs were isolated from buffy coats obtained commercially from the Regional Blood Centre in Warsaw, Poland. PBMCs were isolated by density-gradient centrifugation method using Histopaque®-1077 (Sigma-Aldrich) or Lymphoprep™ (STEMCELL Technologies), according to the manufacturer's protocols. All donors were males between the ages of 18 and 45 years old. Donors were screened for general health and qualified by the physician for blood donation. All donors had negative clinical laboratory tests for HIV-1, HIV2, hepatitis B, and hepatitis C and hematology values within normal ranges.

Peripheral blood samples were obtained from patients hospitalized in the Central Teaching Clinical Hospital, Medical University of Warsaw or treated in the Outpatient Clinic of Central Teaching Clinical Hospital, Medical University of Warsaw, Warsaw. The study was conducted in accordance with the Declaration of Helsinki. The study was approved by the Bioethical Committee of the Medical University of Warsaw (KB/8/2021). Patients with or without anemia based on WHO diagnostic criteria[59] were enrolled in the study. Patients with proliferative diseases, including cancer, were excluded. The blood samples were obtained by venipuncture and subjected to complete blood count evaluation. The remaining blood was used for further examination. Flow cytometry was performed as described below. CountBright™ Absolute Counting Beads (Thermo Fisher Scientific) were used for CECs counting. PBMCs were purified from whole blood of anemic and non-anemic patients by density separation using Lymphoprep (STEMCELL Technologies).

Human bone marrow samples were commercially obtained from Lonza Walkersville, Inc. Aspirates were withdrawn from bilateral punctures of the posterior iliac crests. Every 100 ml of bone marrow was collected into syringes containing 10 ml of heparin (Porcine Intestinal Mucosa) Sodium Injection (~100 units heparin per ml bone marrow). Bone marrow samples were obtained from both healthy males ($n = 6$) and healthy non-pregnant females ($n = 3$) US-based donors between the ages of 23 and 45 years old. Samples were collected after obtaining permission for their use in research applications by informed consent. All donors were screened for general health and negative medical history for heart disease, kidney disease, liver disease, cancer, epilepsy, and blood or bleeding disorders. All donors had negative clinical laboratory tests for HIV-1, HIV2, hepatitis B, and hepatitis C.

Mobilized PBSCs were obtained from familial donors from the material remaining after allogeneic stem cell transplantation. PBSCs were mobilized with the granulocyte colony-stimulating factor (G-CSF) and isolated from the donor peripheral blood using apheresis according to the standard clinical protocol. Informed consent was obtained from PBSC donors. Collected PBSCs were subjected to density-gradient centrifugation using Lymphoprep (STEMCELL Technologies) to remove dead cells and debris, washed three times with RPMI medium, and used for the analysis.

**Mice.** Wild-type C57BL/6 mice, both males and females, 8- to 14-week-old were obtained from the Animal House of the Polish Academy of Sciences, Medical Research Institute (Warsaw, Poland). Transgenic mice of C57BL/6 genetic background, B6.129S4-Arg1$^{tm1Lky}$/J (YARG mice co-expressing Arg1 and eYFP[60], stock #015857), C57BL/6-Tg(TcraTcrb) 1100Mjb/J (OT-I, TCR transgenic mice producing OVA peptide-specific CD8$^+$ T cells[61], stock #003831) and Arg2$^{tm1Weo}$/J (Arg2$^{-/-}$, Arg2 functional knockout[26]), were purchased from the Jackson Laboratories and bred at the animal facility of the Department of Immunology, Medical University of Warsaw. Mice were housed in controlled environmental conditions in specific-pathogen-free (SPF) conditions (breeding cages, OT-I mice) or conventional (others) animal facility of the Department of Immunology, Medical University of Warsaw, with water and food provided ad libitum. For mice genotyping, DNA was isolated with DNeasy Blood & Tissue Kit (Qiagen) according to the manufacturer's instructions. The concentration and purity of DNA were determined using the NanoDrop 2000c spectrophotometer (Thermo Fisher Scientific). PCR reaction was performed using OneTaq® 2× Master Mix with Standard Buffer (New England Biolabs). Primers sequences, PCR, and agarose electrophoresis conditions were set according to genotyping protocols available on The Jackson Laboratory website (https://www.jax.org). The experiments were performed in accordance with the guidelines approved by the II Local Ethics Committee in Warsaw (approval No. WAW2/117/2019 and WAW2/143/2020) and in accordance with the requirements of the EU (Directive 2010/63/EU) and Polish (Dz. U. poz. 266/15.01.2015) legislation.

**Animal anemia models.** To induce NHA mice were phlebotomized 4 and 2 days before harvest. At least 100 µl of blood was collected each time. To induce HA, mice were injected intraperitoneally (i.p.) three days before harvest with 50 mg per kg body weight of PHZ hydrochloride solution (HA-PHZ) or mice were injected intravenously (i.v.) 6 days before harvest with 45 µg of anti-TER119 monoclonal antibody (TER-119, BioXCell) into the caudal vein. For the analysis of stress erythropoiesis, mice were injected i.v. with 30 µg of anti-TER119 monoclonal antibody (TER-119, BioXCell) into caudal vein followed by the monitoring of RBC count in the peripheral blood. For complete blood count, blood was collected into EDTA-coated tubes from inferior palpebral veins and examined using a Sysmex XN-2000 Hematology Analyzer. The parameters of complete blood counts and reference intervals[62] are presented in Supplementary Fig. 1. For the analysis of amino acid concentration, blood was collected into IMPROMINI® Gel & Clot Activator Tubes. Tubes were gently inverted five times to mix the clot activator and incubated for 30 min at room temperature (RT) followed by centrifugation for 10 min at $1000 \times g$ at 4 °C. Serum was collected and stored at −20 °C until analysis. L-Arg concentration in the serum was determined with ultra-performance liquid chromatography-tandem mass spectrometry (UPLC-MS/MS) method on Waters Xevo TQ-S mass spectrometer equipped with Waters Acquity UPLC chromatograph (Waters) in the Mass Spectrometry Lab at the Institute of Biochemistry and Biophysics, Polish Academy of Sciences, Warsaw, Poland.

**Antibodies.** Fluorophore- or biotin-conjugated antibodies specific for mouse cell-surface antigens and cytokines were as follows: anti-CD71 (8D3, NovusBio; R17217, eBioscience, dilution 1:100), anti-TER119 (TER-119, BioLegend, 1:100), anti-CD45.2 (104, BD Biosciences, 1:50), anti-CD45 (30-F11, BioLegend, 1:100), anti-CD44 (IM7, BioLegend, 1:100), anti-CD3e (145-2C11, eBioscience, 1:100), anti-CD4 (GK1.5, eBioscience; RM4-5, eBioscience; 1:100), anti-CD8a (53-6.7, eBioscience, 1:100), anti-CD69 (H1.2F4, eBioscience, 1:100), anti-CD25 (PC61.5, eBioscience, 1:100), anti-CD62L (MEL-14, Invitrogen, 1:100), anti-CD11b (M1/70, BioLegend, 1:100), anti-CD11c (HL3, BD Bioscience, 1:100), anti-CD3 zeta (H146-968, Abcam, 1:500), anti-IFN-γ (XMG1.2, eBioscience, 2.5 µl per test in 50 µl volume), anti-TNF-α (MP6-XT22, eBioscience, 2.5 µl per test in 50 µl volume), anti-Arg1 (polyclonal, IC5868P/F, R&D Systems, 5 µl per test in 100 µl volume), anti-Arg2 (ab81505, Abcam, 1 µl per test in 50 µl volume), goat anti-rabbit IgG (Invitrogen, 1 µl per test in 100 µl volume).

Fluorophore- or biotin-conjugated antibodies specific for human cell-surface antigens and cytokines were as follows: anti-CD71 (CY1G4, BioLegend, DF1513, NovusBio, 1:50), anti-CD235a (HI264, BioLegend, 1:50), anti-CD44 (IM7, BioLegend, 1:100), anti-CD25 (BC96, eBioscience, 1:50), anti-CD69 (FN50, eBioscience, 1:50), anti-CD45 (HI30, BD Bioscience, 1:50), anti-CD49d (9F10, eBioscience, 1:50), anti-CD36 (NL07, eBioscience, 1:50), anti-CD34 (561, 25 BioLegend, 1:50), anti-CD3 (OKT3, eBioscience, 1:50), anti-CD4 (RPA-T4, eBioscience, 1:50), anti-CD8a (RPA-T8, eBioscience, 1:50), anti-IFN-γ (4S.B3, BioLegend, 2.5 µl per test in 50 µl volume), anti-TNF-α (MAb11, BD Bioscience, 2.5 µl per test in 50 µl volume), anti-Arg1 (polyclonal, IC5868P/F, R&D Systems, 5 µl per test in 100 µl volume), anti-Arg2 (ab137069, Abcam, 1 µl per test in 50 µl volume), goat anti-rabbit IgG (Invitrogen, 1 µl per test in 100 µl volume).

**Flow cytometry analysis.** Flow cytometry was performed on FACSCanto II (BD Biosciences) or Fortessa X20 (BD Biosciences) operated by FACSDiva software. For data analysis FlowJo v10.6.1 software (Tree Star) or BD FACSDiva software (BD Biosciences) were used. Murine whole blood was collected from the inferior palpebral vein to EDTA-coated tubes. Spleens were isolated from mice and mechanically dispersed by pressing gently through a 70-µm nylon cell strainer using a rubberized 1 ml syringe piston. Murine bone marrow was isolated from the femur by the centrifugation method. Briefly, femurs were dissected, followed by the removal of any muscle or connective tissue. The condyles and epiphysis were removed and a cleared bone was placed in microcentrifuge tubes followed by centrifugation at $2500 \times g$ for 30 s. Bone marrow cells were filtered through a 70-µm nylon strainer and used for further analysis. No erythrocyte lysis was performed in flow cytometry analysis or experiments that involved analysis or isolation of CECs or RBCs. If only other types of cells were analyzed using flow cytometry, erythrocytes were lysed using ACK (ammonium-chloride-potassium) Lysing Buffer (Thermo Fisher Scientific), according to the manufacturer's protocol.

For cell surface staining, cells were stained with Zombie NIR™, Zombie UV™ or Zombie Aqua™ Fixable Viability Kit (BioLegend), blocked on ice with 5% normal rat serum in FACS buffer (PBS; 1% BSA, 0.01% sodium azide), and then incubated for 30 min on ice with fluorochrome-labeled antibodies. Fluorochrome-conjugated antibodies used for the staining are listed above. After washing in FACS buffer, cells were immediately analyzed.

For nucleus staining, cells were incubated before cell surface staining with Hoechst 33342 Fluorescent Stain (Invitrogen) at a final concentration of 1 µg/ml in Dulbecco's phosphate-buffered saline (DPBS) for 30 min, followed by a wash in DPBS. For mitochondrial staining, cells were incubated before cell surface staining with MitoSpy™ Red CMXRos (BioLegend) at a final concentration of 50 nM in RPMI medium at 37 °C for 30 min, followed by a wash in FACS buffer.

For intracellular staining, membrane-stained cells were fixed using Fixation Buffer for 30 min at RT, followed by a wash with permeabilization buffer, and staining with an antibody diluted in permeabilization buffer for 30 min at RT (Intracellular Fixation & Permeabilization Buffer Set, eBioscience). For anti-ARG2 indirect intracellular staining, cells were fixed using Fixation Buffer for 30 min at RT, followed by a wash with permeabilization buffer, and staining with anti-ARG2 antibody for 1 h at RT, followed by a wash with permeabilization buffer and staining with fluorochrome-conjugated goat anti-rabbit IgG for 30 min at RT. Gating strategies used to analyze the flow cytometry data are presented in Supplementary Figs. 26–44.

**IFN-γ and TNF-α production assay of murine cells.** Murine splenocytes were isolated from anemic or healthy mice. Cells were plated in round-bottomed 96-well plates ($1 \times 10^6$ cell per well) in L-arginine-free RPMI medium (SILAC RPMI medium, Thermo Fisher Scientific) supplemented with 10% dialyzed FBS (Thermo Fisher Scientific), 2 mM glutamine, 100 U/ml penicillin, 100 µg/ml streptomycin, 40 mg/l L-lysine, and 150 µM L-arginine (all from Sigma-Aldrich). Splenocytes were stimulated with heat-killed *E. coli 0111:B4* (HKEc, InvivoGen) at the concentration $1 \times 10^6$ cells per ml or Dynabeads Mouse T-Activator CD3/CD28 (ratio 1:2, Thermo Fisher Scientific) in the presence of protein transport inhibitor (BD GolgiStop™) for 6 h. Then, cells were stained with cell surface antigen-binding antibodies, followed by fixation, permeabilization, and intracellular staining for IFN-γ and TNF-α. Flow cytometry was performed on Fortessa X20 (BD Biosciences). Cell viability after culture in the presence of protein transport inhibitor was >80%.

**IFN-γ and TNF-α production assay of human cells.** Human PBMCs were isolated from the peripheral blood of anemic or non-anemic patients. Cells were plated in round-bottomed 96-well plates ($1 \times 10^6$ cell per well) in L-arginine-free RPMI medium (SILAC RPMI medium, Thermo Fisher Scientific) supplemented with 10% dialyzed FBS (Thermo Fisher Scientific), 2 mM glutamine, 100 U/ml penicillin, 100 µg/ml streptomycin, 40 mg/l L-lysine, and 150 µM L-arginine (all from Sigma-Aldrich). PBMCs were stimulated with heat-killed *E. coli* 0111:B4 (HKEc, InvivoGen) at the concentration $1 \times 10^6$ cells per ml for TNF-α assessment in myeloid cells or Dynabeads Human T-Activator CD3/CD28 (ratio 1:2, Thermo Fisher Scientific) for IFN-γ assessment in T cells in the presence of protein transport inhibitor (BD GolgiStop™) for 12 h. Then, cells were stained with cell surface antigen-binding antibodies, followed by fixation, permeabilization, and intracellular staining for IFN-γ and TNF-α. Flow cytometry was performed on Fortessa X20 (BD Biosciences). Cell viability after culture in the presence of protein transport inhibitor was >93%.

For the analysis of the effect of PBMC-derived CECs on T cell activation and IFN-γ production, PBMCs were isolated from the peripheral blood healthy blood donor. CD4+ or CD8+ T Cells were isolated from PBMC using EasySep™ Human CD4+ or CD8+ T-Cell Isolation Kit (STEMCELL Technologies) according to the manufacturer's protocols. CD4+ or CD8+ T cells were plated in round-bottomed 96-well plates ($2 \times 10^4$ cell per well) in L-arginine-free RPMI medium (SILAC RPMI medium, Thermo Fisher Scientific) supplemented with 10% dialyzed FBS (Thermo Fisher Scientific), 2 mM glutamine, 100 U/ml penicillin, 100 µg/ml streptomycin, 40 mg/l L-lysine, and 150 µM L-arginine (all from Sigma-Aldrich). T cells were stimulated with Dynabeads Human T-Activator CD3/CD28 (ratio 1:2, Thermo Fisher Scientific) in the presence of PBMC-derived CECs (ratio 1:2, $4 \times 10^5$ CECs per well). The arginase inhibitor OAT-1746 (1500 nM) or *N*-acetylcysteine (200 µM) was added as indicated in the figures. In these concentrations, ARGi and ROSi had no effects on T cells nor CECs viability. Human T cells were incubated for 72 h at 37 °C in 5% $CO_2$. Protein transport inhibitor (BD GolgiStop™) was added for the last 12 h. Then, cells were stained with cell surface antigen-binding antibodies, followed by fixation, permeabilization, and intracellular staining for IFN-γ and TNF-α. Flow cytometry was performed on Fortessa X20 (BD Biosciences). Cell viability after culture in the presence of protein transport inhibitor was >86%.

**In vivo OVA immunization and analysis of the humoral response.** Control and NHA mice were immunized with albumin from chicken egg white (OVA, Ovalbumin) from Sigma (Grade VII). Each mouse received 25 µg of OVA with Imject™ Alum Adjuvant (ALUM, Thermo Fisher Scientific) at a ratio of 1:1 in the final volume of 100 µl per mouse administered i.p. After 14 days, mice were challenged once again with the same dose of OVA-ALUM. NHA mice were divided into three groups. NHA before mice were phlebotomized before first immunization, NHA boost mice were phlebotomized before second OVA immunization, and NHA both were phlebotomized before first and second immunization (see Supplementary Fig. 3c). Untreated mice received Imject™ Alum Adjuvant without OVA. Blood was obtained from mice 14 days after the second immunization; plasma was isolated and stored at −80 °C. The concentration of anti-OVA IgG antibodies was determined using Anti-Ovalbumin IgG1 (mouse) ELISA Kit (Cayman Chemical).

**In vivo proliferation assay.** OVA (SIINFEKL)-specific CD8+ T cells were isolated from the spleen and lymph nodes of healthy 6–8-week-old OT-I mice using EasySep™ Mouse CD8+ T-Cell Isolation Kit (STEMCELL Technologies) according to the manufacturer's protocols. Isolated OT-I cells labeled with CTV for 20 min at 37 °C at a final concentration of 2.5 µM in PBS, washed, and transferred into the caudal tail vein of host C57BL/6 mice at a cell number of $7 \times 10^6$ in 150 µl of PBS. Twenty-four hours post OT-I T cells inoculation, host mice were challenged with 7.5 µg of full-length OVA protein (grade V, Sigma-Aldrich) injected into the caudal tail vein. Three mice from controls were injected only with PBS (negative control). On day 3 post OVA immunization, spleens were harvested, mashed through a 70-µm nylon strainer, stained with OVA-specific MHC tetramers (iTAg Tetramer/PE-H-2 Kᵇ OVA (SIINFEKL), MBL Inc., WA, USA) to detect OT-I CD8+ T cells, followed by anti-CD3 and anti-CD8 staining, and analyzed for proliferation by flow cytometry. The gate for proliferating cells (CTVlow) was set using unstimulated negative control. OT-I T cells with lower fluorescence of CTV than non-proliferating T cells were identified as proliferating cells.

**Murine T cell proliferation assay.** Murine T cells were isolated from spleens of healthy 6–8-week-old C57BL/6 mice using EasySep™ Mouse CD4+ or CD8+ T-Cell Isolation Kit (STEMCELL Technologies) according to the manufacturer's protocols. CECs were isolated from the spleens of anemic mice using the EasySep™ Release Mouse Biotin Positive Selection Kit (STEMCELL Technologies) according to the manufacturer's protocols. Biotin-conjugated anti-CD71 antibodies (anti-mouse clone 8D3, NovusBio,) were used at a final concentration of 1 µg/ml. CECs purity was >80%. For cell proliferation assay, T cells were labeled with Cell Trace Violet (CTV) dye (Thermo Fisher Scientific) for 20 min at 37 °C at a final concentration of 2.5 µM in PBS. Next, the labeled T cells were plated in round-bottomed 96-well plates ($5 \times 10^4$ cell per well) in L-arginine-free RPMI medium (SILAC RPMI medium, Thermo Fisher Scientific) supplemented with 10% (v/v)

dialyzed FBS (Thermo Fisher Scientific), 2 mM glutamine, 100 U/ml penicillin, 100 µg/ml streptomycin, 1% (v/v) MEM non-essential amino acids solution (Thermo Fisher Scientific), 50 µM 2-mercaptoethanol (Thermo Fisher Scientific), and 150 µM L-arginine and 40 mg/l L-lysine (Sigma-Aldrich). Proliferation was triggered by the stimulation with Dynabeads Mouse T-Activator CD3/CD28 (ratio 1:2, Thermo Fisher Scientific). The arginase inhibitor OAT-1746 (ARGi, 500 nM), L-arginine (1000 µM), or *N*-acetylcysteine (ROSi, 100 µM) were added as indicated in the figures. In these concentrations, ARGi and ROSi had no effects on T cells nor CEC viability. Murine CECs were added to the wells in a 1:2 ratio ($1 \times 10^5$ CECs per well). Murine T cells were incubated for 72 h at 37 °C in 5% $CO_2$. Then, cells were harvested, stained with live/dead Zombie dye (Biolegend), anti-CD3 and anti-CD4 or anti-CD8 antibody, and analyzed by flow cytometry. The gate for proliferating cells was set based on the unstimulated controls. Cell autofluorescence was determined using CTV-unstained controls. Percentages of proliferating cells were calculated using the FlowJo Software v10.6.1 (Tree Star).

**Human T cell proliferation assay.** Human T cells were isolated from PBMC isolated from buffy coats commercially obtained from the Regional Blood Centre in Warsaw, Poland using EasySep™ Human CD4+ or CD8+ T-Cell Isolation Kit (STEMCELL Technologies) according to the manufacturer's protocols. CECs were isolated from human bone marrow aspirates or PBMC-derived CECs culture using EasySep™ Release Mouse Biotin Positive Selection Kit (STEMCELL Technologies) according to the manufacturer's protocols. Biotin-conjugated anti-CD71 antibodies (anti-human clone 1B513, NovusBio) were used at a final concentration of 1 µg/ml. CECs purity was >80%. For cell proliferation assay, T cells were labeled with Cell Trace Violet (CTV) dye (Thermo Fisher Scientific) for 20 min at 37 °C at a final concentration of 2.5 µM in PBS. Next, the labeled T cells were plated in round-bottomed 96-well plates ($2 \times 10^4$ cell per well) in L-arginine-free RPMI medium (SILAC RPMI medium, Thermo Fisher Scientific) supplemented with 10% (v/v) dialyzed FBS (Thermo Fisher Scientific), 2 mM glutamine, 100 U/ml penicillin, 100 µg/ml streptomycin, 1% (v/v) MEM non-essential amino acids solution (Thermo Fisher Scientific), 50 µM 2-mercaptoethanol (Thermo Fisher Scientific), and 150 µM L-arginine and 40 mg/l L-lysine (Sigma-Aldrich). Proliferation was triggered by the stimulation with Dynabeads Human T-Activator CD3/CD28 (ratio 1:2, Thermo Fisher Scientific). The arginase inhibitor OAT-1746 (1500 nM) or *N*-acetylcysteine (200 µM) was added as indicated in the figures. In these concentrations, ARGi and ROSi had no effects on T cells nor CECs viability. Human CECs were added to the wells in a 1:2 ratio ($4 \times 10^5$ CECs per well). T cells were incubated for 120 h at 37 °C in 5% $CO_2$. Then, cells were harvested, stained with live/dead Zombie dye (Biolegend), anti-CD3 and anti-CD4 or anti-CD8 antibody, and analyzed by flow cytometry. The gate for proliferating cells was set based on the unstimulated controls. Cell autofluorescence was determined using CTV-unstained controls. Percentages of proliferating cells were calculated using the FlowJo Software v10.6.1 (Tree Star).

**CEC-conditioned medium (CM).** Conditioned medium was obtained by culturing CECs in the arginine-free RPMI medium (SILAC RPMI medium, Thermo Fisher Scientific) supplemented with 150 µM L-arginine at the density $1 \times 10^6$ cells/ml for 24 h. Cells were centrifuged and the supernatant was collected and immediately frozen at −80 °C. After thawing, the supernatant was filtered through a 0.45 µm Syringe Filter (Wenk LabTec) and was used in experiments a 1:1 ratio with 150 µM L-arginine RPMI SILAC medium.

**ROS detection.** The level of ROS in cells was determined using CellROX Green Reagent (Thermo Fisher Scientific) or 2′,7′-dichlorodihydrofluorescein diacetate (DCFDA). Cells were stained with CellROX at a final concentration of 5 µM or DCFDA at a final concentration of 10 µM in pre-warmed PBS for 30 min at 37 °C, followed by three washes with PBS. $H_2O_2$-treated cells served as positive controls. For some experiments, cells stained with CellROX or DCFDA were further stained with fluorochrome-labeled antibodies on ice. Stained cells were acquired on Fortessa X20 flow cytometer (BD Biosciences).

**Arginase activity assay and Griess test.** Recombinant enzymes (ARG1 and ARG2) to study ARGi were produced at OncoArendi Therapeutics in *E. coli* expression system and purified by the FPLC method. The proteins were purified by FPLC and stored at −80 °C in the storage buffer containing: 20 mM Tris pH 8.0, 100 mM NaCl, 10 mM DDT, and 10% glycerol. The basic assay buffer was composed of 100 mM sodium phosphate buffer, 130 mM sodium chloride, 1 mg/ml BSA, pH 7.4. The enzymatic reaction was carried out in the presence of 200 µM $MnCl_2$ (cofactor) and 10 or 20 mM L-arginine hydrochloride (for hARG1 or hARG2, respectively), mixed at the final volume of 25 µL. Basic developing buffer contained 50 mM boric acid, 1 M sulfuric acid, 0.03% (m/v) Brij® 35 detergent. PHZ or ABH was diluted in basic assay buffer at the volume of 50 µL. The recombinant enzyme was diluted in a basic assay buffer at the volume of 25 µL. The reaction was performed at the final volume of 100 µL. Developing mixture included freshly prepared equal volume mixture of developing solution A (4 mM *o*-phthaldialdehyde) and solution B (4 mM *N*-(1-naphthyl)ethylenediamine dihydrochloride) prepared in the basic developing buffer. The compound background wells contained each of the tested compounds and the substrate/cofactor mixture,

but not the recombinant enzyme (data were excluded from the analysis when the compound background exceeded 10% of the signal obtained in the wells with enzyme). The "0% activity" background wells contained only the substrate/cofactor mixture. Following 1 h incubation at 37 °C, the freshly prepared developing reagent was added (150 μl) and the colorimetric reaction was developed (12 min at RT, gentle shaking). The absorbance, proportional to the amount of the produced urea, was measured at 515 nm using Tecan's Spark™ microplate reader. Data were normalized by referring the absorbance values to the positive control wells (100% enzyme activity). The $IC_{50}$ value was determined by the nonlinear regression method. Arginase activity in the CECs or splenocytes lysates and cell supernatant was determined using Arginase Activity Assay (Sigma) according to the manufacturer's protocol.

To evaluate nitric oxide (NO) production as a measure of NOS (nitric oxide synthase) activity, Griess Reagent System (Promega) was used according to the manufacturer's protocol. Splenocytes or CECs were isolated from murine spleens and were cultured in non-adherent six-well plate $1 \times 10^6$ or $5 \times 10^5$ cells per 2 ml, respectively, for 24 h followed by supernatants collection and measurement of NO concentration.

**Bioinformatical analysis of arginase structure**. The structure and predicted binding energies for the complexes of PHZ, L-arginine, and 2-amino-6-borono-2-(2-(piperidin-1-yl)ethyl)hexanoic acid with both human and mouse arginases were compared. The 3D models of mouse arginases were proposed using available structures of human arginases (pdb|4hww and pdb|4hze for ARG1 and ARG2, respectively) as templates. Both templates shared more than 87% sequence identity with their respective target. The sequence to structure alignments between mouse arginases and selected templates were calculated with the muscle program[63]. The 3D structure was proposed with MODELLER[64]. Models quality was assessed with the Molprobity webserver[65]. Next, both human and mouse proteins were prepared for docking using the Chimera dock prep module. Molecular docking was carried out with two programs—GOLD[66] and Surflex[67]. The active site was specified based on the position of the inhibitor present in the active site of the arginase 1 (pdb|4hww). The default parameters of both programs were used.

To assess if PHZ remains stably bound to the active site of both human arginases short molecular dynamics simulations were performed. The initial configurations of ligand–protein complexes were derived from docking results for PHZ. For the PHZ–arginase complexes the simulation included the following steps. First protein and ligand were put in a dodecahedron box with the distance between solute and a box equal to 1 nm. The 0.1 M NaCl was added to the system including neutralizing counterions. After energy minimization using the steepest descent algorithm, 100 ps NVT and NPT simulation were carried out. For this modified Berendsen thermostat was used to maintain the temperature at 310 K using and Berendsen barostat to keep the pressure at 1 atm. Positions of both protein and ligand heavy atoms remained constrained. During the following 300 ps of simulation time, the ligand's constraints were gradually removed. Finally, an unconstrained 100 ns simulation is performed in which Berendsen barostat was replaced by Parrinello-Rahman barostat. During simulation short-range nonbonded interactions were cut off at 1.4 nm, with long-range electrostatics calculated using the particle mesh Ewald (PME) algorithm. Bonds were constrained using the lincs algorithm. Simulations were carried out with Gromacs[68] using the gromos54a7 force field, modified to include parameters for $Mn^{2+}$ ion adopted from[69]. Spc model was used for water molecules. Parameters for the ligand were obtained with Automated Topology Builder (ATB)[70].

Additional analyses were performed to assess if PHZ can migrate to the arginase active site when present in solute in high concentration. For this analysis protein was put in dodecahedron box with the distance between solute and a box equal to 1.5 nm in which 6 PHZ molecules were placed randomly. This corresponds to 0.02 M concentration of the compound. A similar simulation setup to one described above was used with exception that ligand molecules remained unconstrained throughout simulation.

**Protein carbonylation assay**. The carbonyl content of proteins was determined in a 2,4-DNPH reaction. Five micrograms of murine ARG1 (Cloud-System Corp) was resuspended in 400 μl of distilled water and incubated with PHZ (10 μM), PHZ (10 μM) with NAC (10 mM), $H_2O_2$ (10 mM), or water (negative control) as indicated in the Fig. 3d for 1 h at 37 °C. Proteins were precipitated with 10% TCA. The precipitates were treated with either 2 N HCl alone (control) or 2 N HCl containing 5 mg/ml 2,4-DNPH at RT for 30 min. The resulting hydrazones were precipitated in 10% TCA and then washed three times with ethanol–ethyl acetate (1:1). Final precipitates were dissolved in 8 M guanidine chloride. Equal amounts of proteins were separated on 4–12% SDS-polyacrylamide gel (Bio-Rad), transferred onto nitrocellulose membranes (Bio-Rad) blocked with TBST [Tris-buffered saline (pH 7.4) and 0.05% Tween 20] supplemented with 5% non-fat milk. Anti-DNP antibodies (Life Diagnostics, Inc.) at concentration 1 U/ml were used for overnight incubation at 4 °C. After washing with TBST, the membranes were incubated with horseradish peroxidase-coupled secondary antibodies (Jackson Immunores.). The reaction was developed using SuperSignal™ West Femto Maximum Sensitivity Substrate (Thermo Fisher Scientific) and imaged using ChemiDoc Touch Gel Imaging System (Bio-Rad). Densitometry was done using ImageJ software.

**RNA isolation from CECs, reverse transcription, and quantitative polymerase chain reaction**. Total RNA was isolated from CECs isolated from murine spleens using RNeasy Mini Kit (Qiagen). RNA was subjected to reverse transcription using the GoScript™ Reverse Transcriptase system (Promega). All qPCRs were performed in MicroAmp Fast Optical 96 WellReaction Plates (Thermo Fisher Scientific) using AppliedBiosystems 7500 Fast Real-Time PCR System with 7500Software V2.0.6 (Thermo Fisher Scientific). Samples were assayed in triplicates. Primers sequences used in the study: *ARG1* forward 5′- CTCCAAGCCAAAGTCCTTAGAG-3′, reverse 5′-AGGAGCTGTCATTAGGGACATC-3′; *ARG2* forward 5′-AGGAGTG-GAATATGGTCCAGC-3′, reverse 5′-GGGATCATCTTGTGGGACATT-3′; and GAPDH forward 5′-GAAGGTGGTGAAGCAGGCATC-3′, reverse 5′-GCATC-GAAGGTGGAAGAGTGG-3′ as an endogenous control. The mean Ct values of a target gene and endogenous control were used to calculate relative expression using the $2^{-\Delta Ct}$ method.

**Western blot**. Splenocytes lysates were prepared using Cell Lysis Buffer (#9803, Cell Signaling Technology) supplemented with protease inhibitors (Roche) according to the manufacturer's protocol. Total protein concentration was assessed using Pierce BCA Protein Assay Kit (Thermo Fisher Scientific). Equal amounts of whole-cell protein lysates samples were boiled in Laemmli loading buffer, separated on 4–12% SDS-polyacrylamide gel (Bio-Rad), transferred onto nitrocellulose membranes (Bio-Rad) blocked with TBST [Tris-buffered saline (pH 7.4) and 0.05% Tween 20] supplemented with 5% non-fat milk. Anti-Arg1 antibodies (polyclonal, GTX109242, GeneTex) at dilution 1:2000 or anti-Arg2 antibodies (polyclonal, ab81505, Abcam) at dilution 1:1000 were used for overnight incubation at 4 °C. After washing with TBST, the membranes were incubated with horseradish peroxidase-coupled secondary antibodies (Jackson ImmunoResearch). The reaction was developed using SuperSignal™ West Femto Maximum Sensitivity Substrate (Thermo Fisher Scientific) and imaged using ChemiDoc Touch Gel Imaging System (Bio-Rad). After imaging, bound antibodies were removed from membranes using Restore™ PLUS Western Blot Stripping Buffer (Thermo Fisher Scientific), followed by blocking with TBST supplemented with 5% non-fat milk. Next, the membranes were incubated with anti-β-Actin (A5060, Santa Cruz) conjugated with peroxidase. Densitometric quantifications were done using ImageJ software.

**Erythroid cells differentiation**. CECs were differentiated from human PBMCs according to the protocol by Heshusius et al.[71] with modifications. Human PBMC were purified from buffy coats from healthy donors by density separation using Lymphoprep (STEMCELL Technologies). PBMC were seeded at $1 \times 10^6$ cells/ml in erythroid differentiation-promoting medium based on StemSpan™ Serum-Free Expansion Medium (SFEM) supplemented with human recombinant EPO (2 U/ml, Roche), human recombinant stem cell factor (25 ng/ml, R&D Systems), dexamethasone (1 μM, Sigma-Aldrich), human recombinant insulin (10 ng/ml, Sigma-Aldrich), L-glutamine (2 mM, Sigma-Aldrich), iron-saturated holo-transferrin (20 μg/ml, Sigma-Aldrich), sodium pyruvate (1 mM, Gibco), MEM non-essential amino acids (1% v/v, Gibco), bovine serum albumin (0.1% m/v, Sigma-Aldrich), EmbryoMax Nucleosides (1% v/v, Merck), and 100 U/ml penicillin and 100 μg/ml streptomycin (Sigma-Aldrich). The expansion and differentiation of CECs were assessed by flow cytometry.

**Statistics and reproducibility**. Data are shown as means ± SD or means ± SEM, as indicated in the figure legends. Graphpad Prism 8.4.3 (GraphPad Software) was used for statistical analyses. Data distribution was tested using the Shapiro–Wilk test, D'Agostino & Pearson test, and Kolmogorov–Smirnov test. Statistical analyses of three or more groups were compared using one-way analysis of variance (ANOVA) or Brown–Forsythe ANOVA followed by Tukey's, Dunnett's, or Bonferroni's multiple comparisons test or Kruskal–Wallis test followed by Dunn's multiple comparisons test. Repeated measures ANOVA with Sidak's or Holm–Sidak's post hoc tests were used to analyze the differences in paired samples. Statistical analyses of two groups were compared using unpaired *t*-test, paired *t*-test, or Mann–Whitney test. Methods of statistical analyses are defined in every figure legend. A *P* value of less than 0.05 was statistically significant. Each experiment was performed in technical duplicates or triplicates. The number of biological replicates for each experiment is mentioned in the figure legends.

**Reporting summary**. Further information on research design is available in the Nature Research Reporting Summary linked to this article.

## Data availability

Source data underlying the graphs and charts presented in the figures are available in the Supplementary Data 2. Uncropped western blots are provided in the Supplementary Data 2 file. Any remaining information can be obtained from corresponding authors upon reasonable request.

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

## Acknowledgements
This work has been co-supported by grants 013/RID/2018/19 (Regional Initiative for Excellence) from the Polish Ministry of Education and Science, project budget 12,000,000 PLN (to J.G.), 2019/35/B/NZ6/00540 (to D.N.), 2017/25/B/NZ6/01139 (to J.G.), and 2016/23/B/NZ6/03463 (to D.N.) from the National Science Center in Poland. D.P. and M.L. are financed by TEAM program from the Foundation for Polish Science co-financed by the European Union under the European Regional Development Fund as well as grants 2019/35/O/ST6/02484 and 2020/37/B/NZ2/03757 from the National Science Center in Poland. M.L. is funded by IDUB against COVID-19 project granted by the Warsaw University of Technology under the program Excellence Initiative: Research University (IDUB). Some elements of the figures were generated with Biorender.com.

## Author contributions
T.M.G. designed and supervised the study, conducted the experiments, analyzed the data, and wrote the manuscript. A.S. participated in in vivo studies. Z.R. participated in in vitro experiments. M.L. and D.P. performed molecular docking and molecular dynamics simulations, K.K. performed real-time qPCR and participated in in vitro experiments, M.M. and O.C. collected and provided human blood samples, A.R.-L. performed analysis of murine blood, M.J. participated in in vitro experiments, P.P. and M.M.-G. carried out arginase activity assays, R.B. designed and synthesized OAT-1746, M.W. bred and provided *Arg2*$^{-/-}$ mice, A.T. and G.B. collected and provided HSPCs. J.G. conceived, designed and supervised the study, provided funding and wrote the manuscript. D.N. provided funding, performed in vivo studies, designed and supervised the study, and wrote the manuscript. All authors edited and approved the final manuscript.

## Competing interests
P.P., M.M.-G. and R.B. are employees of OncoArendi Therapeutics, Warsaw, Poland. The remaining authors declare no competing interests.
