## [Peer Review File · Communications Biology]

Reviewers' comments:

Reviewer #1 (Remarks to the Author):

Here, Grzywa et al. report that specifically immature erythroblasts display immunosuppressive functionality by inhibiting the proliferation of CD4 and CD8 T-cells in vivo (OVA model) but also in vitro (co-culture assays, stimulating T-cells with anti-CD3/anti-CD28). The immune-suppressive role of erythroblast on T-cells has been described before as well as the involvement of ARG2 (and possibly ARG1) in this process, as such the message is not novel [Namdar, 2017; Elahi, 2013; Delyea, 2018]. However, the authors build on this knowledge by addressing the role of erythroid mass during anemia and stress erythropoiesis on adaptive immunity (although one could argue that Zhao et al. in 2019 has shown this in their tumor model). In addition, they show that absence of erythroblast immunosuppressive functionality upon inducing anemia with phenyl-hydrazide [Zhao, 2019] is due to pharmacological interference of PHZ on arginase function itself. They confirm that human immature erythroblasts (CD71+CD235dim) confer the CD4/CD8 immunosuppressive function. The paper is nicely written and easy to read. However, besides novelty, there are some other issues that need be to addressed.

Major points:

Results:

Figure 1 is a repetition of what is already known. Stress erythropoiesis is induced upon anemia. Figure 1D and f: It would be interesting to see CD45+ staining together with TER119 and CD71 to show the exact progenitor stage at which CD45 is expressed. Gating in fig1e is more narrow compared to figure 1a...this may skew the analysis to more differentiated erythroid cells. Where do the CD45+ cells end up in this gating strategy? In general, this also touches on assessing the immune suppressive ability of the erythroid cells, as the sorted cells are a mixed bunch of immature EBL that can have suppressive ability and more mature EBL that are significantly inhibited in this ability. If it is known that the immature erythroblasts are conveying this effect, why did the authors not sort the immature fraction for their experimental approach. Or start with an experiment in which they show that indeed the more mature fractions do not show the immunosuppressive role and then move to sorted populations.

Figure 3A comment on the difference between control and anemia induced EPC. How about CD45+ EPC?

Figure 3c show a representative histogram. Why are only 30-40% of the EPC ARG2 positive? Same question for ARG1. Percentage of ROS levels in ARG cells? A plot in which the ARG staining is combined with CD71 and TER119 (or CD235 for human) would be informative as is it not suspected that the most immature population would be 100% positive?

Figure 4D, please add to the legends what the difference lanes indicate. Does negative mean without oxygen treatment? It is a bit difficult to read now. But it cannot be degradation as the levels of ARG1 and 2 arte high. What about ter119 induced anemia? That should also do the job.

Figure 5b, what about PHZ? To conclude that PHZ is indeed inhibiting ARG, I expected an experiment in which PHZ was added during co-culture in an experimental approach like figure 5b as this would result in the inability to immunosuppress? Importantly, what is the viability and number of EPC during the treatments with NAC and ARGi?

Supplemental figure 9a does not show ARG2^{-/-} cells without the NHA induction, please show this.

Figure 5d. please show that AGR2^{-/-} mice still have good stress erythropoiesis in the spleen. in fact it would be best to have figure 1 completely for the ARG2^{-/-}. What about deltaCD3 in the ARG2 knockout?

Figure 6a and b. why are these called EPC as in fact I would argue that these are just reticulocytes. Please specify this better and rule out these are not just reticulocytes. If you want to make a statement for nucleated erythroid precursors, then please provide a nuclear staining to proof this (maybe using DRAQ5). If these are reticulocytes then this is also not novel of course, as it is a common diagnostic marker. Concerning the CD45+ EPC, show a quantification of this compared to control.

figure 7a. I must stress that this is a rather poor bone marrow aspirate as normally one would see at least 15-20% of nucleated erythroid precursors. In fact bone marrow is not primarily composed of erythrocytes, this is blood contamination.

Supplemental figure 12b is this from the CD71/CD235/CD44 gating strategy? Because if this is the EPC gate as indicated in figure 7a. I am surprised with the CD45 expression. Please backgate the CD45 expression onto the CD71/CD235/CD44 gates or the used erythroid gate.

Figure 7b,c please provide quantifications of this.

Figure 7d-e but these are rather late erythroid precursor cells, if the gate in 7a is used. In fact the maturing cells that do not display the suppressive effect as mentioned in figure 8. Provide a sorting of MACS purity graph in supplemental so the readers understand what EPC were used in figure 7d. Could it be that this is the reason why the suppressive ability is so low (figure 7d end e)? Supplemental data 17: how are these PBSCs defined. Are these CD34+ MACS purified Hemopoietic stem and progenitor cells. It is unclear what with what cells these experiments were performed. Figure 8: So here is your explanation why the bone marrow cells did not perform well in the T-cell proliferation assay in figure 7. The gating was not optimal and only CD235+ CD71+ later stage basophilic erythroblasts were taken in that experiment whereas the PBMC derived erythroid cells at day 13 indicate that it is the erythroblasts that are CD71^{high}, CD235^{low} that actually do the suppressive job. So, what is missing in figure 8 is the inhibitor assay using ARG inhibitors and NAC on the human EPC. In addition, spleens are a bona fide immunological site whereas human bone marrow is less so. What does it mean that EPC or erythroid precursors cells inhibit T-cell proliferations in the bone marrow. Is that to protect the bone marrow from large scale inflammation or something like that? Please comment in the discussion the difference between human and mice (as the authors know, stress erythropoiesis in humans does not occur in the spleen).

Discussion:

First sentence. Well, murine yes, but human anemia, this they did not show.

Please refrain from writing that EPCs are expanded in anemic patients as those are just retics in peripheral blood (if they would look in bone marrow they would indeed have found this). In fact as indicated above these are probably reticulocytes that will not suppress the T-cells as they are too late in differentiation. Moreover, even if these were nucleated erythroid precursors cells these would be too late in diff to exert their effect! As indicated by the authors in figure 8. So, sentence 339 needs to be rewritten or deleted.

Discussion: 354: "We showed that human EPCs acquire immunomodulatory properties during erythroid differentiation and are the most potent in CD71^{hi}CD49d^{hi}CD44^{hi}CD45⁺ EPCs." I did not see this graph!

Minor points:

Intro:

Lines 71 to 73. Please add the word "in mice" as stress erythropoiesis in humans occurs in the bone marrow.

Figure 2c only shows one of the conditions, please add all conditions.

During co-culture what happens to the erythroid cells with respect to ARG1/2 expression, differentiation and viability?

Reviewer #2 (Remarks to the Author):

The paper by Grzywa et al identify increased expression of arginase 1 and 2 as at least one of the mechanisms used by erythroid precursors from anemic subjects to suppress T-cell functions.

The scientific premise of the study is robust and described in sufficient detail.

The experimental design is also robust and involves a good number of in vitro and in vivo mouse and human experimental models. The data presented are overall of good quality and well support the conclusion.

Major comment

1) The presentation of the data is not of the high standard required for publication and needs to be greatly improved:

a) There are too many abbreviations, which make the paper difficult to read.

b) There is little attention to the methodological differences implemented to analyze murine and human cells.

c) There is an awkward use of terminology. The authors indicate in the title and in the text EPC as erythroid progenitor cells while by the markers used for cell identification and the graphical abstract clearly indicate that they mean erythroid precursors instead.

d) The language used to describe and discuss the results is convoluted and should be simplified.

- e) The numerous alternative hypothesis discussed while interpreting the results reduce the perception of the quality of the experimental design.
- 2) Mice phlebotomy. Anemia was induced in mice by phlebotomy of 100 μ L of blood. This volume corresponds to 0.5% of the all blood volume of a mouse. A blood donation, 500 mL, corresponds to 10% of the blood volume of the donor. Do the authors mean that blood donation suppresses the immune functions of the volunteer? The long history of follow up of regular blood donors has not provided any clinical evidence that the procedure induces an immunosuppressive state. This important point should be clarified by analyzing the immunosuppressive potential of EPC and the immune state of T cells from regular blood donations.

Minor comment

- 1) The patients included in the study are generically indicated as anemic or not anemic. Their diagnosis should be clarified to assure that the patients included in the two groups are clinically homogeneous.
- 3) To recover from the hemolytic anemia induced by phenylhydrazine mice activate the glucocorticoid receptor pathway and experience increased levels of cortisol in their blood. Given the strong immuno-suppressive role of cortisol how the authors exclude that the immunosuppressive effects of EPCs from these mice is not mediated by cortisol carried over in the cytoplasm of the cells?

Reviewer #3 (Remarks to the Author):

This manuscript provides some interesting evidence in support of previously published articles in the field. Although this is an area of intense interest, significant modifications and improvements are required.

One major concern is the definition of these cells as EPCs. Based on the vast literature EPCs are considered as erythroid progenitors expressing CD45, a possibility that some are nucleated. However, the authors in many plots combine CD45+ and CD45- cells as EPCs, which is incorrect. Besides, the authors throughout the manuscript refer to CD71+ erythroid cells (CECs) in multiple published articles as EPCs, however, in those articles EPCs are defined as CD45+CD71+ erythroid cells. Therefore, using the term CD71+ erythroid cells is scientifically more acceptable and comprehensive than EPCs. The term EPCs does not reflect the literature and the manuscript content.

Secondly, the manuscript can be more focused and better organized. The method section lacks some experimental details. The ratio of CECs to T cells is inconsistent and on many occasions not mentioned. Is the ratio of 1:3 or 1:2 is physiologically relevant? Why this ratio was used but not like 1:1? It is not clear how spleens were processed, did the author use RBC lysis buffer or not?

Almost 20 supplementary figures are not mentioned in the text and as a reviewer/reader you are unaware of this until you spend hours on the paper and finally you find them at the end. I believe this is any journal's policy to incorporate all the supplementary figures in the text.

Thirdly, although authors have invested a substantial amount of time and resources in this manuscript, it's not up to the standard of a scientific article. This is evident in inconsistency in the flow gating strategy and lack of experimental details. Normally, Fig. 1a should provide the reader with a positive impression of the work, unfortunately, this is not the case in this article.

Overall my points are:

Figure 1a should be considered as the most important part of the paper since all the data are based on this comparison. However, the quality of flow plots is poor and as you can see there is a cut in the flow cytometry plots, so the percentages are inaccurate and the graphs that are made based on these percentages might not be accurate. The authors could have reduced the voltage for CD71. The other issue is that it is unclear whether such staining was performed on splenocytes after using RBC lysis buffer or it's in the presence of mature RBCs. Based on the high % of TER119 cells I suggest its done on total splenocytes but needs to be mentioned. Also, the % of mature

RBCs can skew the % in adults versus newborns as shown by a lower frequency of TER119+ cells in the neonatal spleen.

Figure 1c, is misleading, of course a 3 day old pup has 2-3 x10⁶ splenocytes compared to an adult mouse which has 100 x10⁶ splenocytes. Thus, total CECs even if they constitute 70% of splenocytes they are lower when comparing 2-3 to 100 x10⁶?

Fig 1d, the percentage of CD45+ vs CD45- EPCs are shown. However, there is no representative plot to support the data.

Fig 1e, the authors use CD44 and cell size to categorize CECs into different subsets and they referenced reference # 13. However, the gating strategy in reference #13 is placed on distinct cell populations, which is not the case here. The gate III is placed on two different/distinct cell populations. The gate for II and III population should be moved to the right and data should be re-analyzed again.

Fig 2c shows the purity plot for CECs. Besides the cut in the gating population, the purity is 89.5% isn't that great. This shows that there is contamination by other cells that may have suppressive functions such as T-regs. At least the authors should show the nature of the other 10% of cells.

Antigen-specific T cell assays are very interesting; however, the authors do not explain the rationale for using the ratio of 1:2 (T cell: CECs)? Is this physiologically relevant? This should be mentioned and discussed.

In lines 139 and 140, it has been mentioned that ROS levels in murine EPCs were significantly lower than in myeloid -cells and T-cells. However, the data in fig 3b shows that CD45+ cells have almost the same level of ROS compared to CD3+ cells and CD11b+ cells. Since CD45 EPCs have the main functional suppressive function in CECs, it is recommended that the authors make another conclusion based on the production of ROS on CD45+ EPCs vs other cells. It's unclear how EPCs are defined here since some of them are CD45+. Do CD45+ cells in this plot contains CD45+CECs?

Supplementary Fig 3, lacks representative plots for TNF-a expression in CD11b with proper controls.

Line 125-126, Since in the spleen of anemic mice the expansion of EPCs was the most substantial (Supplementary Fig. 3d). However, the only used marker in this graph is CD71 and how the authors are confident that all CD71+ cells are EPCs? Of course, any activated immune cell can express CD71. Please be careful in data interpretation.

Supplementary Fig. 4a, two subpopulations of CECs as ROS low and ROS high are in fact erythroid precursors (CD45^{low/-}) and erythroid progenitors (CD45^{+/high}). The authors could have done CD45 staining the differentiate these two subpopulations.

Fig 3a, the representative plot shows stained cells but it's unclear the source of cells? NHA or HA-PHZ and having RBCs would be a good negative control to be included.

Fig 3c and 3d show the expression of Arginase 2 and 1 in different study groups. However, there is no representative plot to support the graphs. More importantly, Args are normally presented not by % but by intensity. These data should be presented as the MFI comparing different groups.

Similarly, the data in supplementary fig. 5 is not supported by representative plots.

The q-PCR data in fig 6A shows a significant difference between groups. However, considering the high variability of results and high overlap, it seems that the difference shouldn't be significant. It is suggested that authors reanalyze q-PCR data.

In line 186-189, it is mentioned that incubation of recombinant ARG1 with PHZ in the presence of oxygen led to a significant increase in the carbonylation of the enzyme that was reduced by concomitant incubation with N- acetylcysteine. However, this is not supported by what has been

shown in Fig 4d that shows the addition of NAC results in a partial reversal of the carbonylation of the enzyme which was not statistically significant. It is recommending that the authors provide more convincing data regarding the effect of PHZ on ARG1 oxidative changes.

Fig 5a shows the proliferation and expression of activation markers by CD4 T cells. However, although the graph shows some data for the proliferation of T cells in the presence and absence of EPCs, it is not clarified in the text of the paper. Are they CD4 or CD8 T cells since in the material and method section, it seems that only CD4 and CD8 T cells but not total T cells were isolated. The used ratio of T cells to CECs is not mentioned?

Please clarify if ARGi, OAT-174619 inhibits arginase I or II?

What is the EPC-conditioned medium? The authors didn't describe in the material and method what is this medium?

Fig 5b shows that the addition of ARGi or ROSi completely abrogates the suppressive function of EPCs on T cell proliferation. However, the results of fig 5d show that they only partially reverse the suppressive function of EPCs (stats not provided). It is required that authors describe the reason for this discrepancy.

Fig 6a-b: for showing the percentage of EPCs in adults, the cells MUST be double positive for CD71 and CD235a. However, the gating strategy (b) only shows CD71+ cells. The plots do not show any CD235a+ cells? There are two major problems here; a) CD71+ cells can be a combination of EPCs and other activated cells. b) the anti-CD235a antibody was not included or did not work. These plots show that the gated population is only CD71+ cells. Consequently, all the data presented based on this gating will be questionable?

The authors have provided interesting data on CECs in anemic individuals but the gating strategy in Fig 6g is unacceptable. Its unclear why the gate is so wide to include CD71+CD235a- cells as EPCs? How authors are confident that CD71+ cells are not activated T, B, and other immune cells? Consequently, the cumulative data based on this gating strategy need to be re-evaluated. Interestingly, the authors in Fig. 7a have used the correct gating strategy for EPCs. Now I wonder why there is inconsistency in the gating strategy for the same cell type (EPCs) by simply looking at Fig 6a, 6g and 7a!

Fig. 6h lacks representative plots for IFN-r expression with appropriate controls. Also, the figure legends stated that cells were stimulated in the presence of a Golgi blocker for 12hrs? More than 6 hr incubation, the Golgi blocker kills the cells? Representative plots with viability staining is required, based on the manufacturing instruction 6 hr is the max time.

Supplementary Fig. 8 the ratio of EPCs to T cells is missing.

The authors used human erythroleukemic cell line K562, HEL92.1.7, and TF-I. However, they didn't describe at which stage of differentiation these cells were? It is recommended that they follow with a brief description of these cells' lines. This is further highlighted by the variable expression of CD71 and CD235a on these cells. Again, for the proliferation assay, please justify why the 3:1 ratio of EPCs to CD4 or CD8 T cells has been used. By the way, in Fig 8, it is mentioned that EPCs with suppressive function have high CD71 but fig 13a shows that K562 cells have low CD71, so how they have a suppressive function?

There is another issue that the authors did not reference their supplemental from the Supplemental fig 19 to 36 in the text. This is hard to follow where/if any representative plot is provided for particular staining.

We would like to thank the reviewers for their insightful comments that improve the presentation and quality of our paper. We have performed additional analyses and experiments to address all issues raised by the Reviewers. We have considered their comments and suggestions carefully and made revisions accordingly.

In the revised manuscript, all changes in the text and all new text fragments are indicated in **red**. Similarly, in this document the text from the revised manuscript is indicated in **red**. A point-by-point response to each of the reviewers' comment is provided below. We **have numbered reviewers' comments**, according to the journal guidelines. We divided some comments that were composed of several questions into "a" and "b" subsections to increase the readability and clarity of the answers. Our responses to reviewers' comments are in **blue**. All line numbers in this document refer to those in the revised manuscript. We have attached novel or modified figure panels to our point-by-point response.

Reviewer #1 (Remarks to the Author):

Here, Grzywa et al. report that specifically immature erythroblasts display immunosuppressive functionality by inhibiting the proliferation of CD4 and CD8 T-cells in vivo (OVA model) but also in vitro (co-culture assays, stimulating T-cells with anti-CD3/anti-CD28). The immune-suppressive role of erythroblast on T-cells has been described before as well as the involvement of ARG2 (and possibly ARG1) in this process, as such the message is not novel [Namdar, 2017; Elahi, 2013; Delyea, 2018]. However, the authors build on this knowledge by addressing the role of erythroid mass during anemia and stress erythropoiesis on adaptive immunity (although one could argue that Zhao et al. in 2019 has shown this in their tumor model). In addition, they show that absence of erythroblast immunosuppressive functionality upon inducing anemia with phenyl-hydrazide [Zhao, 2019] is due to pharmacological interference of PHZ on arginase function itself. They confirm that human immature erythroblasts (CD71+CD235dim) confer the CD4/CD8 immunosuppressive function. The paper is nicely written and easy to read. However, besides novelty, there are some other issues that need be to addressed.

Authors' response:

We thank the Reviewer for critical reading of our work and insightful comments that helped us improve the manuscript.

Major points:

Results:

Reviewer #1 comment 1. Figure 1 is a repetition of what is already known. Stress erythropoiesis is induced upon anemia.

Authors' response:

We agree with the reviewer that it is well established that stress erythropoiesis is induced upon anemia in mice and we now refer to the original article reporting this finding (ref #14 and 15). Our data presented in Figure 1 show expansion of CECs in three different models of

anemia and compare it with the physiological abundance of CECs in neonatal mice, in which immunoregulatory properties of CECs are best described¹. Moreover, in Figure 1 we present a detailed analysis of CECs in murine spleen. Further, we show expansion of early-stage CECs, that are further demonstrated to possess significant immunoregulatory properties. Thus, the data shown in Figure 1 provide an overview of parameters that are in our opinion critical to interpret the findings reported in the whole manuscript.

Reviewer #1 comment 2a. Figure 1D and f: It would be interesting to see CD45+ staining together with TER119 and CD71 to show the exact progenitor stage at which CD45 is expressed.

Authors' response:

According to the Reviewer's suggestion we have included additional plots in Fig. 1. We show a representative CD45 staining in CECs presented in Fig. 1d as well as CD45 staining in TER119⁻ cells as a positive control. Moreover, we show CD45 staining of CECs together with CD71 (Fig. 1g, left panel) as well as the comparison of the frequency of CECs in different developmental stages in the fraction of CD45⁺ and CD45⁻ CECs (Fig. 1g, right panel). We have prepared additional graph showing changes in the intensity of staining for CD45 in different developmental stages of CECs (Fig. 1h). CD45 is expressed at the earliest stages of differentiation (stages I-II), followed by a progressive decline in stages III-IV, to a complete loss in stage V.

Fig. 1. h, Percentages of CD45⁺ CECs in different developmental stages in the spleen of NHA mice (n=5). Histograms show the fluorescence of CD45 – BV711. Red blood cells (RBCs) are presented as negative control.

Reviewer #1 comment 2b. Gating in fig1e is more narrow compared to figure 1a...this may skew the analysis to more differentiated erythroid cells.

Authors' response:

We have corrected Fig. 1e to demonstrate representative plots of gating strategy used to analyze the data presented in Fig. 1f. This gating strategy of CECs involves CD71^{mid/high} and TER119^{mid/high} cells to include all developmental stages of CECs. Corrected gating has not changed the results of our analyses.

e

Fig. 1e, Gating strategy for CECs developmental stages based on CD44 expression and cells size.

Reviewer #1 comment 2c. Where do the CD45⁺ cells end up in this gating strategy?

Authors' response:

CD45⁺ CECs are predominantly at the I-III developmental stages. We have added the plot presenting the CD44/FSC gating strategy of CD45⁺ and CD45⁻ CECs (Fig. 1g). Moreover, we show the fraction of CD45⁺ CECs and representative CD45 staining of CECs in different developmental stages (Fig. 1h).

Reviewer #1 comment 3. In general, this also touches on assessing the immune suppressive ability of the erythroid cells, as the sorted cells are a mixed bunch of immature EBL that can have suppressive ability and more mature EBL that are significantly inhibited in this ability. If it is known that the immature erythroblasts are conveying this effect, why did the authors not sort the immature fraction for their experimental approach. Or start with an experiment in which they show that indeed the more mature fractions do not show the immunosuppressive role and then move to sorted populations.

Authors' response:

We have tried to sort the population of immature cells from the spleens of mice. However, acquisition of sufficient number of these cells requires long sorting procedures and is associated with significant compromise in cells viability. Thus, we performed an additional experiment in which we demonstrated that nucleated CECs (developmental stages I-III, Fig. 1i) with high ROS (Supplementary Fig. 3) and ARG levels (Supplementary Fig. 6) are more potent immunosuppressors than a whole heterogeneous population of CECs.

To confirm that immature CECs have the most potent immunosuppressive properties we used a continuous CECs culture from PBMCs (Fig. 10). Using this approach, we were also able to demonstrate that C71^{high}CD235a^{dim} CECs are responsible for the suppressive effects of CECs.

Reviewer #1 comment 4a. Figure 3A comment on the difference between control and anemia induced EPC.

Authors' response:

Based on CellRox Green staining, ROS levels were significantly lower in HA-PHZ CECs than in control CECs (p=0.0024), but the difference between NHA and control CECs were not statistically significant (p=0.9127, Kruskal-Wallis test with Dunn's multiple comparisons

test). On the contrary, based on DCFDA staining, the ROS levels were slightly higher in anemic CECs than in control CECs, however the differences were not statistically significant (control vs NHA $p=0.0743$, control vs HA-PHZ $p=0.0978$, Kruskal-Wallis test with Dunn's multiple comparisons test). However, both CellROX and DCFDA are diffusible probes and cannot differentiate neither cellular compartments nor ROS subspecies. Therefore, we focused on the differences between various developmental stages of CECs or different types of cells within one group, rather than on the comparison ROS levels in different groups (control and anemia), especially since two probes give different results.

Reviewer #1 comment 4b. How about CD45⁺ EPC?

Authors' response:

Indeed, in controls and NHA mice the ROS levels were significantly higher in CD45⁺ than CD45⁻ CECs. We have included graphs presenting ROS levels based on CellRox Green and DCFDA fluorescence intensity in CD45⁺ and CD45⁻ CECs (Supplementary Fig. 5e-f).

Supplementary Fig. 5e,f, ROS levels in CD45⁺ CECs and CD45⁻ CECs based on mean fluorescence intensity (MFI) of CellRox Green – FITC (**e**) and 2',7'-dichlorodihydrofluorescein diacetate (DCFDA) – FITC (**f**) in anemic (NHA n=6, HA-PHZ n=6) and control (n=6) mice. *P* values were calculated with paired *t*-test.

Moreover, we have included the analysis of CD45⁺ CECs in the Fig. 3b to enable the comparison of the ROS levels in CECs, CD45⁺ CECs and immune lineage cells.

Fig. 3b, Mean Fluorescence Intensity (MFI) of CellROX Green – FITC in CECs (CD71⁺TER119⁺), CD45⁺ CECs (CD45⁺CD71⁺TER119⁺), leukocytes (CD45⁺TER119⁻), T-cells (CD45⁺CD3e⁺), and myeloid cells (CD45⁺CD11b⁺) (n=18). *P* values calculated with Friedman test with Dunn’s post-hoc test.

Reviewer #1 comment 5a. Figure 3c show a representative histogram. Why are only 30-40% of the EPC ARG2 positive? Same question for ARG1. Percentage of ROS levels in ARG cells? A plot in which the ARG staining is combined with CD71 and TER119 (or CD235 for human) would be informative as is it not suspected that the most immature population would be 100% positive?

Authors’ response:

We now show representative histograms for ARG2 staining (Fig. 1d).

Fig. 3d, Mean Fluorescence Intensity (MFI) of ARG2-APC in CECs from control mice, anemic mice (NHA, HA-PHZ, HA-TER119), neonatal mice, and isotype control-IgG-treated mice (each group n=5). Histograms show the representative fluorescence of ARG2 – APC in CECs in different groups and unstained controls.

CECs are a heterogenous population and consist of cells at different developmental stages. Detailed analysis revealed that the most immature populations (stages I-III) are characterized by the highest expression of both ARG1 and ARG2. Thus, the percentage of ARG2⁺ CECs

results from the fraction of early-stage cells that are mostly ARG2⁺. We have included histograms of ARG1 and ARG2 levels in subsequent developmental stages of CECs as well as mature RBCs in revised Supplementary Fig. 6b,c. Moreover, we have added graphs presenting a fraction of ARG1⁺ and ARG2⁺ CECs at different developmental stages (Supplementary Fig. 6e,g).

Supplementary Fig. 6. a, Representative plot of CD71 and TER119 levels in live cells in the spleen of NHA mice and gating strategy of CECs developmental stages based on CD44 levels and relative cell size. **b**, Representative histograms of ARG1 expression in different developmental stages of CECs and RBCs in NHA mouse based on YFP mean fluorescence intensity (MFI) in reporter B6.129S4-Arg1^{tm1Lky/J} mice. Negative control represents the background fluorescence of YFP in wild-type C57Bl/6 mouse. **c**, Representative histograms of ARG2 expression in different developmental stages of CECs and RBCs based on intracellular staining in NHA mouse. Unstained control represents sample not stained for ARG2. **d,e**, ARG1 expression (**d**) and percentage of ARG1⁺ cells (**e**) in different developmental stages of CECs in NHA mice based on YFP mean fluorescence intensity (MFI) in reporter B6.129S4-Arg1^{tm1Lky/J} mice (n=8). **f,g**, ARG2 levels (**f**) and percentage of ARG2⁺ cells (**g**) in different developmental stages of CECs in NHA mice based on intracellular staining (n=4).

Plot presenting the CD71 and CD235a level together with histograms presenting ARG1 and ARG2 expression are provided in Supplementary Fig. 20.

Supplementary Fig. 20a, Representative plots of CD71 and CD235a expression in CECs differentiated from PBMC. **b**, Histograms of ARG1 and ARG2 expression in CECs ($CD71^+CD235a^+$), $CD71^+$ cells, RBCs ($CD71^-CD235a^+$) and non-erythroid cells ($CD71^-CD235a^-$).

Reviewer #1 comment 5b. Percentage of ROS levels in ARG cells?

Authors' response:

Due to technical reasons, we are unable to measure both ARG and ROS levels in the same staining. ARGs antibodies staining requires cell fixation, followed by permeabilization and intracellular staining with several washes, that substantially decreases the fluorescence of CellROX Green probe (Rebuttal Fig. 1a). Moreover, there is a substantial fluorescence spectrum overlap of the used ROS probes and YFP in YARG mice.

Rebuttal Fig. 1 CellROX Green – FITC fluorescence after intracellular staining. a, Comparison of ROS levels in CECs from the spleen of anemic mouse based on mean fluorescence intensity (MFI) of CellROX Green – FITC after standard protocol (membrane staining only) and after fixation, permeabilization, and intracellular staining for ARG1 and ARG2.

Nonetheless, based on the analysis of developmental stages, we concluded that early-stage CECs have the highest levels of both ARG and ROS (Supplementary Fig. 5 and Supplementary Fig. 6)

Reviewer #1 comment 6a. Figure 4D, please add to the legends what the difference lanes indicate. Does negative mean without oxygen treatment? It is a bit difficult to read now. But it

cannot be degradation as the levels of ARG1 and 2 are high. What about ter119 induced anemia? That should also do the job.

Authors' response:

Negative lane represents ARG1 precipitates treated with 2N HCl alone instead of 2N HCl containing 5 mg/ml 2,4-DNPH. We have included this information to the figure legends in the revised manuscript.

Reviewer #1 comment 6b. But it cannot be degradation as the levels of ARG1 and 2 are high.

Authors' response:

We have observed increased mRNA expression as well as increased fluorescence of YFP under *Arg1* promoter, that confirms robust expression of ARG1 in HA-PHZ. Based on intracellular staining, the protein levels of ARG1 were also increased by PHZ, however, to a lesser degree. Moreover, arginase activity was decreased in HA-PHZ CECs. Thus, we proposed that PHZ inhibits ARG activity and may promote ARG degradation leading to a compensatory increase in expression.

Reviewer #1 comment 7. Figure 5b, what about PHZ? To conclude that PHZ is indeed inhibiting ARG, I expected an experiment in which PHZ was added during co-culture in an experimental approach like figure 5b as this would result in the inability to immunosuppress?

Authors' response:

We thank reviewer for this comment. We have performed additional experiments and we have demonstrated that PHZ added to the CECs *ex vivo* diminishes their suppressive capacities (Supplementary Fig. 8).

Supplementary Figure 8. a,b, Effects of phenylhydrazine (PHZ, 100 μ M) on the proliferation of α CD3/ α CD28-stimulated CD4⁺ T-cells (a) and CD8⁺ (b) T-cells co-cultured with CECs in a ratio of 1:2 isolated from the spleens of NHA mice (n=5). Representative proliferation histograms of α CD3/ α CD28-stimulated CD4⁺ T-cells (a) and CD8⁺ (b) T-cells co-cultured with CECs in the presence of PHZ. Histograms show the fluorescence of CTV (CellTraceViolet) – BV421. P-values were calculated with repeated-measures ANOVA with Tukey's post-hoc test. Data show means \pm SEM. Each point in a represents data from

individual mice. n values are the numbers of mice used to obtain the data. The source data underlying Supplementary Fig. 8a,b are provided as a Source Data file.

Notably, PHZ (100 μ M) had no effects on T-cells or CECs viability (Rebuttal Fig. 2).

Rebuttal Fig. 2. CECs and T-cells viability during the PHZ treatment. a, Viability of CD3⁻ cells (CECs) incubated with PHZ at 100 μ M for 72h in CECs-T-cells co-culture based on ZombieNIR staining. **b,** Viability of CD3⁺ cells after incubation with PHZ at 100 μ M for 72h in CECs-T-cells co-culture based on ZombieNIR staining.

Reviewer #1 comment 8. Importantly, what is the viability and number of EPC during the treatments with NAC and ARGi?

Authors' response:

Both ARGi (OAT-1746) and ROSi (N-acetylcysteine) have no impact on murine nor human CECs viability in concentrations used in co-culture experiments (Rebuttal Fig. 3).

Rebuttal Fig. 3. CECs viability during the ARGi and ROSi treatment. **a**, Relative viability of erythroid cell lines incubated with ARGi (OAT-1746, 1.5 μ M) and ROSi (N-acetylcysteine, 200 μ M) measured in flow cytometry after propidium iodide (PI) staining (n=2). Viability is relative to controls (diluent-treated CECs). **b**, Relative viability of murine CECs incubated with ARGi (OAT-1746, 500 nM) and ROSi (N-acetylcysteine, 100 μ M) measured in flow cytometry after propidium iodide (PI) staining (n=5). Viability is relative to controls (diluent-treated CECs). **c**, Relative viability of human PBMC-derived CECs during the treatment with ARGi (OAT-1746, 500 nM) and ROSi (N-acetylcysteine, 100 μ M) measured in flow cytometry after propidium iodide (PI) staining (n=4). Viability is relative to controls (diluent-treated CECs). *P* values calculated with repeated measures ANOVA with Holm-Sidak's post-hoc tests. *P* value >0.05 for all comparisons. *n* values are the numbers of biological repetitions (**a**), number of mice (**b**) or individual patients (**c**) used to obtain the data.

We have also included this information in the Materials and methods section.

Lines 671, 729, 758 “ARGi and ROSi had no effects on T-cells nor CECs viability.”

We have included the numbers of CECs in the materials and methods section, subsections IFN- γ and TNF- α production assay of human cells, Murine T-cell proliferation assay, Human T-cell proliferation assay. Murine T-cells were plated in round-bottomed 96-well plates (5×10^4 cell per well). Murine CECs were added to the wells in a 1:2 ratio (1×10^5 CECs per well). Human T-cells were plated in round-bottomed 96-well plates (2×10^4 cell per well). Human CECs were added to the wells in a 1:2 ratio (4×10^5 CECs per well). In some experiments, the number of T-cells was lower. Thus, the number of CECs and the volume were correspondingly lowered to keep the same T-cell : CECs : medium volume ratio.

Reviewer #1 comment 9. Supplemental figure 9a does not show ARG2^{-/-} cells without the NHA induction, please show this.

Authors' response:

We have included these data in a revised Supplementary Figure 14b-c.

Supplementary Figure 14. b, Percentages of ARG1⁺ CECs in the spleen of control (n=5) and anemic wild-type mice (n=4) as well as Arg2^{-/-} knockout anemic (n=4) and control mice (n=3). c, ARG1 expression in CECs based on intracellular staining in the spleen of control (n=5) and anemic wild-type mice (n=4) as well as Arg2^{-/-} knockout anemic (n=4) and control mice (n=3).

Reviewer #1 comment 10a. Figure 5d. please show that ARG2^{-/-} mice still have good stress erythropoiesis in the spleen.

Authors' response:

We have performed additional experiments in which we administered anti-TER119 antibodies to induce hemolytic anemia (HA-TER119) followed by the monitoring of the RBC count in the peripheral blood. We have demonstrated that there are no differences in the stress erythropoiesis between the ARG2^{-/-} mice and ARG2^{+/+} wild-type mice (Supplementary Fig. 14).

Supplementary Figure 14. a, $Arg2^{-/-}$ knockout mice (n=3) and wild-type mice (n=3) were administered with 30 μg of anti-TER119 antibody to induce stress erythropoiesis. Red blood cell (RBC) counts were determined using automated hematology analyzer. $P > 0.05$ in all timepoints. P values were calculated with Mann-Whitney test.

Reviewer #1 comment 10b In fact it would be best to have figure 1 completely for the ARG2^{-/-}.

Authors' response:

According to the reviewer's suggestion, we have modified the figures and prepared two figures completely for the ARG2^{-/-} mice (Figure 6 and Supplementary Figure 14).

Figure 6. CECs from $Arg2^{-/-}$ mice have impaired suppressive properties. a-b, CD3 ζ level in CD4 $^{+}$ T-cells (a) and CD8 $^{+}$ T-cells (b) in the spleen of anemic and non-anemic $Arg2^{-/-}$ and $Arg2^{+/+}$ wild-type mice based on intracellular staining ($Arg2^{-/-}$ NHA n=4, other groups n=3). **c**, Proliferation of α CD3/ α CD28-stimulated CD4 $^{+}$ T-cells co-cultured with CECs isolated from NHA $Arg2^{-/-}$ mice or NHA wild-type $Arg2^{+/+}$ mice at a ratio 1:4 (T:cells:CECs). Representative proliferation histograms of α CD3/ α CD28-stimulated CD4 $^{+}$ T-cells co-cultured with CECs isolated from $Arg2^{-/-}$ mice or wild-type $Arg2^{+/+}$ mice in the presence of ARGi and/or ROSi. Histograms shows the fluorescence of CTV (CellTraceViolet) – V450. P values were calculated with one-way ANOVA with Bonferroni’s post-hoc test.

Supplementary Fig. 14 a, $Arg2^{-/-}$ knockout mice (n=3) and wild-type mice (n=3) were administered with 30 μ g of anti-TER119 antibody to induce stress erythropoiesis. Red blood cell (RBC) counts were determined using automated hematology analyzer. $P > 0.05$ in all timepoints. P values were calculated with Mann-Whitney test. **b**, Percentages of ARG1 $^{+}$ CECs in the spleen of control mice (n=5) and anemic wild-type (n=4) and $Arg2^{-/-}$ knockout anemic mice (n=4) and control mice (n=3). **c**, ARG1 expression in CECs based on intracellular staining in the spleen of control mice (n=5) and anemic wild-type (n=4) and $Arg2^{-/-}$ knockout anemic mice (n=4) and control mice (n=3). P values were calculated with one-way ANOVA with Bonferroni’s post-hoc test.

Reviewer #1 comment 11. What about deltaCD3 in the ARG2 knockout?

Authors’ response:

We have focused on CD3 ζ in T-cells since its levels is regulated by ROS and L-arginine availability^{2,3}. No study demonstrate regulation of CD3 delta by ROS nor arginase. Thus, we have focused on CD3 ζ . We have observed a statistically insignificant decrease of CD3 ζ level in CD4 $^{+}$ and CD8 $^{+}$ T-cells in ARG2 $^{-/-}$ NHA mice *in vivo* (Fig. 6a-b, attached above), confirming our findings from the *ex vivo* experiments (Fig. 5k-l). Altogether, these findings confirm that ARG2 is primarily responsible for the CD3 ζ decrease in anemic mice. A slightly decreased levels of CD3 ζ in ARG2 $^{-/-}$ NHA T-cells may result from the activity of ROS that also decrease CD3 ζ ³.

Reviewer #1 comment 12. Figure 6a and b. why are these called EPC as in fact I would argue that these are just reticulocytes. Please specify this better and rule out these are not just reticulocytes. If you want to make a statement for nucleated erythroid precursors, then please provide a nuclear staining to proof this (maybe using DRAQ5). If these are reticulocytes then this is also not novel of course, as it is a common diagnostic marker. Concerning the CD45+ EPC, show a quantification of this compared to control.

Authors' response:

We agree with the reviewer. Thus, we have changed the nomenclature from EPCs to CECs that include both erythroid progenitors and precursors. Moreover, we have included quantification of CD45+ CECs in anemic patients compared to controls (Supplementary Fig. 15b).

Supplementary Fig. 14b, Percentage of live CD45⁺ CECs in the population of PBMCs in controls (n=12) and anemic patients (n=13). *P* value was calculated using Mann-Whitney test.

Reviewer #1 comment 13. figure 7a. I must stress that this is a rather poor bone marrow aspirate as normally one would see at least 15-20% of nucleated erythroid precursors. In fact bone marrow is not primarily composed of erythrocytes, this is blood contamination. Supplemental figure 12b is this from the CD71/CD235/CD44 gating strategy? Because if this is the EPC gate as indicated in figure 7a. I am surprised with the CD45 expression. Please backgate the CD45 expression onto the CD71/CD235/CD44 gates or the used erythroid gate.

Authors' response:

Supplementary Fig. 16a (Supplementary Figure 12a in the previous version) is from the same gating strategy as in Fig. 8a (CD71⁺ CD235a⁺ cells, Figure 7a in the previous version). We have also included this information at the figures to clarify this issue (Supplementary Fig. 16a, 16c). In the revised form, Supplementary Fig. 16 presents CD45 expression of CECs (gate CD71⁺CD235a⁺, Supplementary Fig. 16c) as well as of CECs/CD44 gate (developmental stages, Supplementary Fig. 16d). We have added additional graphs and histograms, that include percentage of CECs at different developmental stages (Supplementary Fig. 16b).

Supplementary Figure 16. **a**, Representative dot plot of the gating of CECs developmental stages based on CD44 level and relative cell size (FSC-A) in CECs from human bone marrow. **b**, Percentage of CECs in different developmental stages in human bone marrow (n=10). **c**, Representative histogram of CD45 level in CECs from human bone marrow. **d**, Representative histogram of CD45 level in CECs at different developmental stages. Red blood cells (RBCs) showed as CD45-negative control. **e,f**, The level of ARG1 (**e**) and ARG2 (**f**) in CD45⁺ and CD45⁻ CECs from human bone marrow. **g**, Representative plot of CD71 and CD235a of isolated CECs from human bone marrow. *P* values were calculated using Mann-Whitney test (**d,e**).

Moreover, we have removed the sentence “Human bone marrow is predominantly composed of mature erythrocytes, however, EPCs constituted a substantial cell population” and replaced it with

Line 275 “Since the expansion of CECs in the peripheral blood of anemic individuals was not associated with the suppression of T-cells proliferation, we investigated the immunoregulatory properties of CECs from the healthy human bone marrow (Fig. 8a). CECs in the bone marrow are enriched with early-stage CECs (Supplementary Fig. 16a,b) and are predominantly CD45⁺ (Supplementary Fig. 16c,d).”

Reviewer #1 comment 14. Figure 7b,c please provide quantifications of this.

Authors’ response:

We have provided percentages of ARG1⁺ and ARG2⁺ CECs as well as the levels of ARG1 and ARG2 based on the MFI (Figure 8b-c).

Figure 8. b, Percentages of ARG1⁺ and ARG2⁺ CECs in the human bone marrow based on intracellular staining. **c,** Mean Fluorescence Intensity (MFI) of ARG1-PE and ARG2-APC of CECs. **d,** Representative histograms of ARG1 and ARG2 levels in CECs from human bone marrow. Fluorescence-minus-one (FMO) is shown as unstained controls.

Reviewer #1 comment 15a. Figure 7d-e but these are rather late erythroid precursor cells, if the gate in 7a is used. In fact the maturing cells that do not display the suppressive effect as mentioned in figure 8.

Authors' response:

Indeed, most of the CECs in the bone marrow aspirates were late-stage (Please see Supplementary Fig. 15b attached above), which is consistent with the literature data on healthy human bone marrow^{4,5}.

Reviewer #1 comment 15b. Provide a sorting of MACS purity graph in supplemental so the readers understand what EPC were used in figure 7d. Could It be that this is the reason why the suppressive ability is so low (figure 7d end e)?

We now provide a purity plot in Revised Supplementary Fig. 16g that presents representative data pertinent to CECs isolated using immunomagnetic positive selection with anti-CD71 antibody. We agree with the Reviewer that relatively weak suppression of T-cells proliferation may be caused a small fraction of the early-stage CECs that have the most potent immunoregulatory properties. We have discussed it in the revised manuscript:

Line 414 “Here, we showed that human CECs acquire immunomodulatory properties during erythroid differentiation and are the most potent in an early stage characterized by the CD71^{high} CD235a^{mid} phenotype. Further erythroid maturation is associated with the disappearance of the suppressive properties. These early-stage CECs are relatively rare in the peripheral blood of anemic individuals, which may be a reason for the lack of T-cell proliferation suppression. In contrast, human bone marrow is enriched in early-stage CECs that suppress T-cells proliferation. Nonetheless, the suppression of T-cells by bone marrow CECs was substantially weaker than that of CD71^{high} CD235a^{mid} CECs from ex vivo culture. This may be explained by the fact that in human bone marrow CECs at the earliest stages of differentiation are still much less abundant than late-stage CECs^{28,45}.”

Reviewer #1 comment 16. Supplemental data 17: how are these PBSCs defined. Are these CD34+ MACS purified Hemopoietic stem and progenitor cells. It is unclear what with what cells these experiments were performed.

Authors' response:

Peripheral blood stem cells (PBSCs) were defined as a hemopoietic stem and progenitor cells (HSPC)-enriched product of leukapheresis of GM-CSF-mobilized donor. PBSCs contained different populations of stem and progenitor cells and 50% of them were CD34⁺ (Supplementary Fig. 21a). We have clarified this in the revised text of the manuscript and in the materials and methods section. Notably, even in a high ratio (T-cells:PBSC 1:8) these cells had no effect on T-cell proliferation.

Line 316 “First, we investigated whether hematopoietic stem and progenitor cells (HSPCs) exert immunosuppressive effects. Mobilized hematopoietic stem cells obtained from peripheral blood (peripheral blood stem cells, PBSCs, Supplementary Fig. 21a) had high ARG1 and ARG2 levels (Supplementary Fig. 21b) and included only a small percentage of CECs (Supplementary Fig. 21c).”

Line 513 “Mobilized peripheral blood stem cells (PBSCs) were obtained from familial donors from the material remaining after allogeneic stem cell transplantation. PBSCs were mobilized with the granulocyte colony-stimulating factor (G-CSF) and isolated from the donor peripheral blood using apheresis according to the standard clinical protocol. Informed consent was obtained from PBSCs donors. Collected PBSCs were subjected to density gradient centrifugation using Lymphoprep (STEMCELL Technologies) to remove dead cells and debris, washed three times with RPMI medium, and used for the analysis.”

Reviewer #1 comment 17a. Figure 8: So here is your explanation why the bone marrow cells did not perform well in the T-cell proliferation assay in figure 7. The gating was not optimal and only CD235⁺ CD71⁺ later stage basophilic erythroblasts were taken in that experiment whereas the PBMC derived erythroid cells at day 13 indicate that it is the erythroblasts that are CD71^{high}, CD235^{low} that actually do the suppressive job.

Authors' response:

We have isolated CECs from bone marrow aspirates using immunomagnetic positive selection with an anti-CD71 antibody. Thus, we have isolated a whole heterogeneous population of CECs at different developmental stages. Plot presenting CD71 and CD235a levels in CECs isolated from human bone marrow is demonstrated in Supplementary Figure 16g.

We agree with the reviewer – experiments with PBMC-derived CECs demonstrated that CD71^{high}CD235a^{mid} are the most potent immunosuppressive CECs. We have highlighted this in the revised manuscript.

Line 325 “We found that of all CECs developmental stages, cells at the stage of CD71^{high} CD235a^{mid} most strongly inhibited T-cells proliferation.”

Line 414 “We showed that human CECs acquire immunomodulatory properties during erythroid differentiation and are the most potent in an early stage characterized by the CD71^{high} CD235a^{mid} phenotype.”

Moreover, we discussed the differences in T-cell suppression by bone marrow-derived CECs and PBMC-derived CECs.

Line 421 “Nonetheless, the suppression of T-cells by bone marrow CECs was substantially weaker than that of CD71^{high} CD235a^{mid} CECs from *ex vivo* culture. This may be explained by the fact that in human bone marrow CECs at the earliest stages of differentiation are still much less abundant than late-stage CECs^{28, 45}”

Reviewer #1 comment 17b. So, what is missing in figure 8 is the inhibitor assay using ARG inhibitors and NAC on the human EPC.

Authors’ response:

We have added the results of T-cells proliferation assay with human erythroid cell lines (ratio 1:2) with ARG inhibitor and ROS inhibitor (NAC) in the Fig. 9e and Fig. 9i. Quantification of this data is provided in Supplementary Fig. 18.

Figure 9. e, Representative histograms of CTV – BV421 fluorescence of CD4⁺ T-cells co-cultured with erythroid cell lines in a 1:2 ratio in the presence of ARGi (OAT-1746, 1.5 μM) and ROSi (NAC, 200 μM). **i,** Representative histograms of CTV – BV421 fluorescence of CD8⁺ T-cells co-cultured with erythroid cell lines in a 1:2 ratio in the presence of ARGi (OAT-1746, 1.5 μM) and ROSi (NAC, 200 μM).

Moreover, we have included data confirming ARG- and ROS-dependent suppression of T-cells proliferation by PBMC-derived CECs at the CD71^{high} CD235a^{mid} stage (Supplementary Fig. 23).

Supplementary Fig. 23a,b, Proliferation triggered by α CD3/ α CD28 of CTV-labelled CD4⁺ (a, n=4) and CD8⁺ (b, n=4) T-cells co-cultured with PBMC-derived CECs at a 1:2 ratio. CECs were at the stage of CD71^{high}CD235a^{mid} cells on day 13 of differentiation. Data show means \pm SEM. Each point in a, b represents data from one individual patients. n values are the numbers of patients used to obtain the data. The source data underlying Supplementary Fig. 21a,b are provided as a Source Data file.

Reviewer #1 comment 18. In addition, spleens are a bona fide immunological site whereas human bone marrow is less so. What does it mean that EPC or erythroid precursors cells inhibit T-cell proliferations in the bone marrow. Is that to protect the bone marrow from largescale inflammation or something like that?

Authors' response:

Indeed, we think that one of the roles of CECs is to suppress erythropoietic niche from the inflammation that is known to exert strong inhibitory effects on CECs survival and differentiation. We have demonstrated that in addition to inhibiting proliferation, CECs potently suppress IFN- γ production by T-cells (Supplementary Fig. 22c). Previous studies demonstrated that similar role is exerted by T-regulatory cells, whose activity is critical for the suppression IFN- γ production by activated effector T-cells in bone marrow⁶. Moreover, it was demonstrated that IFN- γ secreted by activated T-cells induces apoptosis of CECs⁷. Therefore, we proposed that the suppression of T-cells by CECs primarily is responsible for the prevention of erythropoiesis suppression by inflammation.

Line 426 “The exact role of transient immunomodulatory properties of CECs remains elusive. It was suggested that expansion of CECs in neonates provides tolerance to harmless antigens, including the commensal microbiota⁴, and minimize damage caused by inflammation in the intestines⁴, liver⁷, and lungs⁴⁶ during the first days of postnatal life. In adults, the role of CECs seems to be similar. Recent studies demonstrated that stress erythropoiesis is a key inflammatory response⁴⁷, therefore, expansion of CECs may prevent progression to chronic inflammation. Indeed, the transfer of CECs suppressed inflammatory response and attenuated the wasting syndrome in murine models of colitis¹¹. In cancer, which is characterized by a chronic inflammation⁴⁸, CECs substantially expand and suppress immune response facilitating tumor growth and increasing the susceptibility to pathogens¹⁰. On the other hand, impaired immunoregulatory properties of CECs may exacerbate the damage caused by inflammation⁴⁹. Moreover, CECs by suppressing the production of IFN- γ , a crucial

inflammatory cytokine and a potent inhibitor of erythropoiesis^{50, 51}, may allow the maintenance of erythropoiesis and play a role in preventing systemic inflammation.”

Reviewer #1 comment 19. Please comment in the discussion the difference between human and mice (as the authors know, stress erythropoiesis in humans does not occur in the spleen).

Authors’ response:

We now discuss in more detail the differences between humans and mice in the context of CECs and erythropoiesis.

Line 442 “Importantly, most of the knowledge on CECs has been build based on murine models⁵. However, several crucial divergences between mice and humans may limit the translational character of CECs studies in mice, that include differences in stress erythropoiesis⁵². In mice, stress erythropoiesis primarily takes place in the spleen and relies on the expansion of early-stage CECs⁵³. Thus, suppressive CECs may interact with immune cells in the spleen, which is also an active immune organ⁵⁴. We observed that T-cells in anemic mice are rather locally suppressed in the spleen, while not affected in the lymph nodes, similar to the conditions described for neonates⁴. In contrast, in humans stress erythropoiesis primarily involves the bone marrow, and expansion of CECs in extramedullary sites is rather occasional^{5, 52}. Moreover, CECs at the earliest stages of differentiation are relatively rare in healthy individuals⁴⁵. Thus, it seems that CECs may have the most significant immunoregulatory role under conditions that are characterized by the impaired erythroid differentiation and robust enrichment of early-stage CECs, that include cancer⁵⁵.”

Reviewer #1 comment 20. Discussion:

First sentence. Well, murine yes, but human anemia, this they did not show.

Authors’ response:

We have modified this sentence:

Line 354 “In this study, we demonstrate that suppression of T-cells is a general feature of murine and human CECs.”

Reviewer #1 comment 21. Please refrain from writing that EPCs are expanded in anemic patients as those are just retics in peripheral blood (if they would look in bone marrow they would indeed have found this). in fact as indicated above these are probably reticulocytes that will not suppress the T-cells as they are too late in differentiation. Moreover, even if these were nucleated erythroid precursors cells these would be too late in diff to exert their effect! As indicated by the authors in figure 8. So, sentence 339 needs to be rewritten or deleted.

Authors’ response:

We agree with the reviewer. We have modified this sentence accordingly.

Line 397 “We demonstrated that CECs expand in anemic patients and may suppress the production of cytokines by T-cells.”

Reviewer #1 comment 22. Discussion: 354: “We showed that human EPCs acquire immunomodulatory properties during erythroid differentiation and are the most potent in CD71^{hi}CD49d^{hi}CD44^{hi}CD45⁺ EPCs.” I did not see this graph!

Authors’ response:

We have modified this sentence to increase the clarity of the message.

Line 414 “Here, we showed that human CECs acquire immunomodulatory properties during erythroid differentiation and are the most potent in an early stage characterized by the CD71^{high} CD235a^{mid} phenotype.”

Minor points:

Reviewer #1 comment 23. Intro:

Lines 71 to 73. Please add the word “in mice” as stress erythropoiesis in humans occurs in the bone marrow.

Authors’ response:

We have corrected this sentence.

Line 72 “In mice, when steady-state erythropoiesis becomes insufficient to meet increased tissue oxygen demands, CECs are released from the bone marrow to the circulation and expand in the extramedullary hematopoietic sites. In humans, increased RBCs damage or loss of blood is compensated by increased erythropoietic activity in the bone marrow.”

Reviewer #1 comment 24. Figure 2c only shows one of the conditions, please add all conditions.

Authors’ response:

We have included representative purity plots of CECs isolated from all conditions (non-hemolytic anemia (NHA) mice and hemolytic anemia mice (phenylhydrazine-induced, HA-PHZ mice and anti-TER119-induced, HA-TER119) in revised Supplementary Fig. 4.

a

Supplementary Fig. 4a, Representative plots of CD71 and TER119 of CECs isolated from the spleen of anemic mice (NHA, HA-PHZ, HA-TER119) using positive immunomagnetic selection with anti-CD71 antibody.

Reviewer #1 comment 25. During co-culture what happens to the erythroid cells with respect to ARG1/2 expression, differentiation and viability?

Authors' response:

We have performed additional analysis and found that there were no significant changes in the expression of ARG1 (Rebuttal Fig. 4a) nor ARG2 (Rebuttal Fig. 4b). The viability of CECs was decreased from ~90% to ~75% in co-culture with α CD3/ α CD28-stimulated CD4⁺ T-cells and slightly decreased in co-culture with α CD3/ α CD28-stimulated CD8⁺ T-cells (Rebuttal Fig. 4c). These effects may be caused by the production of cytokines, including IFN- γ by activated T-cells that are known to induce apoptosis of erythroid cells^{1,8}. Moreover, we observed changes in the levels of CD71 only in CECs cocultured with resting CD8⁺ T-cells (Rebuttal Fig. 4d), but no significant changes in the levels of CD235a (Rebuttal Fig. 4e). Therefore, we did not observe significant effects of T-cells on CECs differentiation in our co-culture assays (Rebuttal Fig. 4f).

Rebuttal Fig. 4 Effects on T-cells on ARG levels, viability and differentiation of CECs. **a,b**, The levels of ARG1 (**a**) and ARG2 (**b**) of CECs after 72h of co-culture with CD4⁺ or CD8⁺ T-cells based on the intracellular staining (n=4). **c**, Viability of CECs after 72h of co-culture with CD4⁺ or CD8⁺ T-cells based on the ZombieNIR staining (n=4). **d,e**, The levels of CD71 (**d**) and CD235a (**e**) of CECs after 72h of co-culture with CD4⁺ or CD8⁺ T-cells (n=4). **f**, Representative plots of CD71 and CD235a of CECs after 72h of co-culture with CD4⁺ or CD8⁺ T-cells. P values were calculated with repeated measures ANOVA with Holm-Sidak's post-hoc. Data show means ± SEM.

Reviewer #2 (Remarks to the Author):

The paper by Grzywa et al identify increased expression of arginase 1 and 2 as at least one of the mechanisms used by erythroid precursors from anemic subjects to suppress T-cell functions.

The scientific premise of the study is robust and described in sufficient detail.

The experimental design is also robust and involves a good number of in vitro and in vivo mouse and human experimental models. The data presented are overall of good quality and well support the conclusion.

Authors' response:

We thank Reviewer for the appreciation of our study as well as for the constructive comments that helped us improve our manuscript.

Major comment

Reviewer #2 comment 1

1) The presentation of the data is not of the high standard required for publication and needs to be greatly improved:

Reviewer #2 comment 1a) There are too many abbreviations, which make the paper difficult to read.

Authors' response:

We have rewritten some parts of the manuscript and removed unnecessary abbreviations to increase its readability. We have formatted our manuscript according to the Communications biology Style and formatting guide and we have kept acronyms and abbreviations if they appear three or more times in the text.

Reviewer #2 comment 1b) There is little attention to the methodological differences implemented to analyze murine and human cells.

Authors' response:

We have enriched the Materials and methods section to clarify the methodological differences. Specifically, we have described separately all methods use to analyze murine and human cells.

Reviewer #2 comment 1c) There is an awkward use of terminology. The authors indicate in the title and in the text EPC as erythroid progenitor cells while by the markers used for cell identification and the graphical abstract clearly indicate that they mean erythroid precursors instead.

Authors' response:

Based on nuclear staining, the cells investigated in this work include both erythroid progenitors and erythroid precursors (please see Fig. 1i). According to the comments of the Reviewers, we have modified the nomenclature of cells to CD71⁺ erythroid cells (CECs), which is a broader term that also includes erythroid cells at different developmental stages.

Reviewer #2 comment 1d) The language used to describe and discuss the results is convoluted and should be simplified.

Authors' response:

We have rewritten results section to increase the readability.

Reviewer #2 comment 1e) The numerous alternative hypothesis discussed while interpreting the results reduce the perception of the quality of the experimental design.

Authors' response:

We have removed irrelevant alternative hypothesis to clarified the message of our manuscript.

Reviewer #2 comment 2) Mice phlebotomy. Anemia was induced in mice by phlebotomy of 100 uL of blood. This volume corresponds to 0.5% of the all blood volume of a mouse. A blood donation, 500 mL, corresponds to 10% of the blood volume of the donor. Do the authors mean that blood donation suppresses the immune functions of the volunteer? The long history of follow up of regular blood donors has not provided any clinical evidence that the procedure induces an immunosuppressive state. This important point should be clarified by analyzing the immunosuppressive potential of EPC and the immune state of T cells from regular blood donations.

Authors' response:

Based on the literature data, the average total blood volume of a mouse is about 77-80 ml/kg^{9, 10)} which gives about 1.9-2 ml per mouse (~25 g). Thus, our model of non-hemolytic anemia involves collection of at least 5% of blood volume twice. The blood donation in humans involves donation of higher percentage of total blood volume (about 10%). Indeed, there is no significant clinical data indicating that blood donation induces an immunosuppressive state. Only one study found that repeat whole-blood donors have decreased leukocytes and NK-cells compared to first-time donors, however, without differences in the cytokine levels¹¹. Indeed, it would be of a great clinical interest to analyze the immunosuppressive potential of CECs and T-cell immune state from regular blood donors. However, due to the COVID-19 pandemic, we were unable to collect the blood samples from the appropriate group of regular blood donors and first-time blood donors. Moreover, the ongoing COVID-19 vaccination campaign with different types of vaccines could bias the results.

Minor comment

Reviewer #2 comment 31) The patients included in the study are generically indicated as anemic or not anemic. Their diagnosis should be clarified to assure that the patients included in the two groups are clinically homogeneous.

Authors' response:

We have included patients in the study only based on the presence or not of the anemia based on the WHO diagnostic criteria. We have excluded patients with proliferative diseases, including cancer, that are known to have dysregulated erythropoiesis. For the rest of patients we did not collect detailed clinical diagnosis.

Reviewer #2 comment 3) To recover from the hemolytic anemia induced by phenylhydrazine mice activate the glucocorticoid receptor pathway and experience increased levels of cortisol in their blood. Given the strong immuno-suppressive role of cortisol how the authors exclude that the immunosuppressive effects of EPCs from these mice is not mediated by cortisol carried over in the cytoplasm of the cells?

Authors' response:

We agree with the reviewer that it is a possible mechanism of the regulation of immune response by CECs. Indeed, it was demonstrated that mature RBCs are able to bind large amount of the cortisol¹².

Since we did not observe impaired T-cell proliferation in hemolytic anemia induced by phenylhydrazine (HA-PHZ, Fig. 2b), we chose NHA as a model to investigate this mechanism. We have induced non-hemolytic anemia using our described protocol and we have isolated CD4⁺ T-cells from the spleen of anemic (n=4) and control mice (n=4). Then, we have assessed the activation of glucocorticoid receptors by western blot using anti-phospho-glucocorticoid receptor (Ser211) and anti-total glucocorticoid receptor antibodies. However, we did not observe any significant differences in the activation of glucocorticoid receptor, which suggests that this mechanism has rather little or no function *in vivo* (Rebuttal Fig. 5).

Rebuttal Fig. 5 Induction of non-hemolytic anemia (NHA) does not activate glucocorticoid receptors (GR). **a**, CD4⁺ T-cells were isolated from the spleen of anemic (NHA, n=4) and control mice (n=4). Dexamethasone-treated mice (DEXA, 10 mg/kg, 2h) served as a positive control of GR activation (n=2). **b**, The results of densitometry analysis of blots. Relative phospho-GR was calculated as (density of phospho-GR/β-actin) / (density of total-GR /β-actin).

Reviewer #3 (Remarks to the Author):

This manuscript provides some interesting evidence in support of previously published articles in the field. Although this is an area of intense interest, significant modifications and improvements are required.

Reviewer #3 comment 1. One major concern is the definition of these cells as EPCs. Based on the vast literature EPCs are considered as erythroid progenitors expressing CD45, a possibility that some are nucleated. However, the authors in many plots combine CD45+ and CD45- cells as EPCs, which is incorrect. Besides, the authors throughout the manuscript refer to CD71+ erythroid cells (CECs) in multiple published articles as EPCs, however, in those articles EPCs are defined as CD45+CD71+ erythroid cells. Therefore, using the term CD71+ erythroid cells is scientifically more acceptable and comprehensive than EPCs. The term EPCs does not reflect the literature and the manuscript content.

Authors' response:

We agree with the Reviewer. We have changes the nomenclature from EPCs to CECs in the manuscript. Moreover, we have performed additional analysis to demonstrate that investigated CECs consist of both erythroid progenitors and precursors (please see Fig. 1i).

Reviewer #3 comment 2a. Secondly, the manuscript can be more focused and better organized. The method section lacks some experimental details. The ratio of CECs to T cells is inconsistent and on many occasions not mentioned. Is the ratio of 1:3 or 1:2 is physiologically relevant? Why this ratio was used but not like 1:1?

Authors' response:

We have enriched the materials and methods section. We have ensured that the ratio of T-cells to CECs is mentioned in every figure legend. In general, ratio of T-cells to CECs used in the experiments is 1:2, unless otherwise stated. Moreover, we have also added graphs presenting physiological ratio of CECs to T-cells (Supplementary Fig. 3f,g).

Supplementary Fig. 3f, Ratio of CECs to CD4⁺ T-cells in the spleen of control (n=5) and anemic mice (n=5). **g** Ratio of CECs to CD8⁺ T-cells in the spleen of control (n=5) and anemic mice (n=5).

Furthermore, we have performed additional experiments, which demonstrated that *ex vivo* CECs even at a ratio typical for anemia in our studies (1:10) completely inhibit T-cells proliferation (Fig. 5e,f).

Figure 5e,f, Proliferation of α CD3/ α CD28-stimulated CD4⁺ T-cells (e) or CD8⁺ T-cells (f) co-cultured with CECs isolated from NHA mice (n=4) at a ratio 1:10 (T-cells:CECs).

Reviewer #3 comment 2b It is not clear how spleens were processed, did the author use RBC lysis buffer or not?

Authors' response:

We have included a more detailed description of the murine tissue processing in the Materials and methods section.

Line 594 “Murine whole blood was collected from the inferior palpebral vein to EDTA-coated tubes. Spleens were isolated from mice and mechanically dispersed by pressing gently through a 70 μ m nylon cell strainer using a rubberized 1 ml syringe piston. Murine bone marrow was isolated from the femur by the centrifugation method. Briefly, femurs were dissected, followed by the removal of any muscle or connective tissue. The condyles and epiphysis were removed and a cleared bone was placed in microcentrifuge tubes followed by centrifugation at 2500 x g for 30 sec. Bone marrow cells were filtered through a 70 μ m nylon strainer and used for further analysis.”

We did not perform RBC lysis in any flow cytometry analysis that involved CECs or RBCs analysis. We have included this information in the Materials and Methods section.

Line 602 “No erythrocyte lysis was performed in flow cytometry analysis or experiments that involved analysis or isolation of CECs or RBCs. If only other types of cells were analyzed using flow cytometry, erythrocytes were lysed using ACK (Ammonium-Chloride-Potassium) Lysing Buffer (Thermo Fisher Scientific), according to the manufacturer’s protocol.”

Reviewer #3 comment 3. Almost 20 supplementary figures are not mentioned in the text and as a reviewer/reader you are unaware of this until you spend hours on the paper and finally you find them at the end. I believe this is any journal’s policy to incorporate all the supplementary figures in the text.

Authors' response:

These figures (Supplementary Fig. 26-44) present gating strategies used to analyze the flow cytometry data. These figures are mentioned in Materials and Methods section in the Flow cytometry subsection. It complies with the Journal Guidelines and the inclusion of gating strategies to supplementary figures was confirmed by us in Reporting summary.

Reviewer #3 comment 4. Thirdly, although authors have invested a substantial amount of time and resources in this manuscript, it's not up to the standard of a scientific article. This is evident in inconsistency in the flow gating strategy and lack of experimental details. Normally, Fig. 1a should provide the reader with a positive impression of the work, unfortunately, this is not the case in this article.

Authors' response:

We have corrected all flow cytometry gating strategies and revised our manuscript accordingly. Revised figure legends provide all crucial experimental details. Moreover, we have substantially enriched the materials and methods section to include all details of the experimental procedures.

Reviewer #3 comment 5a. Overall my points are:

Figure 1a should be considered as the most important part of the paper since all the data are based on this comparison. However, the quality of flow plots is poor and as you can see there is a cut in the flow cytometry plots, so the percentages are inaccurate and the graphs that are made based on these percentages might not be accurate. The authors could have reduced the voltage for CD71.

Authors' response:

We have modified the y-axis of the plots presented in Fig. 1a.

Reviewer #3 comment 5b. The other issue is that it is unclear whether such staining was performed on splenocytes after using RBC lysis buffer or it's in the presence of mature RBCs. Based on the high % of TER119 cells I suggest its done on total splenocytes but needs to be mentioned. Also, the % of mature RBCs can skew the % in adults versus newborns as shown by a lower frequency of TER119+ cells in the neonatal spleen.

Authors' response:

We did not perform RBC lysis in any flow cytometry analysis that involved CECs or RBCs analysis, as mentioned above. Moreover, we have indicated in the figure legend, that analyses were performed on total splenocytes, according to the reviewer's suggestion.

Reviewer #3 comment 6. Figure 1c, is misleading, of course a 3 day old pup has 2-3 x10⁶ splenocytes compared to an adult mouse which has 100 x10⁶ splenocytes. Thus, total CECs even if they constitute 70% of splenocytes they are lower when comparing 2-3 to 100 x10⁶?

Authors' response:

We agree with the reviewer. Therefore, in Fig. 1 we present both the frequency of CECs of live cells (Fig. 1b) and the absolute number of CECs per spleen (Fig. 1c).

Reviewer #3 comment 7. Fig 1d, the percentage of CD45⁺ vs CD45⁻ EPCs are shown. However, there is no representative plot to support the data.

Authors' response:

We have included representative plot of CD45 staining of CECs in the spleen in the Figure 1d. Moreover, we included the plot of CD45 staining of TER119⁻ cells, as CD45 positive control.

Figure 1d, Percentages of CD45.2⁻ and CD45.2⁺ cells within CECs (CD71⁺TER119⁺) population (n=5). Representative plot of CD45 level in CECs and TER119⁻ cells in the spleen of NHA mouse.

Reviewer #3 comment 8. Fig 1e, the authors use CD44 and cell size to categorize CECs into different subsets and they referenced reference # 13. However, the gating strategy in reference #13 is placed on distinct cell populations, which is not the case here. The gate III is placed on two different/distinct cell populations. The gate for II and III population should be moved to the right and data should be re-analyzed again.

Authors' response:

We have changed the representative plot of gating strategy used for the analysis of the data presented in Fig 1f since the previous one was not correct. Fig. 1e represents the proper gating strategy of data presented in Fig. 1f. Moreover, to provide the correctness of gating strategy, we have presented plots of gating strategies of different developmental stages for each anemia model in Rebuttal Fig. 6. Importantly, the gating of developmental stages was adjusted in each experiment separately, due to the differences in the population distribution.

Figure 1e, Gating strategy for CECs developmental stages based on CD44 expression and cells size¹³.

Rebuttal Fig. 6 The representative plots of gating strategy of CECs developmental stages in different anemia models.

Gating strategy of CECs developmental stages based on CD44 level and relative cell size (FSC). Data corresponds to the Fig. 1f.

Reviewer #3 comment 9. Fig 2c shows the purity plot for CECs. Besides the cut in the gating population, the purity is 89.5% isn't that great. This shows that there is contamination by other cells that may have suppressive functions such as T-regs. At least the authors should show the nature of the other 10% of cells.

Authors' response:

We have performed additional analysis to investigate the nature of the other cells isolated together with CECs (Rebuttal Fig. 7a). We have found that the fraction of GR-1⁺, CD11b⁺, FoxP3⁺ or F4/80⁺ cells with a well-established immunoregulatory role was insignificant (Rebuttal Fig. 7b). The CECs purity of 89-96% is similar to the purity of CECs isolated by other teams^{14, 15}. Moreover, in the ratio of 1:2 (T-cells : CECs), the ratio of not-CECs as contamination in isolated CECs was lower than the 5:1 (T-cells : not-CECs). Therefore, their immunoregulatory effects are most likely negligible.

Rebuttal Fig. 7 The percentage of immunoregulatory immune cells in the CD71⁺TER119⁺ fraction of isolated cells.

a, Plot presenting the level of CD71 and TER119 in cells isolated from the spleen of NHA mice using immunomagnetic selection with anti-CD71 antibodies. **b**, Histograms presenting the GR-1, CD11b, FoxP3 and F4/80 levels in the fraction of not-CECs (CD71⁺TER119⁺) cells isolated with CECs. The gates for the positive cells were set up based on the fluorescence-minus-one (FMO) controls.

Reviewer #3 comment 10. Antigen-specific T cell assays are very interesting; however, the authors do not explain the rationale for using the ratio of 1:2 (T cell: CECs)? Is this physiologically relevant? This should be mentioned and discussed.

Authors' response:

Using a ratio of cells that would reflect in vivo conditions is problematic due to several reasons. First, the ratio in normal vs anemic mice is different. Moreover, we choose a 1:2 ratio for ex vivo experiments as we were afraid that using a large number of CECs (vs T-cells as in 1:8 for CD4⁺ and 1:15 for CD8⁺ ratio observed in anemic mice) would be associated with additional nonspecific effects associated with excessive crowding of cell in the co-cultures. Thus, by using a lower T-cells to CECs ratio we have shown that the suppressive effects of CECs are very strong. We have added graphs presenting the ratios of CECs to T-cells (Supplementary Fig. 3f,g) and performed additional experiments that demonstrated that *ex vivo* CECs at the ratio that is physiological for anemic spleen (1:10) completely inhibit T-cells proliferation (Fig. 5e,f).

Reviewer #3 comment 11. In lines 139 and 140, it has been mentioned that ROS levels in murine EPCs were significantly lower than in myeloid -cells and T-cells. However, the data in fig 3b shows that CD45⁺ cells have almost the same level of ROS compared to CD3⁺ cells and CD11b⁺ cells. Since CD45⁺ EPCs have the main functional suppressive function in CECs, it is recommended that the authors make another conclusion based on the production of ROS on CD45⁺ EPCs vs other cells.

It's unclear how EPCs are defined here since some of them are CD45⁺. Do CD45⁺ cells in this plot contains CD45⁺CECs?

Authors' response:

We have clarified the titles of Figure 3b and we have added in the figure legend that leukocytes (previously CD45⁺ cells) were defined as CD45⁺TER119⁻ to exclude CD45⁺ CECs. Therefore, the levels of ROS in CD11b⁺ and CD3⁺ cells is similar to CD45⁺ cells (leukocytes).

Figure 3b, Mean Fluorescence Intensity (MFI) of CellROX Green – FITC in CECs ($CD71^{+}TER119^{+}$), $CD45^{+}$ CECs ($CD45^{+}CD71^{+}TER119^{+}$), leukocytes ($CD45^{+}TER119^{-}$), T-cells ($CD45^{+}CD3e^{+}$), myeloid cells ($CD45^{+}CD11b^{+}$) (n=18).

We have also included graphs presenting ROS levels in $CD45^{+}$ and $CD45^{-}$ CECs in the Supplementary Figure 5e, 5f.

Supplementary Fig. 3e,f, ROS levels in $CD45^{+}$ CECs and $CD45^{-}$ CECs based on mean fluorescence intensity (MFI) of CellRox Green – FITC (e) and 2',7'-dichlorodihydrofluorescein diacetate (DCFDA) – FITC (f) in anemic (NHA n=6, HA-PHZ n=6) and control mice (n=6).

Reviewer #3 comment 12. Supplementary Fig 3, lacks representative plots for TNF- α expression in $CD11b^{+}$ with proper controls.

Authors' response:

We have added representative plots for TNF- α in $CD11b^{+}$ cells in Supplementary Fig. 3.

Reviewer #3 comment 13. Line 125-126, Since in the spleen of anemic mice the expansion of EPCs was the most substantial (Supplementary Fig. 3d). However, the only used marker in this graph is CD71 and how the authors are confident that all CD71⁺ cells are EPCs? Of course, any activated immune cell can express CD71. Please be careful in data interpretation.

Authors' response:

In this experiment, most of the CD71⁺ cells in the spleen were negative for the immune lineage markers and a negligible fraction of CD71⁺ cells were activated immune cells (Rebuttal Fig. 8), thus, they were in the vast majority CECs.

Rebuttal Fig. 8. The expression of immune lineage markers in CD71⁺ cells and respective immune cell population. a,b,c,d, The levels of CD45 (a), CD3ε (b), CD11b (c) and CD11c (d) in CD71⁺ cells and respective immune cell population (a – leukocytes, b – T-cells, c – myeloid cells, d – dendritic cells) in the spleen. Data correspond to Supplementary Fig. 3e.

Nonetheless, we have modified this sentence to ensure more careful data interpretation.

Line 130 “Since the expansion of CD71⁺ cells was the most substantial in the spleens of anemic mice (Supplementary Fig. 3e) and the ratio of CECs number to T-cells number was

significantly increased in anemia (Supplementary Fig. 3f,g), we hypothesized that CECs might be responsible for T-cells suppression.”

Reviewer #3 comment 14. Supplementary Fig. 4a, two subpopulations of CECs as ROS low and ROS high are in fact erythroid precursors (CD45^{low/-}) and erythroid progenitors (CD45^{+/high}). The authors could have done CD45 staining the differentiate these two subpopulations.

Authors’ response:

Subpopulation of CECs with high ROS levels was indeed at the earlier stages of differentiation than CECs with low ROS levels (please see Supplementary Fig. 5a, right panel – CD44 and FSC of ROS^{high} and ROS^{low} CECs). However, CECs subpopulations cannot be divided only based on CD45 expression, since both CD45⁺ and CD45⁻ CECs consist of ROS^{high} and ROS^{low} cells (Rebuttal Fig. 9). We have also included graphs presenting ROS levels in CD45⁺ and CD45⁻ CECs in the Supplementary Figure 5e, 5f).

Rebuttal Fig. 9 ROS level in CD45⁺ CECs and CD45⁻ CECs.

a, Gating strategy of CECs and RBCs and a representative plot of CD45 in CECs. **b,** Representative histograms of ROS levels based on CellROX Green – FITC fluorescence in RBCs and CECs from the spleen of NHA mouse. Data corresponds to the Supplementary Fig. 5.

Reviewer #3 comment 15. Fig 3a, the representative plot shows stained cells but it’s unclear the source of cells? NHA or HA-PHZ and having RBCs would be a good negative control to be included.

Authors’ response:

The representative plot shows ROS in CECs from NHA mouse. We have included this information in the manuscript:

Line 1258 “Histograms show representative fluorescence of CellROX Green – FITC in CECs and RBCs from the spleens of the NHA mouse.”

Moreover, we have added the representative plot of ROS levels in RBCs from NHA mouse, according to the Reviewer’s suggestion.

Reviewer #3 comment 16. Fig 3c and 3d show the expression of Arginase 2 and 1 in different study groups. However, there is no representative plot to support the graphs. More importantly, Args are normally presented not by % but by intensity. These data should be presented as the MFI comparing different groups.

Authors' response:

We agree with the reviewer. We have included both graphs presenting MFI of ARG1 (Fig. 3g) and ARG2 (Fig. 3d) and representative plots (Fig. 3d and Fig. 3h)

Figure 3c, Percentages of ARG2⁺ CECs in control mice (n=11), anemic mice (NHA, n=5; HA-PHZ, n=11; HA-TER119, n=11), neonatal mice (n=5), and isotype control-IgG-treated mice (control-IgG, n=7). **d**, Mean Fluorescence Intensity (MFI) of ARG2-APC in CECs from control mice, anemic mice (NHA, HA-PHZ, HA-TER119) neonatal mice, and isotype control-IgG-treated mice (each group n=5). Histograms show the representative fluorescence of ARG2 – APC in CECs in different groups and unstained control. **e**, Percentages of ARG1⁺ CECs based on intracellular staining (n=5). *P* values were calculated with one-way ANOVA with Dunnet’s post-hoc test and with unpaired *t*-test for HA-TER119. **f**, Percentages of YFP⁺ CECs in reporter B6.129S4-Arg1^{tm1Lky}/J mice (controls n=4, NHA n=8, HA-PHZ n=8, neonatal n=5, control-IgG n=4, HA-TER119 n=8). *P* values were calculated with one-way ANOVA with Dunnet’s post-hoc test and with unpaired *t*-test for HA-TER119. **g**, Mean Fluorescence Intensity (MFI) of YFP – FITC in CECs of reporter B6.129S4-Arg1^{tm1Lky}/J mice in control mice (n=4), anemic (NHA n=8, HA-PHZ n=4, HA-TER119 n=8), neonatal mice (n=5) and isotype control-IgG-treated mice (n=4). *P* values were calculated with one-way ANOVA with Dunnet’s post-hoc test. **h**, Representative fluorescence of ARG1 – YFP in CECs in reporter B6.129S4-Arg1^{tm1Lky}/J control mice and anemic mice (NHA, HA-PHZ). Background fluorescence of YFP in CECs from wild-type C57Bl/6 mice presented as negative control.

Reviewer #3 comment 17. Similarly, the data in supplementary fig. 5 is not supported by representative plots.

Authors’ response:

We have included representative histograms of ARG1 and ARG2 fluorescence in different developmental stages of CECs as well as in mature RBCs (Supplementary Fig. 6). Moreover, we have included both MFI and percentages of ARG1/2⁺ CECs at different developmental stages.

Supplementary Fig. 6a, Representative plot of CD71 and TER119 levels in live cells in the spleen of NHA mice and gating strategy of CECs developmental stages based on CD44 level and relative cell size. **b**, Representative histograms of ARG1 expression in different developmental stages of CECs and RBCs in NHA mouse based on YFP mean fluorescence

intensity (MFI) in reporter B6.129S4-Arg1^{tm1Lky}/J mice. Negative control represents the background fluorescence of YFP in wild-type C57Bl/6 mouse. **c**, Representative histograms of ARG2 expression in different developmental stages of CECs and RBCs based on intracellular staining in NHA mouse. Unstained control represents sample unstained for ARG2. **d,e**, ARG1 expression (**d**) and percentage of ARG1⁺ cells (**e**) in different developmental stages of CECs in NHA mice based on YFP mean fluorescence intensity (MFI) in reporter B6.129S4-Arg1^{tm1Lky}/J mice (n=8). **f,g**, ARG2 levels (**f**) and percentage of ARG2⁺ cells (**g**) in different developmental stages of CECs in NHA mice based on intracellular staining (n=4).

Reviewer #3 comment 18. The q-PCR data in fig 6A shows a significant difference between groups. However, considering the high variability of results and high overlap, it seems that the difference shouldn't be significant. It is suggested that authors reanalyze q-PCR data.

Authors' response:

The data was analyzed using Normality and lognormality tests, Shapiro Wilk test and Kolmogoroc-Smirnov test that revealed normal distribution of the data. Thus, unpaired *t*-test was used to calculate statistical significance, as indicated in the figure legend. The *P*-value calculated using unpaired *t*-test was 0.0310 for Arg1 and 0.0320 for Arg2.

Reviewer #3 comment 19. In line 186-189, it is mentioned that incubation of recombinant ARG1 with PHZ in the presence of oxygen led to a significant increase in the carbonylation of the enzyme that was reduced by concomitant incubation with N- acetylcysteine. However, this is not supported by what has been shown in Fig 4d that shows the addition of NAC results in a partial reversal of the carbonylation of the enzyme which was not statistically significant. It is recommending that the authors provide more convincing data regarding the effect of PHZ on ARG1 oxidative changes.

Authors' response:

We have performed additional experiments to investigate this issue. However, we were unable to demonstrate that the effects of PHZ on ARG1 are mediated by oxidative changes (Supplementary Fig. 9). Therefore, we remove this conclusion from our manuscript.

Supplementary Fig. 9 a-b, YARG mice were injected intraperitoneally (i.p.) three days before harvest with 50 mg per kg body weight of phenylhydrazine (PHZ) and three times every 24h with 150 mg per kg body weight of N-acetylcysteine (NAC) (each group n=4). Percentage of ARG1⁺ CECs (**a**) and mean fluorescence intensity (MFI) of YFP-ARG1 (**b**). **c,d**, CECs were isolated from the spleen of NHA YARG mice and incubated with 100 μ M phenylhydrazine (PHZ), 100 μ M NAC and/or 100 μ g/ml catalase. The fluorescence of YFP/ARG1 was assessed after 24h by flow cytometry (**c**). Histograms shows representative fluorescence of ARG1 – YFP of CEC (**d**).

Reviewer #3 comment 20. Fig 5a shows the proliferation and expression of activation markers by CD4 T cells. However, although the graph shows some data for the proliferation of T cells in the presence and absence of EPCs, it is not clarified in the text of the paper. Are they CD4 or CD8 T cells since in the material and method section, it seems that only CD4 and CD8 T cells but not total T cells were isolated. The used ratio of T cells to CECs is not mentioned?

Authors' response:

The data presented at Fig. 5a shows the data from CD4⁺ T-cells. We have included this information in the axis title and figure description. Moreover, we have included this in the manuscript text. We have also mentioned the ratio of T-cells to CECs which was 1:2.

Line 212 “We found that CD4⁺ T-cells stimulated with anti-CD3/CD28 beads in the presence of CECs showed downregulation of activation markers CD25 and CD69, which was less pronounced for CD62L (Fig. 5a).”

Line 1311 Fig. 5 “a, Proliferation and surface markers in α CD3/ α CD28-stimulated CD4⁺ T-cells co-cultured with CECs isolated from NHA mice (n=4) at a ratio 1:2 (T-cells:CECs). *P*-value was calculated with an unpaired t-test.”

Reviewer #3 comment 21. Please clarify if ARGi, OAT-174619 inhibits arginase I or II?

Authors' response:

OAT-1746 is a potent inhibitor of arginase I and arginase II that inhibits both extracellular and intracellular arginases activity¹⁶. We have included this information in the revised manuscript.

Line 215 “...(ARGi, OAT-1746, a membrane-permeable, potent inhibitor of both arginase isoforms)...”

Reviewer #3 comment 22. What is the EPC-conditioned medium? The authors didn't describe in the material and method what is this medium?

Authors' response:

We thank Reviewer for this comment. CECs-conditioned medium was obtained by culturing CECs in the in the arginine-free RPMI-medium (SILAC RPMI-medium, ThermoFisher Scientific) supplemented with 150 μ M L-arginine at the density 1×10^6 cells/ml for 24h. We have included the description of CECs-conditioned medium in the revised manuscript.

Line 768 “Conditioned medium was obtained by culturing CECs in the arginine-free RPMI-medium (SILAC RPMI-medium, Thermo Fisher Scientific) supplemented with 150 μ M L-arginine at the density 1×10^6 cells/ml for 24h. Cells were centrifuged and the supernatant was collected and immediately frozen at -80°C. After thawing, the supernatant was filtered through a 0.45 μ m Syringe Filter (Wenk LabTec) and was used in experiments a 1:1 ratio with 150 μ M L-arginine RPMI SILAC medium.”

Reviewer #3 comment 23. Fig 5b shows that the addition of ARGi or ROSi completely abrogates the suppressive function of EPCs on T cell proliferation. However, the results of fig 5d show that they only partially reverse the suppressive function of EPCs (stats not provided). It is required that authors describe the reason for this discrepancy.

Authors' response:

Fig. 6c (Figure 5c in the previous version of the manuscript) presents the suppression of T-cells by CECs at ratio 1:4, not 1:2 as in the others figures. We have chosen this higher ratio to ensure observation the suppressive effects of CECs isolated from ARG2^{-/-} mice to confirm that ARGi has no effect in ARG2^{-/-} mice. Thus, it this higher CECs ratio ARGi or ROSi was not sufficient to restore T-cells proliferation alone. Nonetheless, combination of ARGi and ROSi completely restored T-cells proliferation.

Reviewer #3 comment 24. Fig 6a-b: for showing the percentage of EPCs in adults, the cells MUST be double positive for CD71 and CD235a. However, the gating strategy (b) only shows CD71+ cells. The plots do not show any CD235a+ cells? There are two major problems here; a) CD71+ cells can be a combination of EPCs and other activated cells. b) the anti-CD235a antibody was not included or did not work. These plots show that the gated population is only CD71+ cells. Consequently, all the data presented based on this gating will be questionable?

The authors have provided interesting data on CECs in anemic individuals but the gating strategy in Fig 6g is unacceptable. Its unclear why the gate is so wide to include CD71+CD235a- cells as EPCs? How authors are confident that CD71+ cells are not activated T, B, and other immune cells? Consequently, the cumulative data based on this gating strategy need to be re-evaluated. Interestingly, the authors in Fig. 7a have used the correct gating strategy for EPCs. Now I wonder why there is inconsistency in the gating strategy for the same cell type (EPCs) by simply looking at Fig 6a, 6g and 7a!

Authors' response:

We agree with the Reviewer. We have corrected gating strategy and re-analyzed the data. The data presented in Figure 7 was obtained based on the gating strategy presented in Rebuttal Fig. 10a. We also presents the histograms of CD71 and CD235a of cells gated as CECs compared to unstained control (Rebuttal Fig. 10b).

Rebuttal Fig. 10. Revised gating strategy used for the analysis of the data presented at Figure 6. a, Gating strategy of human CECs in whole blood samples. **b,** The fluorescence intensity of CD71 – PerCP-Cy5.5 and CD235a – PE-Cy7 in CECs is based on the presented gating strategy. Unstained sample was used as a staining control.

Reviewer #3 comment 25. Fig. 6h lacks representative plots for IFN-r expression with appropriate controls.

Authors' response:

We have added representative plots of IFN- γ levels in T-cells presented in Figure 7h that shows unstimulated and α CD3/ α CD28-activated T-cells as well as fluorescence of isotype control-stained cells (Mouse IgG1 – APC) (Figure 7i).

Reviewer #3 comment 26. Also, the figure legends stated that cells were stimulated in the presence of a Golgi blocker for 12hrs? More than 6 hr incubation, the Golgi blocker kills the cells? Representative plots with viability staining is required, based on the manufacturing instruction 6 hr is the max time.

Authors' response:

We have used BD GolgiStop™ that according to the manufacturer's technical data sheet can be kept in cell culture up to 12 hours. In our experiments, cell viability did not change after 12 hours of incubation with BD GolgiStop™. Mean cell viability after 12 hrs of culture in the presence of BD GolgiStop™ was >93% (range 84.07% - 99.37%) (Rebuttal Fig. 11). We have included this information in Materials and methods section.

Rebuttal Fig. 11. Cells viability after 12 hrs of culture in the presence of BD GolgiStop™. a, Percentage of live cells based on ZombieNIR staining (ZombieNIR⁺ single cells). Data from the experiments presented at Figure 7h. b, Representative plots of ZombieNIR staining of unstimulated and stimulated (α CD3/ α CD28) cells. Additional control presents mix of live cells and dead cells (liquid nitrogen-treated cells).

Reviewer #3 comment 27. Supplementary Fig. 8 the ratio of EPCs to T cells is missing.

Authors' response:

The ratio of T-cells to CECs was 1:2. We have included this information in the figure legends.

Reviewer #3 comment 28a. The authors used human erythroleukemic cell line K562, HEL92.1.7, and TF-I. However, they didn't describe at which stage of differentiation these cells were? It is recommended that they follow with a brief description of these cells' lines. This is further highlighted by the variable expression of CD71 and CD235a on these cells.

Authors' response:

We have added plots of CD71 and CD235a levels in the Fig. 9a. Indeed, the levels of CD71 and CD235a vary between cell lines substantially, which may lead to wrong conclusions. Therefore, we have performed additional experiments and analyzed each cell line separately by flow cytometry. In the revised figure, we have attached histograms presenting CD235a and CD71 staining fluorescence together with corresponding isotype controls. Moreover, we have included a brief description of these cell lines.

Reviewer #3 comment 28b. Again, for the proliferation assay, please justify why the 3:1 ratio of EPCs to CD4 or CD8 T cells has been used.

We have modified Figure 9b-d and Figure 9f-h to demonstrate Percentage of CTV^{low} CD4⁺ and CD8⁺ T-cells in ratio 1:1, 1:2 and 1:5 to allow the comparison with data presented in Figure 10b and Figure 10c.

Reviewer #3 comment 28b. By the way, in Fig 8, it is mentioned that EPCs with suppressive function have high CD71 but fig 13a shows that K562 cells have low CD71, so how they have a suppressive function?

Indeed, K562 are CD71⁺ positive, however, its levels are relatively lower than in other two cell lines. Thus, when analyzed together it seemed that the CD71 levels in K562 are negligible. Therefore, we have modified Fig. 9a (attached above).

Reviewer #3 comment 29. There is another issue that the authors did not reference their supplemental from the Supplemental fig 19 to 36 in the text. This is hard to follow where/if any representative plot is provided for particular staining.

Authors' response:

Supplemental figures 19 to 36 represent flow cytometry gating strategies and are referenced in Materials and Methods section, subsection Flow cytometry. These figures are mentioned in Materials and Methods section in the Flow cytometry subsection. It complies with the Journal Guidelines and the inclusion of gating strategies to supplementary figures was confirmed by us in Reporting summary.

References

1. Grzywa TM, Nowis D, Golab J. The role of CD71+ erythroid cells in the regulation of the immune response. *Pharmacology & Therapeutics* **228**, 107927 (2021).
2. Rodriguez PC, *et al.* Arginase I Production in the Tumor Microenvironment by Mature Myeloid Cells Inhibits T-Cell Receptor Expression and Antigen-Specific T-Cell Responses. *Cancer Research* **64**, 5839-5849 (2004).
3. Otsuji M, Kimura Y, Aoe T, Okamoto Y, Saito T. Oxidative stress by tumor-derived macrophages suppresses the expression of CD3 zeta chain of T-cell receptor complex and antigen-specific T-cell responses. *Proc Natl Acad Sci U S A* **93**, 13119-13124 (1996).
4. Huang P, *et al.* Putative regulators for the continuum of erythroid differentiation revealed by single-cell transcriptome of human BM and UCB cells. *Proceedings of the National Academy of Sciences* **117**, 12868-12876 (2020).
5. Hu J, *et al.* Isolation and functional characterization of human erythroblasts at distinct stages: implications for understanding of normal and disordered erythropoiesis in vivo. *Blood* **121**, 3246-3253 (2013).
6. Chopra M, Langenhorst D, Beilhack A, Serfling E, Patra AK. Interleukin-2 critically regulates bone marrow erythropoiesis and prevents anemia development. *Eur J Immunol* **45**, 3362-3374 (2015).
7. Hou Y, *et al.* Radiotherapy and immunotherapy converge on elimination of tumor-promoting erythroid progenitor cells through adaptive immunity. *Science Translational Medicine* **13**, eabb0130 (2021).
8. Felli N, *et al.* Multiple members of the TNF superfamily contribute to IFN-gamma-mediated inhibition of erythropoiesis. *J Immunol* **175**, 1464-1472 (2005).
9. Mitruka BM, Rawnsley HM, Vadehra BV. *Clinical Biochemical and Hematological Reference Values in Normal Experimental Animals*. Masson Pub. USA (1977).
10. Harkness JE, Turner PV, VandeWoude S, Wheeler CL. *Harkness and Wagner's Biology and Medicine of Rabbits and Rodents*. Wiley (2010).
11. Lange S, Riggert J, Humpe A, Dittmann J, Simson G, Köhler M. [Immunologic effects of blood donation]. *Beitr Infusionsther Transfusionsmed* **33**, 93-97 (1996).
12. Farese RV, Plager JE. The in vitro red blood cell uptake of C-14-cortisol; studies of plasma protein binding of cortisol in normal and abnormal states. *J Clin Invest* **41**, 53-60 (1962).
13. Chen K, Liu J, Heck S, Chasis JA, An X, Mohandas N. Resolving the distinct stages in erythroid differentiation based on dynamic changes in membrane protein expression during erythropoiesis. *Proc Natl Acad Sci U S A* **106**, 17413-17418 (2009).
14. Shahbaz S, *et al.* CD71+VISTA+ erythroid cells promote the development and function of regulatory T cells through TGF-beta. *PLoS biology* **16**, e2006649 (2018).

15. Elahi S, *et al.* Immunosuppressive CD71+ erythroid cells compromise neonatal host defence against infection. *Nature* **504**, 158 (2013).
16. Sosnowska A, *et al.* Inhibition of arginase modulates T-cell response in the tumor microenvironment of lung carcinoma. *OncoImmunology* **10**, 1956143 (2021).

REVIEWERS' COMMENTS:

Reviewer #1 (Remarks to the Author):

I must congratulate the authors with their concise response to the inquiries. This reviewer's comments have been sufficiently addressed. No further comments remain from my side.

Reviewer #2 (Remarks to the Author):

All my previous comments are addressed. No additional comments.

Reviewer #3 (Remarks to the Author):

I appreciate the authors for their efforts to bring the quality of their article to a more accepted level. Obviously, the manuscript has substantially improved. However, some improvements are needed.

a) The authors have highlighted the important role of CECs in T cell suppression in the bone-marrow, which is similar to regulatory T cells function. However, the authors did not cite the important article on the cross-talk between CECs and regulatory T cells. This article should be included and discussed (Shahbaz et al. PLOS Biology 2018).

b) Since one major aspect of this manuscript is on the impact of anemia, extra medullary erythropoiesis and the expansion of CECs in anemic condition. I recommend including and discussing the article (by Mashhuri et al. Frontiers in Immunology 2021) which highlights the important role of CECs in anemia/infection.

c) The reference #3 must be corrected, First and last names are switched! It must be presented as Elahi S, and Mashhour S.

d) Figure 1a, is it possible for the authors to reduce the voltage for PerCp cy5.5 (Cd71) preventing the cut in the cell population? In particular, the CEC population is cut for NHA and neonatal. This can be seen throughout the manuscript, as well.

Especially, Figure 1a is very important for the impression of readers in terms of the quality of data presentation.

Reviewer #3:

I appreciate the authors for their efforts to bring the quality of their article to a more accepted level. Obviously, the manuscript has substantially improved. However, some improvements are needed.

a) The authors have highlighted the important role of CECs in T cell suppression in the bone-marrow, which is similar to regulatory T cells function. However, the authors did not cite the important article on the cross-talk between CECs and regulatory T cells. This article should be included and discussed (Shahbaz et al. PLOS Biology 2018).

Answer:

Thank you for bringing it up. We have introduced the brief summary and the citation of this work into the second paragraph of the main manuscript "Discussion" (reference No. 32).

b) Since one major aspect of this manuscript is on the impact of anemia, extra medullary erythropoiesis and the expansion of CECs in anemic condition. I recommend including and discussing the article (by Mashhuri et al. Frontiers in Immunology 2021) which highlights the important role of CECs in anemia/infection.

Answer:

We have included a response to this comment into the direct correspondence with the Editor.

c) The reference #3 must be corrected, First and last names are switched! It must be presented as Elahi S, and Mashhour S.

Answer:

We do apologize for this mistake done unwittingly while using the EndNote reference manager software. We have corrected this citation.

d) Figure 1a, is it possible for the authors to reduce the voltage for PerCp cy5.5 (Cd71) preventing the cut in the cell population? In particular, the CEC population is cut for NHA and neonatal. This can be seen throughout the manuscript, as well. Especially, Figure 1a is very important for the impression of readers in terms of the quality of data presentation.

Answer:

We have corrected the Figure 1a accordingly.